# Cross-trait multivariate GWAS confirms health implications of pubertal timing

Siquan Zhou[1,2,3], Yujie Xu[2,3], Jingyuan Xiong [1,4] ✉ & Guo Cheng[2,3,5] ✉

Pubertal timing is highly variable and is associated with long-term health outcomes. Phenotypes associated with pubertal timing include age at menarche, age at voice break, age at first facial hair and growth spurt, and pubertal timing seems to have a shared genetic architecture between the sexes. However, puberty phenotypes have primarily been assessed separately, failing to account for shared genetics, which limits the reliability of the purported health implications. Here, we model the common genetic architecture for puberty timing using a multivariate GWAS, with an effective population of 514,750 European participants. We find 266 independent variants in 197 loci, including 18 novel variants. Transcriptomic, proteome imputation and fine-mapping analyses reveal genes causal for pubertal timing, including *KDM4C*, *LEPR*, *CCNC*, *ACP1*, and *PCSK1*. Linkage disequilibrium score regression and Mendelian randomisation analysis establish causal associations between earlier puberty and both accelerated ageing and the risk of developing cardio-vascular disease and osteoporosis. We find that alanine aminotransferase, glycated haemoglobin, high-density lipoprotein cholesterol and *Parabacteroides* levels are mediators of these relationships, and establish that controlling oily fish and retinol intake may be beneficial for promoting healthy pubertal development.

Pubertal timing exhibits substantial variation across different populations, while many environmental and genetic factors contribute to the variation[1]. Elucidating the underlying mechanisms is pivotal to comprehend why advanced pubertal timing is linked to elevated risks of developing adulthood diseases, including breast and endocrine cancers, cardiovascular disease, and metabolic syndrome[2–5].

Genome-wide association studies (GWASs) have revealed pubertal timing-related loci via single-phenotype approaches, including the age at menarche for women and the age at voice break and first facial hair for men[5,6]. Furthermore, previous elucidations of genetic determinants for pubertal timing predominantly rely on the age at

menarche, a commonly recalled and extensively measured indicator of female sexual development[3], and the age at voice break, a key feature in male puberty[7]. Nonetheless, the above single-endpoint analyses did not consider the shared genetic architecture among these traits or other traits associated with pubertal timing, including the growth spurt, a distinctive characteristic of childhood development representing the onset of central puberty, and the Tanner stage, a well-established measure of pubertal development[7,8]. In contrast to single-endpoint GWASs, multivariate GWASs incorporate univariate summary statistics and enhance the discovery of biological correlates by increasing statistical power through increased effective sample sizes;

[1]West China School of Public Health and West China Fourth Hospital, Sichuan University, Chengdu, China. [2]Laboratory of Molecular Translational Medicine, Center for Translational Medicine, West China Second University Hospital, Sichuan University, Chengdu, China. [3]Key Laboratory of Birth Defects and Related Diseases of Women and Children (Sichuan University), Ministry of Education, Maternal & Child Nutrition Center, West China Second University Hospital, Sichuan University, Chengdu, China. [4]Food Safety Monitoring and Risk Assessment Key Laboratory of Sichuan Province, Sichuan University, Chengdu, China. [5]Children's Medicine Key Laboratory of Sichuan Province, Sichuan University, Chengdu, China. ✉e-mail: jzx0004@tigermail.auburn.edu; gcheng@scu.edu.cn

moreover, multivariate GWASs were recently conducted to identify genomic loci shared across ageing phenotypes[9], neuropsychiatric disorders[10], alcohol consumption behaviours[11] and externalising behaviours[12] via genomic structural equation modelling (SEM)[10]. Since a shared genetic architecture has been reported between pubertal timing in males and females[5,6,13], multivariate GWASs for pubertal timing across sex can provide valuable genetic information on pubertal timing.

Thus, we applied genomic SEM for GWASs of age at menarche, age at voice break, age at first facial hair and growth spurt to identify common genetic variants. We then performed bioannotation, including fine mapping, a transcriptome-wide association study (TWAS), a proteome-wide association study (PWAS) and cell type enrichment. We used linkage disequilibrium (LD) score regression to evaluate genetic associations between pubertal timing and health outcomes, organ development, and region-specific brain features. We used two-step Mendelian randomisation (MR) analysis to explore potential mediators between pubertal timing and adulthood health outcomes and to identify modifiable dietary factors and micronutrients (Fig. 1).

## Results

### Structural equation modelling

LD score regression revealed that univariate input GWASs (age at menarche, age at voice break, age at first facial hair and pubertal height spurt) were significantly correlated (Fig. 2a; Supplementary Data 1–2). The common factor model fit of the implied genetic covariance matrix between input GWASs with an empirical covariance matrix supported the presence of a shared genetic factor (designated mvPuberty) (comparative fit index=0.998, standardised root mean square residual=0.026) (Fig. 2b; Supplementary Data 3–4).

### Multivariate GWAS meta-analysis

We integrated SEM with individual variants, and conducted a multivariate GWAS that estimated 1,893,092 associations for mvPuberty at the SNP level (Supplementary Data 5). The mean $\chi^2$ and λGC (genomic control) values were 1.98 and 1.60, respectively, while the LD score intercept was 0.997 (se=0.024), indicating inflation attributed to polygenic heritability signals rather than population stratification bias (Supplementary Fig. 1). The study comprised an effective sample size of 514,750, with mvPuberty summary statistics constrained within MAF limits of 10% and 40% to ensure the reliability of the estimates. We identified 266 lead SNPs in 197 genomic loci ($P < 5 \times 10^{-8}$) (Fig. 2c; Supplementary Data 6–7) enriched for traits such as the risk of developing coronary artery disease, systolic and diastolic blood pressure, height, and the risk of developing insomnia according to the GWAS catalogue database (Supplementary Data 8). Eighteen loci were not reported in the four input GWASs (Table 1). Among the 18 SNPs, eight were described in previous GWASs other than the four inputs: rs5742915 was related to pubertal timing, whereas rs11556924, rs17400325 and rs3922468 were associated with traits that may affect pubertal timing (birth weight, height, and bone mineral density). Ten variants were not previously associated with any GWAS in the GWAS catalogue, including rs4076654, an intergenic SNP in the *INHBB* locus, and the other nine variants were indirectly obtained from SNPs in the LD region with the lead SNPs (Supplementary Data 8–9).

### Fine mapping

We performed fine-mapping for the 266 regions of interest by including all variants up- and downstream of the 266 lead SNPs at a 250 kb window from the mvPuberty GWAS. As a result, we identified 19 causal variants (posterior probability > 0.95), located on chromosomes 2 (rs17400325, an intergenic SNP in the *PDE11A* locus), 7 (rs11556924, an exonic SNP in the *ZC3HC1* locus), 9 (rs913588, an intergenic SNP in the *KDM4C* locus) and 15 (rs5742915, an exonic SNP in the *PML* locus). In the regional plots, we observed clear peaks at these loci with credible set variants via functional mapping and annotation of genetic associations (FUMAs) (Supplementary Figs. 2–20; Supplementary Data 10).

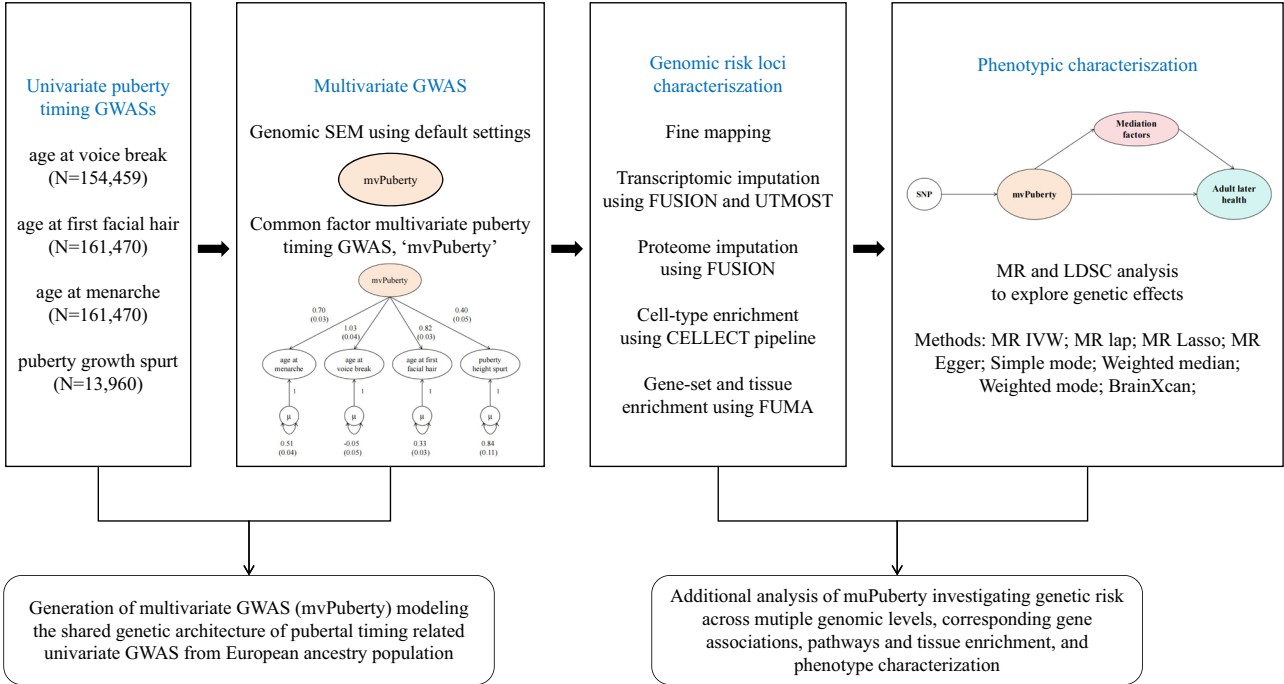

**Fig. 1 | An overview of this study's data sources, analytical flow and methodology.** Structural equation modelling, SEM; GWAS, genome-wide association study; mvPuberty, the common factor model for puberty; FUSION, functional summary-based imputation; UTMOST, unified test for molecular signatures; FUMA, functional mapping and annotation of genetic associations; LDSC, linkage disequilibrium score regression; MR, mendelian randomisation analysis; CELLECT, CELL-type Expression-specific integration for Complex Traits; IVW, inverse variance weighted; MR LASSO, MR Least Absolute Shrinkage and Selection Operator. μ reflects the residual variance in the genetic indicators for the input univariate puberty timing related GWASs not explained by the mvPuberty common factor.

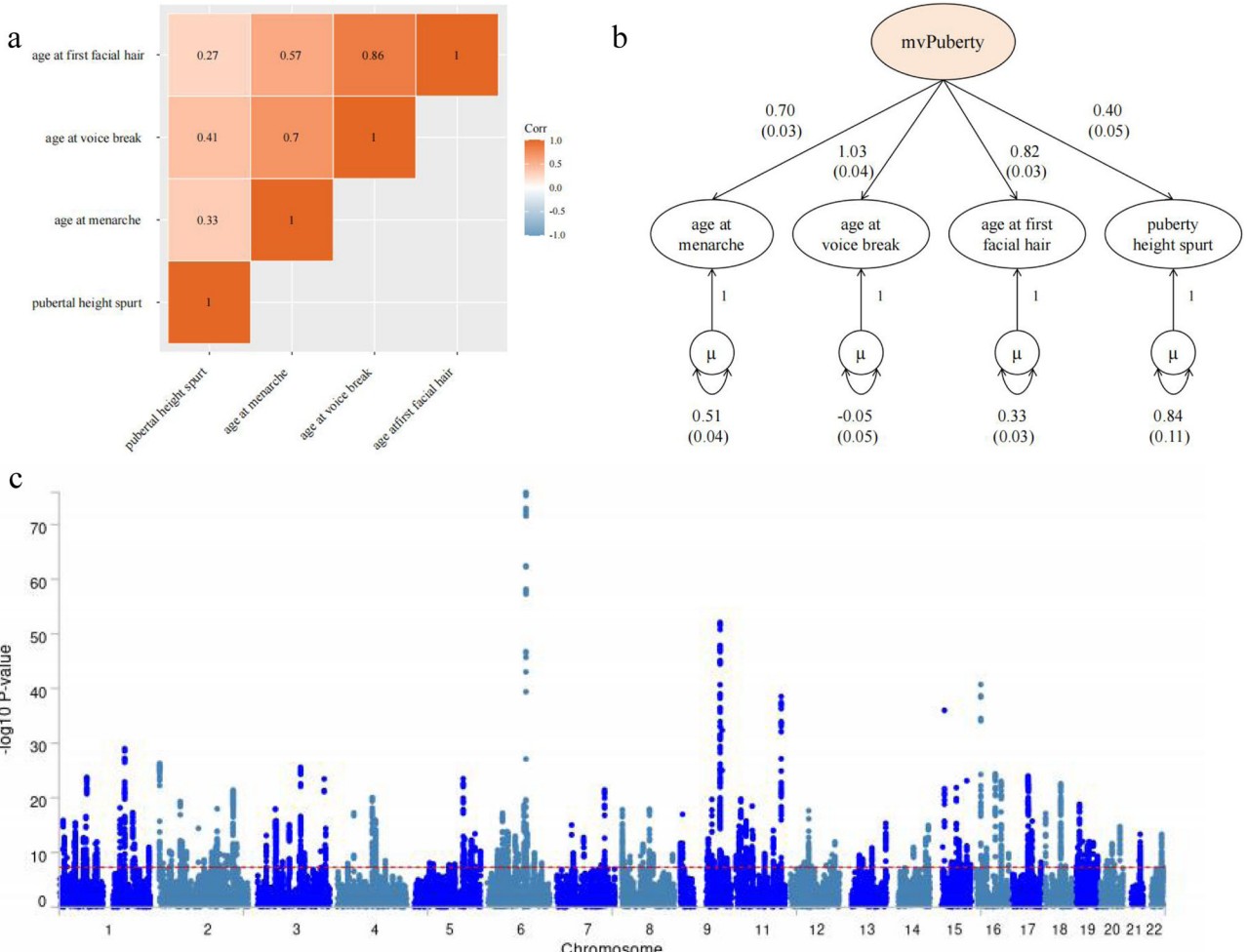

**Fig. 2 | Multivariate puberty timing GWAS modelled with genomic SEM.**
**a** Genetic correlations for SEM with genomic SEM, displaying pairwise LD score genetic correlation estimates for the five univariate phenotypes. Corr, genetic correlation. **b** Path diagram of the common factor model estimated with genomic SEM, with standardised factor loadings (standard error in parentheses). μ reflects the residual variance in the genetic indicators for the input univariate puberty timing related GWASs not explained by the mvPuberty common factor.

**c** Manhattan plot showing SNP associations (-log10($P$ value)) with mvPuberty, ordered by chromosome. The red dashed line indicates the threshold for conventional genome-wide significance ($P = 5 \times 10^{-8}$). The blue dots represent variants for mvPuberty. $P$ values are derived from two-sided Wald tests for each SNP on mvPuberty. mvPuberty, the common factor model for puberty. The mvPuberty GWAS data is available on Zenodo (https://doi.org/10.5281/zenodo.14183879).

## $Q_{SNP}$ heterogeneity

We evaluated whether the SNP associations are appropriately modelled through a multivariate framework using $Q_{SNP}$ heterogeneity statistics, in which significant $Q_{SNP}$ tests in mvPuberty would suggest that SNPs impact single-phenotype GWASs via pathways other than the shared genetics of pubertal timing modelled by mvPuberty. We found that 27 of the 266 lead SNPs surpassed the Bonferroni-corrected threshold, $P < 1.88 \times 10^{-4}$ (0.05/266), whereas none of the 18 newly reported loci had a $Q_{SNP}$ $P < 1.88 \times 10^{-4}$ (Supplementary Data 6). The results indicated that the 18 newly reported loci impact four input puberty phenotypes via mvPuberty. The sources of the 27 lead SNPs with heterogeneity are shown in Supplementary Data 11.

## Significant Cross-trait OUtliers and Trends in JOint York regression (SCOUTJOY)

For the 266 mvPuberty-associated loci, we compared the genetic effect on Tanner stage in males versus females and revealed differences at four loci: rs10217141, rs10221489, rs1036526 and rs10470054 (Supplementary Data 12). We conducted SCOUTJOY regression and reported that the magnitude of the genetic effect was concordant (slope=0.74, se=0.12) between the sexes, except four loci had greater

effects in males (Supplementary Fig. 21; Supplementary Data 13), suggesting that the relationship of mvPuberty loci with the Tanner stage is not largely mediated by sex in the context of pubertal timing and that the observed structure is generally consistent across sexes.

## Transcriptomic imputation

TWAS leveraging functional summary-based imputation (FUSION) combines gene expression data with summary statistics of GWAS association to pinpoint genes whose cis-regulatory expression correlates with complex traits. TWAS leveraging unified test for molecular signatures (UTMOST) consolidates multiple single-tissue associations into a robust, unified metric, enabling the quantification of overall gene-trait associations at the organismal level. We performed TWAS using FUSION and UTMOST to identify gene-level associations with the mvPuberty genetic signature. We found that 121 genes surpassed the Bonferroni-corrected threshold for multiple comparisons (Fig. 3a; Supplementary Data 14). We conducted further analysis via fine-mapping of causal gene sets (FOCUS). Five genes represent potentially causal signals associated with mvPuberty, including *KDM4C*, *LEPR*, *SMARCAD1*, *CCNC* and *ACP1*. The TWAS Z scores for *KDM4C*, *LEPR*, *SMARCAD1* and *CCNC* are > 0, indicating a positive association with

**Table 1 | Novel loci associated with mvPuberty**

| SNP | CHR | BP | A1 | A2 | MAF | P | Beta | SE | Nearest gene | Q$_{pval}$ |
|---|---|---|---|---|---|---|---|---|---|---|
| rs7587651 | 2 | 10368606 | C | T | 0.3638 | $1.00 \times 10^{-8}$ | 0.012 | 0.002 | C2orf48 | 0.93 |
| rs2674036 | 2 | 59830697 | T | C | 0.2286 | $3.13 \times 10^{-8}$ | −0.013 | 0.002 | RP11-444A22.1; AC007131.2 | 0.97 |
| rs4076654 | 2 | 121155824 | A | T | 0.4026 | $2.32 \times 10^{-9}$ | −0.013 | 0.002 | INHBB | 0.24 |
| rs17400325 | 2 | 178565913 | T | C | 0.0676 | $9.42 \times 10^{-9}$ | 0.025 | 0.004 | PDE11A; AC012499.1 | 0.74 |
| rs2137182 | 3 | 68594889 | C | T | 0.3042 | $5.26 \times 10^{-9}$ | −0.013 | 0.002 | FAM19A1 | 0.88 |
| rs3922468 | 5 | 95613007 | G | A | 0.3449 | $1.77 \times 10^{-8}$ | 0.012 | 0.002 | CTD-2337A12.1; RP11-254I22.2 | 0.07 |
| rs12173082 | 5 | 132389687 | C | G | 0.2316 | $6.36 \times 10^{-9}$ | −0.014 | 0.002 | HSPA4 | 0.41 |
| rs11754643 | 6 | 84313180 | C | T | 0.2286 | $1.96 \times 10^{-10}$ | −0.015 | 0.002 | SNAP91 | 0.52 |
| rs11556924 | 7 | 129663496 | C | T | 0.3767 | $2.24 \times 10^{-9}$ | −0.012 | 0.002 | RP11-306G20.1; ZC3HC1 | 0.93 |
| rs12056794 | 8 | 9553915 | A | G | 0.0686 | $3.66 \times 10^{-9}$ | 0.023 | 0.004 | TNKS | 0.65 |
| rs714417 | 11 | 45247176 | T | C | 0.3280 | $5.61 \times 10^{-9}$ | 0.013 | 0.002 | PRDM11; CTD-2560E9.3 | 0.27 |
| rs4394914 | 12 | 33837788 | A | G | 0.4732 | $3.17 \times 10^{-8}$ | 0.011 | 0.002 | RP13-359K18.1 | 0.52 |
| rs7309053 | 12 | 38444776 | G | T | 0.3847 | $6.47 \times 10^{-9}$ | −0.012 | 0.002 | RP11-297L6.2 | 0.17 |
| rs10876864 | 12 | 56401085 | G | A | 0.4085 | $4.81 \times 10^{-8}$ | 0.011 | 0.002 | IKZF4 | 0.87 |
| rs3894415 | 12 | 57376840 | C | T | 0.1282 | $3.20 \times 10^{-8}$ | −0.018 | 0.003 | GPR182 | 0.49 |
| rs5742915 | 15 | 74336633 | T | C | 0.4384 | $1.45 \times 10^{-8}$ | −0.011 | 0.002 | PML | 0.19 |
| rs12456065 | 18 | 57674133 | T | A | 0.2247 | $2.93 \times 10^{-8}$ | 0.014 | 0.002 | SDCCAG3P1 | 0.07 |
| rs8110135 | 19 | 58998464 | G | A | 0.2594 | $4.97 \times 10^{-10}$ | −0.015 | 0.002 | SLC27A5 | 0.24 |

*Note*: Lead SNPs were defined as novel if they were > 1 Mb from previously identified loci in the univariate pubertal timing related GWASs comprising the mvPuberty. Q$_{SNP}$ heterogeneity statistics, Q statistic of $\chi^2$ distributed, evaluated whether the multivariate SNP associations are appropriately modelled through a multivariate framework (Q$_{pval}$ sourced two-sided test and passed Bonferroni corrected). Because the null hypothesis of the Q$_{SNP}$ test is that the SNP associations on the univariate GWASs are statistically mediated by the resultant multivariate GWAS, significant Q$_{SNP}$ tests in the multivariate GWAS summary statistics suggest that the SNP impacts the univariate GWASs by pathways other than mvPuberty (see "Methods" and Supplementary Methods for additional information). *CHR* chromosome, *BP* position, *A1* effect allele, *A2* Non effect allele, *MAF* Minor Allele Frequency.

mvPuberty. The TWAS Z score for *ACP1* is < 0, suggesting that down-regulation of *ACP1* is associated with mvPuberty.

### Pathway, cell type and tissue enrichment

We performed multi-marker analysis of genomic annotation (MAGMA) gene-based mapping and identified 456 genes, with the majority correlated with the transcriptional regulation of DNA binding (Supplementary Figs. 22–25; Supplementary Data 15–16). We conducted cell type enrichment and revealed that six cell types surpassed the Bonferroni-corrected threshold for multiple comparisons (Supplementary Fig. 26; Supplementary Data 17). The top two cell types were nonmyeloid neurons in the brain and pancreatic A cells. We found that mvPuberty is enriched primarily in pancreatic cells, including pancreatic A, B, D, and PP cells ($P < 0.05$). We performed tissue expression enrichment and identified seven tissue types that surpassed the Bonferroni-corrected threshold for multiple comparisons (Supplementary Fig. 27), with the strongest enrichment in the cerebellar hemisphere, cerebral cortex, hypothalamus, pituitary, and ovary.

### Proteome imputation

We conducted PWAS using FUSION to identify proteome-level associations with the mvPuberty genetic signature. We detected 34 plasma proteins that surpassed the Bonferroni-corrected threshold for multiple comparisons via two plasma proteome reference panels and identified seven plasma proteins in both reference panels (Fig. 3b; Supplementary Data 18). Through further fine-mapping, we revealed that six plasma proteins, namely, *ACP1*, *DLK1*, *HPGDS*, *PCSK1*, *RMDN1* and *TIE1*, represent potentially causal signals associated with mvPuberty. The PWAS Z scores for *PCSK1*, *RMDN1* and *TIE1* were > 0, indicating a positive association with mvPuberty. The PWAS Z scores for *ACP1*, *DLK1* and *HPGDS* were < 0, suggesting that the downregulation of *ACP1*, *DLK1* and *HPGDS* is associated with mvPuberty.

### Genetic correlations with adult traits

To explore the impact of pubertal timing on complex traits later in life, we used 78 traits as outcomes in the LD score regression. We found that genetically predicted ageing-related phenotypes, organ volume, and the presence of cardiovascular diseases, metabolic syndrome, osteoporosis, gastrointestinal tract disorders, and psychiatric diseases are genetically associated with mvPuberty status (Supplementary Data 19).

### Brain image imputation

To show mvPuberty-related brain features, we applied the full BrainXcan pipeline to mvPuberty. Among the 261 brain features, 37 were significantly correlated with mvPuberty status, surpassing Bonferroni the Bonferroni-corrected threshold of $P < 1.92 \times 10^{-4}$ (0.05/261; Supplementary Data 20): mvPuberty-associated structural features, including the total volume of grey matter, the volume of grey matter in the left and right X cerebellum, the volume of grey matter in the right insular cortex, the volume of grey matter in the right temporal occipital fusiform cortex, the volume of grey matter in the right paracingulate gyrus, and the volume of grey matter in the left temporal occipital fusiform cortex; mvPuberty-associated diffusion magnetic resonance imaging (dMRI) features, including the principal components of the intracellular volume fraction in the inferior cerebellar peduncle, the external capsule, the superior corona radiata, the superior longitudinal fasciculus, and the cingulate gyrus; and the principal components of fractional anisotropy and the orientation dispersion index (Supplementary Figs. 28–32). Among the top 20 features associated with mvPuberty status, four causal flow brain features with nominal significance ($P < 0.05$) were identified: the mean orientation dispersion index in the uncinate fasciculus (right) on the fractional anisotropy skeleton (dMRI), the mean intracellular volume fraction in the medial lemniscus (left) on the fractional anisotropy skeleton (dMRI), the mean fractional anisotropy in the uncinate

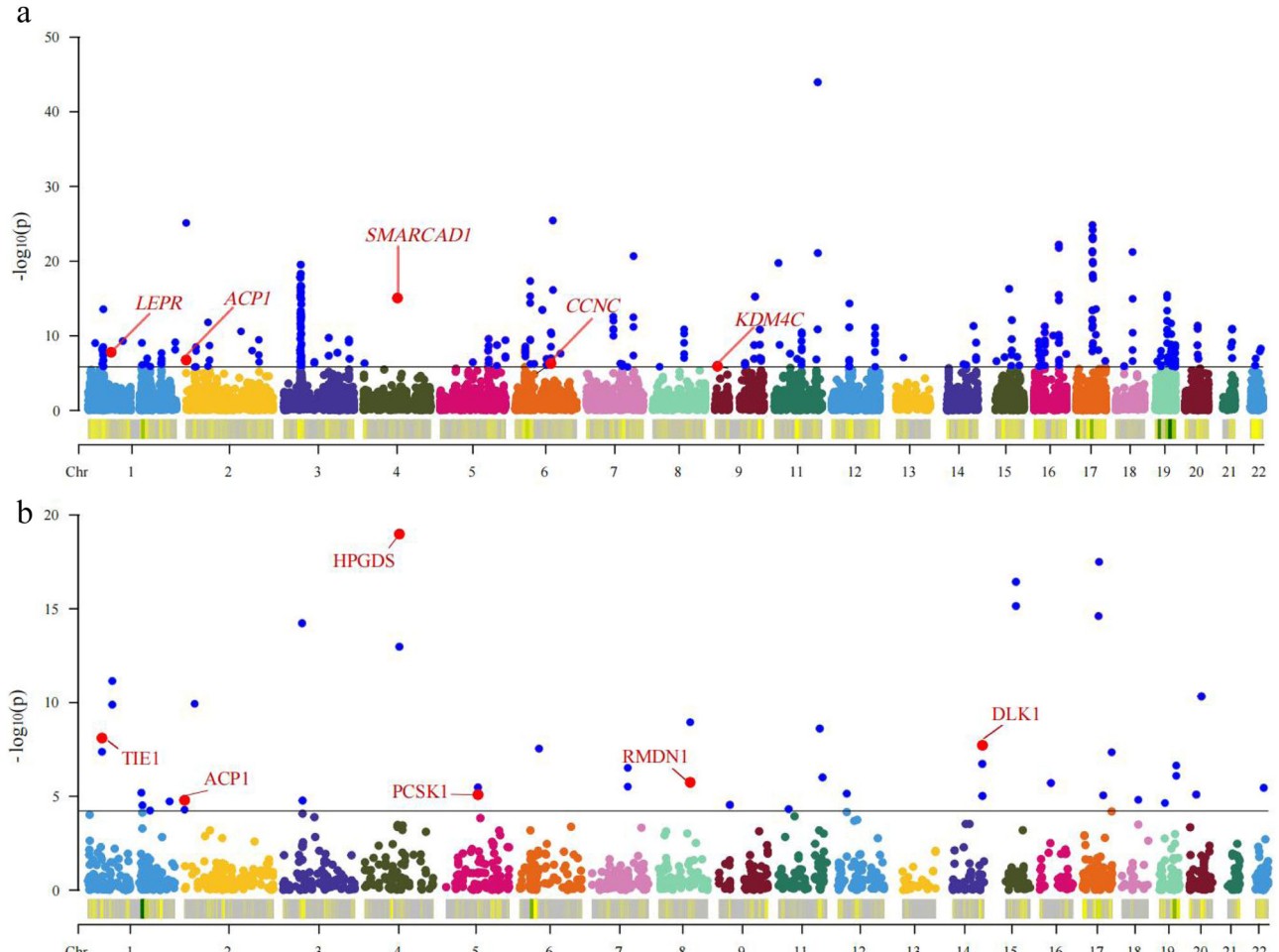

**Fig. 3 | Gene and protein level Manhattan plot of transcriptome and proteome wide association study. a** TWAS -log10(*P* value) (effect estimates of the imputed gene expression on mvPuberty) are plotted. The expression quantitative trait loci (eQTL) data are derived from the GTEx Version 8 sparse canonical correlation analysis. **b** PWAS -log10(*P* value) (effect estimates of the imputed protein expression on mvPuberty) are plotted. The cis-protein quantitative trait loci (pQTL) data are derived from the INTERVAL and Atherosclerosis Risk in Communities study.

The black dashed line indicates the threshold for genes or proteins surpassing the Bonferroni adjusted threshold. Red points and labels indicate genes or proteins surpassing the Bonferroni adjusted threshold and Focus analysis. The blue dots represent only genes or proteins surpassing the Bonferroni adjusted threshold. Other coloured dots represent genes or proteins on different chromosomes. *P* values are derived from two-sided TWAS tests for each gene or protein on mvPuberty. Chr: chromosome. Source data are provided as a Source Data file.

fasciculus (right) on the fractional anisotropy skeleton (dMRI), and the volume of grey matter in the left X cerebellum (Supplementary Fig. 33; Supplementary Data 21).

**Mendelian randomisation**

In the step-1 MR analysis for mvPuberty on 78 phenotypes, we identified ten traits, including osteoporosis, coronary heart diseases, ageing, and metabolic syndrome, whose causal estimates surpassed the Bonferroni corrected threshold ($P < 6.49 \times 10^{-4}$; 0.05/78; Supplementary Fig. 34; Supplementary Data 22).

In the step-2 MR analysis for mvPuberty on 34 biomarkers, we found that glycated haemoglobin (HbA1c), high-density lipoprotein cholesterol (HDL-C), apolipoprotein A, and alanine aminotransferase were influenced by mvPuberty, with a Bonferroni-corrected threshold of $P < 1.47 \times 10^{-3}$ (0.05/34; Supplementary Fig. 34; Supplementary Data 23). In the step-2 MR analysis for mvPuberty status on the abundances of 207 gut microbial taxa, 13 microbial taxa were potentially related to mvPuberty on the basis of the inverse-variance weighting (IVW) method, with nominal significance ($P < 0.05$) (Supplementary Fig. 35; Supplementary Data 24). The gut microbiota most strongly associated with mvPuberty status was *Parabacteroides*, with a β value of −0.42 ($P = 0.0007$).

We utilised Bayesian model averaging MR (MR−BMA) analysis to explore the mediating effects of HbA1c, HDL-C, apolipoprotein A, alanine aminotransferase, and *Parabacteroides* on the associations between mvPuberty status and the presence of several adult phenotypes (Supplementary Fig. 36). Our study revealed causal associations between HbA1c, HDL-C, apolipoprotein A, and alanine aminotransferase levels with the progression of coronary heart disease and ageing. Additionally, we found HDL-C levels are causally linked with the onset of heart failure, and the abundance of *Parabacteroides* is causally related to lumbar spine bone mineral density (BMD) and total body BMD (Supplementary Data 25−27). We revealed strong genetic correlations among HbA1c, HDL-C, apolipoprotein A, and alanine aminotransferase levels in the LD score regression (Supplementary Fig. 37). Thus, we included HbA1c, HDL-C, apolipoprotein A, and alanine aminotransferase levels in MR−BMA to analyse the mediating effects of mvPuberty on coronary heart disease and ageing. We ranked the top models by posterior probability and biomarkers by marginal inclusion probability (MIP) and provided the model-averaged causal effect (MACE) for each biomarker (Table 2). For coronary artery disease, the top-ranked model included alanine aminotransferase and HDL-C levels (posterior probability=0.916), which are biomarkers with strong overall evidence (alanine aminotransferase: MIP = 0.999, MACE =

0.200; HDL-C: MIP = 0.992, MACE = −0.265). For ageing, the top-ranked model included alanine aminotransferase, HbA1c, and HDL-C levels (posterior probability=0.943), which had strong effects on the outcome (HbA1c: MIP = 1.0, MACE = −0.074; alanine aminotransferase: MIP = 0.999, MACE = −0.028; HDL-C: MIP = 0.947, MACE = 0.028).

On the basis of the above analyses, we found that early pubertal timing causally increased alanine aminotransferase and HbA1c levels, decreases HDL-C levels, and is associated with greater ageing, with mediated proportions of 5.25%, 5.25%, and 9%, respectively (Fig. 4a; Supplementary Data 22-24); early pubertal timing causally increased alanine aminotransferase levels, decreased HDL-C levels, and was associated with a greater risk of developing coronary heart disease, with a mediated proportion of 5.12% and 10.73%, respectively (Fig. 4b; Supplementary Data 22, 23 and 25); early pubertal timing causally decreased HDL-C levels and was associated with a greater risk of developing heart failure, with a mediated proportion of 3.75% (Fig. 4b; Supplementary Data 22, 23 and 25); and the abundance of *Parabacteroides* mediated the causal effect of pubertal timing on lumbar spine osteoporosis and total body BMD, with mediated proportions of 14.18% and 13.26%, respectively (Fig. 4c; Supplementary Data 22, 24 and 26).

### Modifiable dietary factors

We used MR to assess possible causal relationships between genetically predicted dietary intake and the levels of various micronutrients (Supplementary Data 28–29). We identified several nominally significant ($P < 0.05$) causal dietary factors associated with mvPuberty status, with oily fish intake ($\beta = -0.11$, $P = 6.55 \times 10^{-3}$) and the serum retinol level ($\beta = -0.18$, $P = 4.49 \times 10^{-4}$) surpassing the Bonferroni-corrected threshold for multiple comparisons (Table 3).

## Discussion

Here, we reported a multivariate GWAS with an effective sample size of 514,750 European participants and identified 266 independent genome-wide loci for pubertal timing, which are highly consistent with the Tanner stage across sexes. Common loci explained 9.43% of the variation in pubertal timing in our multivariate analysis, whereas they explained 6.67–20.29% of the variation in previous univariate analyses[5,7,14]. Eighteen loci were not previously identified in the input GWASs, highlighting the robust statistical power of our multivariate GWAS.

We used fine mapping to prioritise strongly associated variants. The causal SNP rs11556924 located on chr7q32.2 is associated with several traits, including sex hormone-binding globulin levels[15], coronary artery disease[16], systolic blood pressure[17], and diastolic blood pressure[18]. The gene closest to rs11556924 is *ZC3HC1*, which can regulate mitotic entry time and may promote carcinogenesis[19,20], was not previously associated with pubertal timing. *ZC3HC1* is closely related to the risk of developing essential hypertension through endothelial dysfunction[21]. In addition, maintaining endothelial function is essential for the nitric oxide synthase pathway, which regulates the expression of genes associated with reproduction[21,22]. Therefore, *ZC3HC1* may regulate puberty-related reproductive maturation by affecting endothelial function. Another plausible causal SNP, rs5742915, located on chr15q14.1, is associated with lung function and body development[23,24]. The gene closest to rs5742915 is *PML*, also known as *TRIM19*. *TRIM19* is the core component of promyelocytic leukaemia nuclear bodies, which are tightly associated with the nuclear matrix. Promyelocytic leukaemia nuclear bodies, which interact with different proteins, play many instrumental roles in DNA damage responses, apoptosis, cellular senescence, and angiogenesis[25]. Previous studies have shown that the knockdown of *PML* inhibits the proliferation of oestrogen receptor-positive breast cancer cells and promotes the expression of oestrogen receptor target genes, thereby increasing oestrogen receptor-positive breast cancer cell stemness[26]. We found that *PML* also plays an important role in the timing of puberty, which is related to the regulation of oestrogen receptor target genes. The gene closest to rs913588 is *KDM4C*, which is linked to the regulation of cell proliferation, differentiation, and maintenance in various stem cell types[27]. *KDM4C* was associated with mvPuberty status in our TWAS signal test. Other credible causal genes found by the TWAS include leptin receptor (*LEPR*), SWI/SNF-related matrix-associated actin-dependent regulator of chromatin subfamily A containing DEAD/H box 1 (*SMARCAD1*), cyclin C (*CCNC*), and acid phosphatase 1 (*ACP1*). Leptin binds to *LEPR* in hypothalamic neurons[28], and *LEPR* activation is crucial to energy balance and body weight homoeostasis[28,29]. Leptin can modulate reproductive function such that its absence results in hypothalamic hypogonadism[30], and leptin is a critical mediator influencing puberty onset in females[31]. Our observations, along with these findings, support the vital role of *LEPR* in the timing of puberty. *SMARCAD1*, a UUU codon-enriched protein and key factor in controlling endogenous retroviruses, is related to neuron differentiation, cell proliferation, migration, and invasion[32]. *CCNC*, a direct target of the nuclear hormone all-trans retinoic acid, controls cell proliferation and gene transcription in the mammalian testis[33]. *ACP1*, a phosphoprotein tyrosine phosphatase, exhibits the capability to dephosphorylate platelet-derived growth factor receptors, thereby modulating their activity as a growth factor, and plays a pivotal role in the regulation of male gonad development during embryonic and postnatal stages[34]. Similarly, we found that the *ACP1* translated protein is associated with mvPuberty via proteome-based tests and proteome-based fine mapping.

Gene set enrichment analysis revealed significant enrichment of mvPuberty SNP heritability in the regulation of DNA-binding transcription. Previous studies have shown that many DNA-binding transcription factors, such as *PCSK1*, *LEP*, *TAC3*, and *VAX1*, influence gonadotropin-releasing hormone release and target the hypothalamic–pituitary–gonadal axis, which is closely associated with pubertal timing[35]. Interestingly, our TWAS and PWAS analyses identified *LEP* and *TAC3* receptors and *PCSK1*. Tissue enrichment analysis revealed significant enrichment of mvPuberty SNP heritability in the cerebellar hemisphere, cerebral cortex, hypothalamus, pituitary, and ovary, suggesting that mvPuberty-mapped genes target the hypothalamic–pituitary–gonadal axis. These results highlight heritability for pubertal timing enriched in the hypothalamic–pituitary–gonadal axis, which influences hormone release to regulate body functions[36]. In addition, previous studies have shown that the cerebellar hemisphere and cerebral cortex, identified in our tissue enrichment analysis for mvPuberty, regulate muscle tone and coordinate movement, cognitive abilities and social behaviours through connections between neurons[37]. Moreover, connections between neurons in the cerebellar hemisphere and cerebral cortex are regulated by the phagocytosis of microglia during brain development[38]. Microglia are responsive to oestrogens and prostaglandins, which regulate each other's synthesis in a cerebellar hemisphere- and cerebral cortex-dependent manner, thereby modulating neurogenesis and neuronal survival[38,39]. Therefore, adolescents with early puberty status are at significantly greater risk of poor mobility and cognitive performance than those with typical pubertal timing are[40], which may be attributed to abnormal changes in oestrogens and prostaglandins, leading to abnormal function of microglia in brain development. Cell type analysis demonstrated significant enrichment of mvPuberty SNP heritability in pancreatic cells, which are involved in blood glucose regulation through insulin and glucagon secretion. Insulin resistance is a physiological phenomenon observed during puberty, where insulin sensitivity experiences a decline at the onset of puberty, reaches its lowest point around mid-puberty, and subsequently returns to prepubertal levels upon completion of puberty[41]. These previous findings support the links between various pancreatic cells and the timing of puberty[42,43]. For example, human growth hormone levels also peak during puberty and are associated with pubertal timing via positive

**Table 2 | Prioritisation of causal biomarkers of coronary artery disease and mvAge using the MR-BMA method**

| Outcome | Biomarkers | MIP | MACE | Empirical P value | Bonferroni-adjusted P value | Models | Posterior probability |
|---|---|---|---|---|---|---|---|
| Coronary artery disease | Alanine aminotransferase | 0.999 | 0.200 | $1.00 \times 10^{-4}$ | $4.00 \times 10^{-4}$ | Alanine aminotransferase,HDL cholesterol | 0.916 |
| | HDL cholesterol | 0.992 | −0.265 | $2.00 \times 10^{-4}$ | $4.00 \times 10^{-4}$ | Alanine aminotransferase,Glycated haemoglobin,HDL cholesterol | 0.054 |
| | Glycated haemoglobin | 0.056 | 0.009 | $9.83 \times 10^{-1}$ | $9.85 \times 10^{-1}$ | Alanine aminotransferase,Apolipoprotein A,HDL cholesterol | 0.020 |
| | Apolipoprotein A | 0.030 | 0.001 | $9.85 \times 10^{-1}$ | $9.85 \times 10^{-1}$ | Alanine aminotransferase,Apolipoprotein A | 0.007 |
| mvAge | Glycated haemoglobin | 0.999 | −0.074 | $1.00 \times 10^{-4}$ | $1.33 \times 10^{-4}$ | Alanine aminotransferase,Glycated haemoglobin,HDL cholesterol | 0.943 |
| | Alanine aminotransferase | 0.999 | −0.028 | $1.00 \times 10^{-4}$ | $1.33 \times 10^{-4}$ | Alanine aminotransferase,Glycated haemoglobin,Apolipoprotein A | 0.051 |
| | HDL cholesterol | 0.947 | 0.028 | $1.00 \times 10^{-4}$ | $1.33 \times 10^{-4}$ | Alanine aminotransferase,Glycated haemoglobin,Apolipoprotein A,HDL cholesterol | 0.002 |
| | Apolipoprotein A | 0.053 | 0.001 | $9.56 \times 10^{-1}$ | $9.56 \times 10^{-1}$ | Alanine aminotransferase,Glycated haemoglobin | 0.002 |

*Note:* We incorporated biomarkers that were found to be nominally significantly associated with coronary artery disease and mvAge in the univariate Mendelian randomisation analysis into the Mendelian randomisation analysis based on Bayesian model averaging (MR-BMA) analysis. Biomarkers ranked by the marginal inclusion probability in the MR-BMA analysis after model diagnostics. Empirical P-values were computed using 10 000 permutations and passed Bonferroni-adjusted. The prior probability and prior variance used in the MR-BMA analysis were 0.1 and 0.5. *MIP* marginal inclusion probability. *MACE* model-averaged causal effect. *HDL* High-Density Lipoprotein. *mvAge,* multivariate GWAS of age.

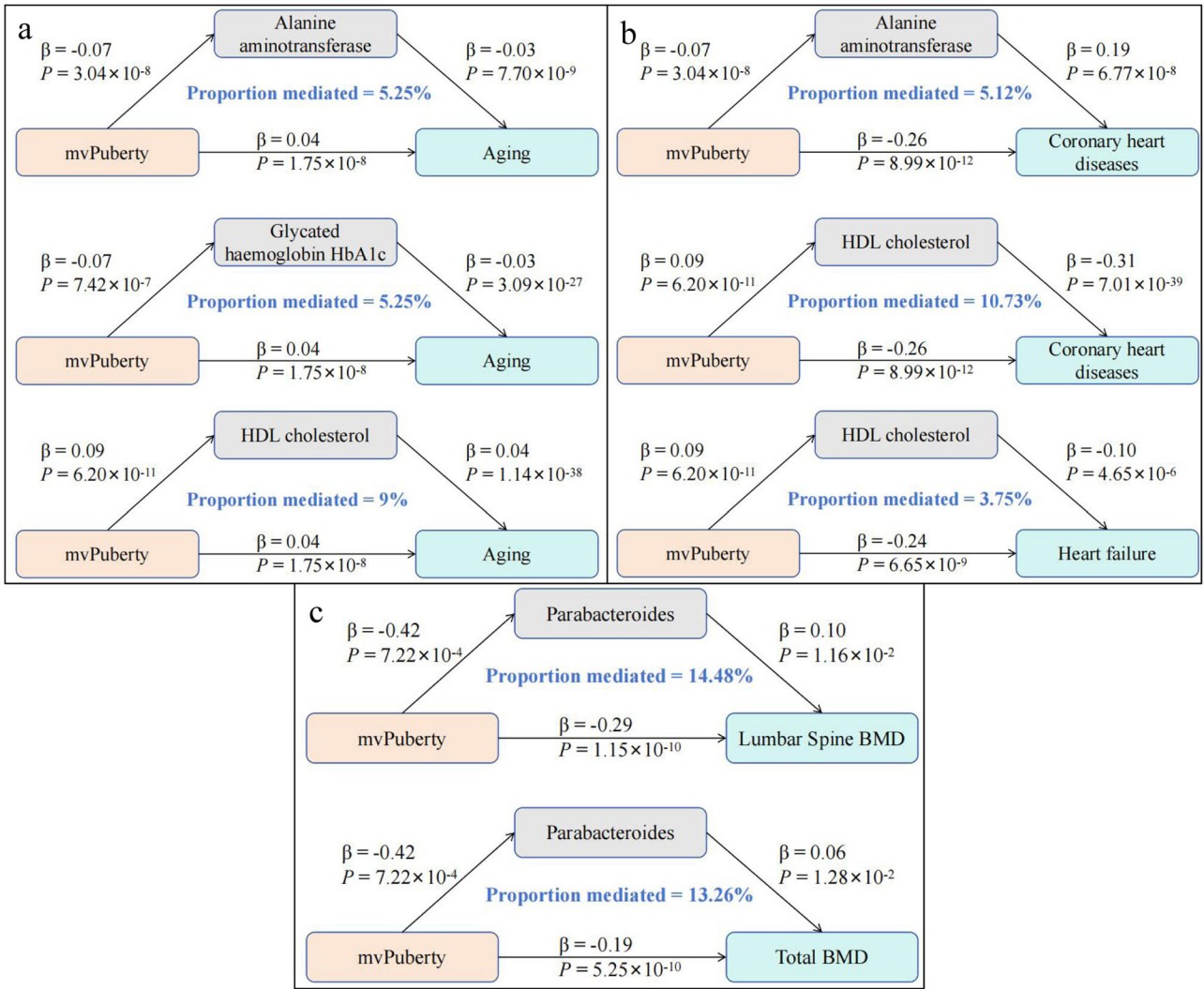

**Fig. 4 | The biomarkers mediated the causal effect of mvPuberty on adulthood traits. a** The biomarkers mediated the causal effect of mvPuberty on aging. **b** The biomarkers mediated the causal effect of mvPuberty on cardiovascular disease. **c** The biomarkers mediated the causal effect of mvPuberty on osteoporosis. The β value are the MR estimates using the inverse–variance weighted method. *P* values are derived from Mendelian randomisation tests using inverse variance weighted method, and are surpassing the Bonferroni adjusted threshold. BMD, Bone mineral density. HDL, High-Density Lipoprotein. mvPuberty, the common factor model for puberty.

effects on maintaining the pancreatic B-cell mass[44]. In summary, our enrichment analysis further confirmed that pancreatic cells play an important role in the onset of puberty by secreting insulin and glucagon to meet the needs of blood sugar changes.

Notably, advanced puberty exerts enduring effects on brain maturation, which are manifested in the enhancement of neural connections and the organisation and promotion of cognitive abilities and social behaviours, partly attributed to alterations in levels of pubertal hormones[45–47]. We highlight relationships of brain region development identified in our BrainXcan analysis and pubertal course in the Supplementary Discussion.

Our analyses of the shared genetic risk between mvPuberty status and adult traits provide further insights into the health implications in later life. In line with previous observations[2,48], we confirmed significant genetic correlations of mvPuberty status with ageing and the risk of developing cardiovascular diseases, osteoporosis, endometrial cancer, and metabolic syndrome. We observed the strongest genetic correlations for metabolic syndrome, indicating that early pubertal timing may influence blood pressure and lipid and glucose metabolism in adulthood. Our results support the hypothesis that early pubertal timing is one of the predisposing factors for metabolic syndrome[49].

Features of metabolic syndrome, such as low HDL-C levels, high fasting triglyceride levels, high blood pressure, and elevated fasting plasma glucose concentrations, are associated with many adverse health consequences, including the development of cardiovascular diseases and accelerated ageing[50–52]. Therefore, we analysed the mediating role of these metabolic biomarkers in the associations between early pubertal timing and adverse health outcomes in adults. Extended elaborations of mediating role for metabolic biomarkers can be found in the Supplementary Discussion.

Recent studies have indicated a correlation between early pubertal timing and gut microbiota, and disruption of the gut flora is associated with abnormal sexual maturation[53,54]. Moreover, previous studies emphasised the significant role of gut microbiota in bone metabolism, particularly in the progression of osteoporosis. Oestrogen depletion has been identified as a main contributor to alterations in bone microarchitecture, a key characteristic of osteoporosis[55]. Our analysis revealed that *Parabacteroides* mediated the causal relationship between mvPuberty status and the risk of developing osteoporosis. *Parabacteroides* belongs to the phylum *Bacteroidetes*, which includes many starch-fermenting bacteria[56] that promote calcium absorption to prevent osteoporosis by fermenting nondigestible

**Table 3 | Mendelian randomisation analysis investigating the causal role of dietary factors on mvPuberty**

| Type of exposure | Exposure | Methods | Nsnp | Beta | P | Q | Q_df | Q_pval | Correct causaldirection | SteigerP value |
|---|---|---|---|---|---|---|---|---|---|---|
| Dietary factors | Oily fish intake | Inverse variance weighted | 18 | −0.22 | $1.40 \times 10^{-2}$ | 98.27 | 16 | $7.30 \times 10^{-14}$ | TRUE | $1.78 \times 10^{-44}$ |
| | | MR LASSO | 20 | −0.11 | $6.55 \times 10^{-3}$ | - | - | - | - | - |
| | | MRlap | 21 | −0.29 | $5.44 \times 10^{-3}$ | - | - | - | - | - |
| | Coffee intake | Inverse variance weighted | 16 | −0.19 | $5.37 \times 10^{-2}$ | 162.88 | 14 | $1.87 \times 10^{-27}$ | TRUE | $1.02 \times 10^{-112}$ |
| | | MR LASSO | 13 | −0.06 | $4.69 \times 10^{-2}$ | - | - | - | - | - |
| | | MRlap | 18 | −0.18 | $4.23 \times 10^{-2}$ | - | - | - | - | - |
| | Tea intake | Inverse variance weighted | 10 | −0.06 | $1.41 \times 10^{-1}$ | 20.02 | 8 | $1.03 \times 10^{-2}$ | TRUE | $1.37 \times 10^{-109}$ |
| | | MR LASSO | 9 | −0.06 | $4.72 \times 10^{-2}$ | - | - | - | - | - |
| | | MRlap | 14 | −0.12 | $5.74 \times 10^{-2}$ | - | - | - | - | - |
| | Cooked vegetable intake | Inverse variance weighted | 5 | −0.72 | $3.78 \times 10^{-2}$ | 59.00 | 3 | $9.62 \times 10^{-13}$ | TRUE | $5.93 \times 10^{-4}$ |
| | | MR LASSO | 2 | −0.37 | $1.40 \times 10^{-2}$ | - | - | - | - | - |
| | | MRlap | 6 | −0.74 | $6.25 \times 10^{-2}$ | - | - | - | - | - |
| | Tea intake | Inverse variance weighted | 10 | −0.06 | $1.41 \times 10^{-1}$ | 20.02 | 8 | $1.03 \times 10^{-2}$ | TRUE | $1.37 \times 10^{-109}$ |
| | | MR LASSO | 9 | −0.06 | $4.72 \times 10^{-2}$ | - | - | - | - | - |
| | | MRlap | 14 | −0.12 | $5.74 \times 10^{-2}$ | - | - | - | - | - |
| Serum micronutrients | Retinol | Inverse variance weighted | 2 | 0.18 | $4.49 \times 10^{-4}$ | - | - | - | TRUE | $4.74 \times 10^{-7}$ |

*Note: P* were calculated for the respective method of Mendelian randomisation analysis using inverse variance weighted, MR LASSO and MRlap methods. The heterogeneity test in the inverse variance weighted methods was performed using Cochran's Q statistic (Q_pval sourced two-sided test and passed Bonferroni corrected). Q, Cochran's Q statistic; df, freedom for Cochran's Q statistic; MR LASSO, MR Least Absolute Shrinkage and Selection Operator; *Nsnp* number of single-nucleotide polymorphism.

carbohydrates[57]. *Parabacteroides* metabolises phytoestrogens and generates compounds, including secoisolariciresinol, enterolactone, and equol[58]. Thus, we speculate that early pubertal timing may affect *Parabacteroides* abundance via sex hormones, increasing osteoporosis risk.

To support public health and clinical implications, we tried to identify modifiable dietary factors and reported that oily fish intake is negatively associated with early pubertal timing, which is in agreement with the findings of a previous longitudinal study[59]. We also found that the serum retinol concentration is negatively correlated with early puberty timing. *CCNC*, a credible causal gene identified by our TWAS, is a target for all-trans retinoic acid to interact with steroid receptors[60]. Accordingly, oily fish intake may affect serum retinol and the metabolite retinoic acid, which act on the cyclin C protein and interact with steroid receptors, coactivating and inhibiting oestrogen receptor transcriptional activity[60,61] and ultimately influencing puberty onset. These results suggest that controlling oily fish intake may be beneficial for preventing early puberty.

There are several limitations in our study. First, genomic SEM offers a composite phenotype that represents the shared genetic architecture of broad liability across multiple complex traits. Consequently, multivariate GWASs, such as those for mvPuberty, lack conventional units, complicating the interpretation of our MR analyses. Given that MR results are lifelong estimates, these numerical values may pose challenges in clinical interpretation. Additionally, genomic SEM relies on an additive model for genetic variants, which may fail to capture the effects of recessive variants that influence pubertal timing adversely. Second, our mvPuberty status and subsequent analyses are restricted to individuals of European ancestry. Future research should include trans-ancestral multivariate GWASs with larger sample sizes to investigate pubertal timing across diverse populations. Third, the input univariate GWAS data may have been a limitation of this study. The growth spurt GWAS included 14,040 participants, which may not

fully capture pubertal growth features. While "sample size" is not an input to SCOUTJOY analysis, the low sample size of Tanner stage GWAS is reflected in the standard error, therefore, our results of SCOUTJOY analysis should be interpreted with caution. In addition, food intake data for MR and expression measurements for TWASs and PWASs, derived from adults or individuals with disease status, may not be ideal. Moreover, cell data for cell type enrichment analysis, assessed in mice, may not be comparable to those of humans.

In conclusion, we leveraged a multivariate GWAS approach to elucidate the genetic basis of the broad propensity for pubertal timing. We characterised putatively causal loci, genes, proteins, cell types, tissue types and enrichment pathways for pubertal timing. We reported genetic associations of early pubertal timing with accelerated ageing and the risk of developing cardiovascular disease and osteoporosis, with mediators including alanine aminotransferase, HbA1c, HDL-C and *Parabacteroides* levels. We found that controlling oily fish and retinol intake is beneficial for promoting healthy pubertal development.

## Methods

### Data sources and ethics

We obtained four univariate input GWAS datasets from participants of European ancestry, encompassing age at menarche[5], age at voice break[14], age at first facial hair[14], and the take-off phase of the growth spurt[8]. All input GWASs passed the Genomic SEM criteria, were estimated with LD score regression ($h^2 > 0.05$ and $\chi^2 > 1.05$), had existing ethical permissions from respective institutional review boards and included participant informed consent with rigorous quality control. The study was designed and conducted in compliance with all the relevant regulations regarding the use of human study participants and the criteria set by the Declaration of Helsinki.

The GWAS for the age at menarche ($n = 252,514$) included 40 studies from the ReproGen consortium ($n = 179,117$) and UK Biobank

($n$ = 73,397)[5]. GWASs for the age at voice break and the age at first facial hair, with 154,459 and 161,470 males, respectively, were available from the UK Biobank[14]. The GWAS for pubertal height spurt included 14,040 subjects (7161 males and 6879 females) from cohorts included in the Early Growth Genetics Consortium[8]. Detailed information is provided in the original studies and Supplementary Data 1.

## Genomic SEM

We used genomic SEM implemented in the Genomic SEM R package (v.0.0.5c) to perform multivariate GWAS analysis for age at menarche, age at voice break, age at first facial hair, and pubertal height spurt, investigating the broad genetic liability underlying these pubertal timing-related traits. Genomic SEM is a recently developed multivariate method that enables the investigation of multiple potential multivariate models of the underlying architecture of traits[10]. The technical details of genomic SEM are described in the Supplementary Methods. Genomic SEM is not biased by sample overlap, that is, UK Biobank participants in multiple input GWASs, or imbalanced sample sizes. Genomic SEM facilitates the identification of variants that influence only some but not all complex traits and therefore do not represent broad cross-trait liability[9,10].

Genomic SEM was performed in two stages. At stage 1, we prepared four univariate input GWASs and used multivariate extension of cross-trait LD score regression[62] to generate the empirical genetic covariance matrix between the four traits as inputs for the SEM common factor model (Supplementary Data 3). We restricted four univariate input summary statistics to the HapMap3 variants for estimating genetic covariance and the sampling covariance matrix in LD score regression with a MAF > 0.01 and information scores < 0.9 to remove rare SNPs. At stage 2, we tested the common factor model, minimising the hypothesised covariance matrix and the empirical covariance matrix calculated in stage 1[10], to identify a genetic signature underlying the four pubertal timing-related traits. Model fit was assessed via the standardised root mean square residual (SRMR), model $\chi^2$, the Akaike information criterion and the comparative fit index (CFI; Supplementary Data 4)[9,12,63].

In the multivariate GWAS, we removed SNPs with a MAF < 0.01 (to avoid error due to fewer samples within the genotype cluster), SNPs with effect estimates exactly equal to zero (to avoid compromising the matrix inversion necessary for genomic SEM), SNPs not matching the 1000 Genomes Phase 3 EUR reference panel, and SNPs with mismatched alleles. Applying the appropriate common factor SEM specification (SRMR > 0.95 and CFI < 0.08)[9,12,63], we incorporated the individual autosomal SNP associations into the genetic and associated sample covariance matrices to generate the multivariate GWAS (mvPuberty), which represented the shared covariance across univariate input GWASs[10].

## Q$_{SNP}$ heterogeneity

We calculated Q$_{SNP}$ heterogeneity statistics to evaluate whether mvPuberty SNP associations were appropriately modelled within a multivariate SEM framework. The null hypothesis of the Q$_{SNP}$ test is that SNP associations in single-phenotype GWASs are statistically mediated by mvPuberty[10]. Thus, significant Q$_{SNP}$ tests in mvPuberty suggest that SNPs impact single-phenotype GWASs via pathways other than the shared genetics of pubertal timing modelled by mvPuberty[9,10]. We used a Bonferroni-corrected $P$ value threshold to determine Q$_{SNP}$ heterogeneity.

## SCOUTJOY

To evaluate the influence of sex on mvPuberty-associated loci, we compared the effects of loci in the Tanner stage[7], a measure of sexual maturation, across sex via SCOUTJOY, addressing heterogeneity detection while allowing both sample overlap and estimation error in comparison GWASs[64]. We tested each of the lead SNPs from mvPuberty for differences in Tanner stage effect sizes between males and females via the values of effect sizes (beta) and standard errors (se) in the Tanner stage GWAS, including data for 3769 boys and 6147 girls, and we tested whether the overall trend in effect sizes for the lead SNPs differed when Tanner in males versus females was compared via SCOUTJOY. The outlier for lead SNPs and overall trend were used to assess whether the observed genetic effects were consistent across different sex groups[64]. Detailed information for SCOUTJOY is available in the Supplementary Methods.

## Identification of genomic loci and variants

We utilised FUMA v.1.6.0[65,66] to identify genomic loci and identified SNPs in LD ($R^2 < 0.1$) associated with mvPuberty status at the genome-wide significance level ($P < 5 \times 10^{-8}$). We defined a locus by lead SNPs within a 250 kb range and all SNPs in high LD ($R^2 > 0.6$) with at least one independent SNP. First, we extracted the summary statistics for these lead mvPuberty SNPs from the input univariate GWASs to assess the strength of associations. We then compared lead SNPs with the original univariate GWAS and defined SNPs to be new if they were > 1 Mb from SNPs identified in the univariate GWAS data. To determine if any lead SNPs in mvPuberty show evidence of pleiotropic associations, we looked up GWAS-significant associations ($P < 5 \times 10^{-8}$) via the GWAS Catalogue database in FUMA analysis[67].

We used the approximate Bayes factor (ABF), probabilistic annotation integrator (PAINOR), CAVIAR Bayes factor (CAVIARBF), FINEMAP and sum of single effects (SuSiE) in the python pipeline and easyfinemap (https://Jianhua-Wang.github.io/easyfinemap) to identify the most plausible causal variants associated with mvPuberty. We analysed 266 regions of interest by including all variants up- and downstream of the 266 lead SNPs at a 250 kb window from the mvPuberty GWAS. We performed fine-mapping for the 266 regions and refined the maximum number of outputs for each region to one SNP so that we could identify variants with the highest causal likelihood. Detailed information on the methods of fine mapping is available in the Supplementary Methods.

## Transcriptomic and proteome imputation

We used TWAS, including FUSION and UTMOST, to prioritise genes associated with mvPuberty[68,69]. The FUSION method involves three steps: (1) identify gene expression features that are cis-heritable (i.e., variants associated with gene expression within or near the genomic locus); (2) construct a linear predictor for each cis-heritable gene (i.e., a SNP-based prediction weight of the gene feature); and (3) calculate both TWAS test statistics incorporating these SNP-based prediction weights and summary-level GWAS Z scores. FUSION uses several penalised linear regression and Bayesian sparse linear mixed models (e.g., GBLUP, LASSO, elastic net, and BLSMM) and computes out-of-sample $R^2$ statistics to identify the best model via cross-validation of each gene–GWAS model. UTMOST consists of two steps: (1) a single-tissue association test for 44 tissues is run and (2) gene–trait associations in 44 tissues are combined via the joint generalised Berk–Jones (GBJ) test. UTMOST uses the GBJ test with single-tissue association Z statistics and their covariance matrix as inputs to provide powerful inference results while explicitly taking the correlation among single-tissue test statistics into account even under a sparse alternative. In our studies, weights for 37,920 precomputed expression quantitative trait loci features from genotype–tissue expression (GTEx) v.8 and precalculated covariance matrices for single-tissue/joint tests with GTEx v.8 were used for FUSION and UTMOST, respectively. We performed fine mapping for TWAS genes associated with mvPuberty, surpassing the Bonferroni corrected threshold, using FOCUS, which was designed for TWAS studies[70]. FOCUS fine maps TWAS/PWAS statistics at genomic risk regions and prioritises genes/proteins with strong evidence for causality.

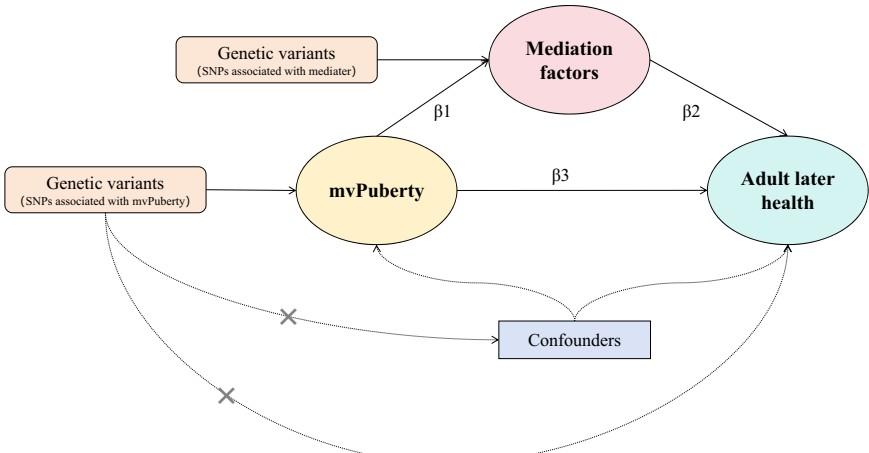

**Fig. 5 | Two-step Mendelian randomisation analysis of the effects of mvPuberty on adulthood traits via biomarkers.** β1 is the effect of mvPuberty on mediators, β2 is the effect of mediators on adult phenotypes, and β3 is the effect of mvPuberty on adult phenotypes. mvPuberty, the common factor model for puberty.

We considered TWAS-significant genes likely to be causal when the FOCUS posterior inclusion probability was > 0.5[9,71]. We prioritised mvPuberty genes identified by FUSION and UTMOST on the basis of evidence from FOCUS.

We used PWAS FUSION to explore the genetic regulation of plasma proteins in mvPuberty. PWAS weights include 2,004 precomputed cis-protein quantitative trait loci features from the Atherosclerosis Risk in Communities study and 3,154 precomputed cis-protein quantitative trait loci features from the INTERVAL study. We performed Bonferroni correction and FOCUS. Detailed information for fine mapping and transcriptomic and proteome imputation is available in the Supplementary Methods.

### Gene set, tissue expression and cell type enrichment
We used MAGMA with data from GTEx v.8 to perform gene set and tissue expression analyses for mvPuberty. To identify the cell types associated with mvPuberty, we integrated single-cell RNA sequencing (scRNA-seq) data via cell type expression-specific integration for complex traits (CELLECT)[72]. We used scRNA-seq data from *Tabula Muris*, which included transcriptomic data from 100,000 cells and 20 organs, and *Mus musculus* tissues served as a reference for adult cells and organs[73]. We prepared the *Tabula Muris* scRNA-seq data via CELLEX and calculated expression specificity likelihood scores for each gene following normalisation and preprocessing[72]. With the CELLECT default settings, we conducted cell-type enrichment with MAGMA. In CELLECT, MAGMA measures the extent to which genetic associations with a phenotype increase as a function of gene expression specificity for a given cell type. We categorised the cell types following the nomenclature used in the original *Tabula Muris* study and used a false discovery rate (FDR) threshold of 0.05.

### Genetic correlations with health outcomes
To identify the genetic effect of pubertal timing on adulthood health, we selected 78 phenotypes previously suggested to be associated with pubertal timing. We used LD score regression to estimate the genetic correlation between mvPuberty status and the risk of developing a range of traits/diseases, including 20 types of cancers, 11 types of organ volume, 11 types of psychiatric diseases, 11 types of ageing phenotypes, seven types of gastrointestinal tract disorders, six types of osteoporosis, five types of coronary heart diseases, four types of neurodegenerative diseases, two types of renal disorders, and metabolic syndrome. Detailed information is available in the Supplementary Methods.

### BrainXcan imputation
We used BrainXcan to explore the relationships between brain image-derived phenotypes and pubertal timing[74]. BrainXcan tests the associations between genetic predictors of brain image-derived features and complex traits to pinpoint relevant region-specific and cross-brain features[74]. Detailed information is available in the Supplementary Methods.

### Mendelian randomisation
All MR analyses were conducted in accordance with the STROBE-MR guidelines. We implemented MR analysis via the following R packages: MRlap (v.0.0.3), MendelianRandomization (v.0.10.0), and TwoSampleMR (v.0.6.2). We used two-step MR analysis to determine the mediating effects of biomarkers and gut microbiota on the associations between mvPuberty status and the presence of several adult phenotypes. A graphical overview of the MR analyses is shown in Fig. 5.

In step 1, we performed MR with 78 phenotypes derived from a GWAS of participants with European ancestry to investigate whether complex phenotypes, used in the LD score regression analysis, are causally affected by mvPuberty (Supplementary Data 30–31). We performed Bonferroni correction and included only significant phenotypes in the downstream analyses.

Considering the important role of biomarkers and gut microbiota in the relationship between sexual development and the presence of adult traits, in step 2, we curated 34 biomarkers (for example, lipids, blood pressure, and markers of inflammation) and 207 gut microbial taxa. We performed MR to identify biomarkers and gut microbial taxa causally influenced by mvPuberty status via Bonferroni correction.

To explore mediating effects, we included only biomarkers and gut microbial taxa that surpassed the Bonferroni-corrected threshold for multiple comparisons. We utilised MR−BMA[75], a two-sample multivariable MR approach, to identify true causal risk factors despite high correlations of candidate factors and to determine the true mediation factors between mvPuberty status and the risk of developing several important adult phenotypes. Because these mediators appear to be independent of each other, we estimated the mediation proportion separately by dividing the indirect effect by the total effect ($β1 × β2/β3$)[76], where β1 is the effect of mvPuberty on mediators, β2 is the effect of mediators on adult phenotypes, and β3 is the effect of mvPuberty on adult phenotypes.

We performed MR with 25 dietary factors and micronutrients derived from a GWAS with participants of European ancestry in the UKB datasets to investigate whether mvPuberty status is causally influenced by dietary factors and micronutrients represented by SNPs

(Supplementary Data 32). Information on the dietary factors was collected retrospectively using a dietary frequency questionnaire, which is publicly available in the UK Biobank (https://biobank.ctsu.ox.ac.uk/crystal/label.cgi?id=100052). Information on the micronutrients was collected by online 24-h dietary recall questionnaire, which is publicly available in the UK Biobank (https://biobank.ctsu.ox.ac.uk/crystal/label.cgi?id=100098). We used a Bonferroni-corrected threshold to adjust for multiple comparisons.

Given that mvPuberty status and many of the traits included in our MR analyses are derived from the UK Biobank, sample overlap may introduce bias. We thus applied the MRlap method, which accounts for sample overlap (even when the exact overlap percentage is unknown) and assesses weak instrument bias and the winner's curse, as an additional sensitivity test for all our MR analyses[77]. To avoid bias sourced from heterogeneity among the instruments, we used MR LASSO[9], which applies lasso-type penalisation to the direct effects of instruments with the post-lasso estimator, as an additional sensitivity test for our MR analyses. Detailed information on the motivation, instrumentation, MR assumption, two-step MR and MR–BMA, and sensitivity test is available in the Supplementary Methods.

### Reporting summary
Further information on research design is available in the Nature Portfolio Reporting Summary linked to this article.

## Data availability
All analyses were based on publicly available data. The mvPuberty GWAS data generated in this study are provided on Zenodo (https://doi.org/10.5281/zenodo.14183879). The TWAS and PWAS data generated in this study are provided in the Source Data file. Summary-level statistics for age of menarche are available at https://www.reprogen.org/; summary-level statistics for age of first facial hair are available at https://broad-ukb-sumstats-us-east-1.s3.amazonaws.com/round2/; summary-level statistics for age of voice break are available at https://broad-ukb-sumstats-us-east-1.s3.amazonaws.com/round2/; summary-level statistics for puberty height spurt are available at http://egg-consortium.org/Pubertal_Growth/; summary-level statistics for Tanner stage are available at http://egg-consortium.org/tanner-stage.html. GTEx weights for FUSION analyses are available at http://gusevlab.org/projects/fusion/weights/sCCA_weights_v8_2.zip. Single-cell gene expression data from the Tabula Muris study are available at https://tabula-muris.ds.czbiohub.org/. Summary-level statistics used for Mendelian randomisation are shown in Supplementary Data 30-32. Source data are provided with this paper.

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

## Acknowledgements

We appreciate the participants and investigators for making the GWAS data available, Dr. Xuejiao Song and Dr. Xunxun Deng from West China School of Public health. This work was supported by the National Natural Science Foundation of China (82173512 to Guo Cheng; 82304135 to Yujie Xu) and the Department of Science and Technology of Sichuan Province (25NSFSC2411 to Jingyuan Xiong).

## Author contributions

Z.S.: data collection, model training, analyses and draughting the manuscript. X.Y.: validation, funding and reviewing the manuscript. X.J.: idea conceiving, supervising, funding, reviewing and revising the manuscript. C.G.: idea conceiving, funding, reviewing and revising the manuscript. All authors critically reviewed the manuscript for important intellectual content.

## Competing interests

The authors declare no competing interests.
