## [Peer review File · Nature Communications]

REVIEWER COMMENTS

Reviewer #1 (Remarks to the Author):

This study models a common genetic component across various measures of pubertal timing (mvPuberty) to determine common genetic loci associated with the pubertal initiation program across both sexes. This is an interesting paper with some novel results; the study reports a handful of novel loci that have not previously been reported and provides additional insight into the biomarkers mediating the links between puberty timing and adverse adult health outcomes. However, the paper could do a better job of distinguishing the results that are largely known and confirmatory from those that are novel. For instance, I found the link between the effects of biomarkers mediating pubertal timing and adverse adult health outcomes, such as that between Parabacteroides and lumbar spine osteoporosis and total body BMD, particularly interesting and quite novel. Overall, this is a nice study that provides a valuable approach to distill out the key genetic factors mediating pubertal timing and its impact on adult health that overcomes previous limitations of measuring pubertal timing in boys and girls.

Below are some specific questions and comments for your consideration:

1. There are many English language errors throughout. Have the language carefully proofread by a Native English speaker or a professional editing service.
2. Which pubertal height growth phenotype was included in this study? There were three in the Cousminer (2013) paper.
3. Lines 71-73: Explain how you determined that the significant loci were enriched for association with the other traits (CAD, blood pressure, etc).
4. Line 73: Change “Eighteen SNPs were not reported” to “Eighteen loci were not reported” to clarify that you checked for overlap of LOCI rather than individual SNPs. Also line 219- refer to the novel associations as “loci” rather than “SNPs”
5. Line 82: This description of fine-mapping is not clear. You performed fine-mapping for 266 association signals, and identified only 19 loci? What does this mean? That nineteen loci could be fine-mapped to a single causal variant..? What happened to the rest of the loci? Provide some

description of how many loci could be fine-mapped to a credible set of SNPs less than some number, such as one SNP, ten SNPs, etc.

6. Lines 92-94: Help the reader understand what it means by providing some interpretation of “We found 27 of the 266 lead SNPs surpassing the Bonferroni correction P...” – what does the Q_{snp} Pval signify?

7. Line 105: Provide a brief intuition of how UTMOST and FUSION work.

8. Line 106: In the TWAS and PWAS, when were the expression measurements made, and in which populations? Does it matter that the gene expression was likely measured in adults (and likely quite older adults) rather than in adolescents? How might this impact the results? Also, were the protein levels measured in healthy individuals or in cases with some disease status (INTERVAL and Atherosclerosis Risk in Communities study)? This study would be improved by using general population-based protein levels/pQTLs, such as from the UK Biobank.

9. Line 208: In what population were these dietary factors assessed? Were the participants adults or adolescents? Does it matter if these dietary factors were not measured in pre-pubertal children or pubescent adolescents? Clearly, dietary habits can change over time, so it would be hard to imagine that measuring coffee intake in a 50-year-old would be relevant for their pubertal timing 35 years earlier.

10. Lines 222-245: Many of these genes are well known to be involved in pubertal timing and are therefore more like positive controls than novel results. This paragraph would be more interesting and informative if you focused on 1) genes at the NOVEL loci you have identified, and 2) whether any of the prioritized genes differ from those reported previously or clear up any loci that were previously ambiguous.

11. Line 252: Again, the role of the HPG axis is “old news”—it would be much more interesting to focus on the novel tissue enrichment results and speculate on the role of the cerebellar hemisphere, cerebral cortex, and pancreas (as you do for pancreas, but please acknowledge what is known and what is novel in this study.)

12. Lines 265-267: Note there is a newer pubertal growth GWAS with a much larger sample size (Bradfield et al 2024: <https://pubmed.ncbi.nlm.nih.gov/38229171/>) which also examines adult health correlations in depth.

13. Did your analyses unearth any novel adult health links? Again, it is not new that early puberty is a risk factor for metabolic syndrome, so you need to highlight the novelty your study brings or whether your analysis was simply confirmatory.

14. The mediation analysis with biomarkers such as HDL-C, alanine aminotransferase, etc, is the most interesting novel aspect of your study and should be highlighted as such. In particular, the link between gut microbiota and osteoporosis is fascinating and the discussion of this point is excellent, drawing a line between Parabacteroides and the promotion of calcium absorption to prevent osteoporosis.

15. Lines 331-332: As mentioned above, there is a newer, larger pubertal height growth GWAS (the Bradfield paper), so please consider updating the results to incorporate that data rather than noting the small sample size of the previous paper as a limitation. Or, explain that future work could be done with the newer/larger data.

16. Line 332: Additional limitations should be added:

-Cell data was assessed in mouse, not human

-Measurement timing of TWAS, PWAS, and food intake data may not be ideal if done in adults

17. The BrainXscan results are not discussed at all. Please add a paragraph on these results to the Discussion.

18. Line 348: I believe the summary stats for the GWAS of age at menarche were published WITHOUT 23andMe samples included. Please check this and amend the text as necessary.

19. Lines 399-402: As you correctly mention, the Tanner stage study sample size is quite small (I am not criticizing this, just pointing it out)—do you think this could impact the SCOUTJOY results? The SCOUTJOY results are plausible and make sense, so perhaps this is not a concern.

20. Line 424: I am not familiar with FUSION and UTMOST, please add a short description of these.

21. Line 430: How does the FUSION/UTMOST TWAS analysis determine likely causal genes? Is there any element of colocalization analysis between the eQTL and GWAS signals? Similarly, for pQTL and GWAS signals?

22. The title for Table 1 should be “Novel loci associated with mvPuberty”. Add which reference genome build variant positions are reported for. “SNP function” does not add anything unless we know that the lead SNP is the most likely causal SNP. This column can be removed.

23. Extended data Fig. 1 and 3 are very hard to read—make the axis along the bottom larger so it is legible and the 0 line darker. Perhaps this could be split across multiple pages to make it easier to read, with larger text.

Reviewer #2 (Remarks to the Author):

The authors conduct a multivariate GWAS using a genomic structural equation modeling (SEM) approach to increase discovery power across related traits. The resulting 'Common factor' shows an appropriate model fit (Figure 2b) with strongest weighting for age at voice break (1.03) followed by age at facial hair (0.82), menarche (0.70) and then less for height spurt (0.40). However this leads to only modest progress, finding only 18 new signals out of a total of 266 independent variants.

Of more interest are some of the causal analyses that implicate mediators of the links between puberty and aging related diseases and also expression, proteins and dietary determinants of puberty timing.

Comments

QSNP heterogeneity - 27 of 266 SNPs show significant heterogeneity - what is the source of this, e.g. Sex differences or specific puberty features? Are these 27 SNPs still considered to be valid signals for puberty?

SCOUTJOY - describe the sample size available for Tanner stage in boys and girls and comment whether there is sufficient power in this sample as a test for sex differences.

"Transcriptomic and proteome imputation" is used to implicate causal genes - clarify in the Abstract that this is a causal analysis rather than the often used simpler approach to link genes to association signals

Genetic correlations with puberty timing - is positive with lifespan (which is logical) but why negative with healthspan? Is healthspan modelled in the opposite direction? If so this seems illogical.

"Early puberty timing causally increases alanine aminotransferase and HbA1c, decreases HDL-C, and is associated with higher risk of aging, with a mediated proportion of 5.25%, 5.25%, and 9%, respectively" - they report that these biomarkers show strong genetic correlations, so it is important to clarify if these mediation estimates are independent of each other? i.e. Do they together explain 19.5% of the association between puberty and aging?

MR with 25 dietary factors and micronutrients - describe the source of this data? Are these data also from UK Biobank and if so how were they derived?

They report that claim that "Controlling oil fish and retinol intake is beneficial in promoting healthy pubertal development" - while they show that higher intakes is associated with earlier puberty, it would be interesting to also test if this might lead to healthier adult outcomes.

Other downstream analyses seem to be conducted because they can rather than because of any well rationalised question and without meaningful interpretation - e.g. Brain imputation links 37 brain features to puberty timing. What do we learn from this?

REVIEWER COMMENTS

Reviewer #1 (Remarks to the Author):

This study models a common genetic component across various measures of pubertal timing (mvPuberty) to determine common genetic loci associated with the pubertal initiation program across both sexes. This is an interesting paper with some novel results; the study reports a handful of novel loci that have not previously been reported and provides additional insight into the biomarkers mediating the links between puberty timing and adverse adult health outcomes. However, the paper could do a better job of distinguishing the results that are largely known and confirmatory from those that are novel. For instance, I found the link between the effects of biomarkers mediating pubertal timing and adverse adult health outcomes, such as that between Parabacteroides and lumbar spine osteoporosis and total body BMD, particularly interesting and quite novel. Overall, this is a nice study that provides a valuable approach to distill out the key genetic factors mediating pubertal timing and its impact on adult health that overcomes previous limitations of measuring pubertal timing in boys and girls.

Below are some specific questions and comments for your consideration:

1. There are many English language errors throughout. Have the language carefully proofread by a Native English speaker or a professional editing service.

Thank you for your advice. We have revised our manuscript carefully and polished our language by a professional English editing service and the certificate is shown below.

This certificate may be verified on the AJE website using the verification code AAEC-63C1-3A09-E5CD-674P (<https://china.aje.com/api/certificate/AAEC-63C1-3A09-E5CD-674P/pdf>).

"REDACTED"

[The figure here shows an editing certificate confirming that the language in the manuscript was professionally edited. This has been removed to avoid reproducing third party images.]

2. Which pubertal height growth phenotype was included in this study? There were three in the Cousminer (2013) paper.

Thank you for pointing out this issue. We used GWAS data for the take-off phase of the growth spurt from the Cousminer (2013) paper in our study. The reason is that, based on previous reports and Genomic SEM adaptation conditions, data used to perform Genomic SEM should have sufficient SNP-based heritability and qualified GWAS signal ($h^2 > 0.05$ and $\chi^2 > 1.05$) (1,2). The other two GWAS in the Cousminer paper did not fulfil this requirement (χ^2 for the other two GWAS is 1.044 and 1.02, respectively), so we used GWAS data for the take-off phase of the growth spurt in our study ($h^2 = 0.419$ and $\chi^2 = 1.071$). Based on your advice, we've added the relevant description in the Methods section.

Methods section, line 460-465: We obtained four univariate input GWAS datasets from participants of European ancestry, encompassing age at menarche⁵, age at voice break¹⁴, age at first facial hair¹⁴, and the take-off phase of the growth spurt⁸. All input GWASs passed the Genomic SEM criteria, were estimated with LD score regression ($h^2 > 0.05$ and $\chi^2 > 1.05$), had existing ethical permissions from respective institutional review boards and included participant informed consent with rigorous quality control.

- (1). Karlsson Linnér R, Mallard TT, Barr PB, et al. Multivariate analysis of 1.5 million people identifies genetic associations with traits related to self-regulation and addiction. *Nat Neurosci.* 2021;24(10):1367-1376. doi:10.1038/s41593-021-00908-3
 - (2). Bulik-Sullivan BK, Loh PR, Finucane HK, et al. LD Score regression distinguishes confounding from polygenicity in genome-wide association studies. *Nat Genet.* 2015;47(3):291-295. doi:10.1038/ng.3211
3. Lines 71-73: Explain how you determined that the significant loci were enriched for association with the other traits (CAD, blood pressure, etc).

Thank you for pointing out this issue. Based on previous reports and FUMA instructions (3,4), we used our mvPuberty GWAS as input data in FUMA analysis, in which GWAS-catalog, a database of reported SNP-trait associations, enables us to determine if significant loci are enriched for the association with other traits. Detailed SNP-trait associations were shown in Supplementary Table 8 in our study. Based on your advice, we've added the relevant description in the Methods and Results section.

Methods section, line 535-537: To determine if any lead SNPs in mvPuberty show evidence of pleiotropic associations, we looked up GWAS-significant associations ($P < 5 \times 10^{-8}$) via the GWAS Catalog database in FUMA analysis⁹².

Results section, line 71-75: We identified 266 lead SNPs in 197 genomic loci ($P < 5 \times 10^{-8}$) (Fig. 2c; Supplementary Table 6) enriched for traits such as the risk of developing coronary artery disease, systolic and diastolic blood pressure, height, and the risk of developing insomnia according to the GWAS catalogue database (Supplementary Table 8).

- (3). Rosoff DB, Mavromatis LA, Bell AS, et al. Multivariate genome-wide analysis of aging-related traits identifies novel loci and new drug targets for healthy aging. *Nat Aging.* 2023;3(8):1020-1035. doi:10.1038/s43587-023-00455-5
 - (4). MacArthur J, Bowler E, Cerezo M, et al. The new NHGRI-EBI Catalog of published genome-wide association studies (GWAS Catalog). *Nucleic Acids Res.* 2017;45(D1): D896-D901. doi:10.1093/nar/gkw1133
4. Line 73: Change “Eighteen SNPs were not reported” to “Eighteen loci were not reported” to

clarify that you checked for overlap of LOCI rather than individual SNPs. Also line 219- refer to the novel associations as “loci” rather than “SNPs”

Thanks for your helpful suggestion. We have revised our manuscript according to your advice.

Results section, line 75-76: Eighteen loci were not reported in the four input GWASs.

Discussion section, line 241-245: Common loci explained 9.43% of the variation in pubertal timing in our multivariate analysis, whereas they explained 6.67–20.29% of the variation in previous univariate analyses^{5,7,14}. Eighteen loci were not previously identified in the input GWASs, highlighting the robust statistical power of our multivariate GWAS.

5. Line 82: This description of fine-mapping is not clear. You performed fine-mapping for 266 association signals, and identified only 19 loci? What does this mean? That nineteen loci could be fine-mapped to a single causal variant? What happened to the rest of the loci? Provide some description of how many loci could be fine-mapped to a credible set of SNPs less than some number, such as one SNP, ten SNPs, etc.

Thank you for pointing out this issue.

GWAS typically provides significant association signals (SNPs) that are linked but not essentially causal to a trait. Since GWAS SNPs are usually correlated with neighboring SNPs via linkage disequilibrium, they can be used as surrogates for large genomic regions that contain unmeasured SNPs that are causal to a trait. Fine-mapping of a summary statistic can determine the causal variants by identifying and analyzing the regions containing each of the lead GWAS SNPs. Thus, in our study, we analyzed 266 regions of interest by including all variants up- and down-stream of the 266 lead SNPs at a 250 kb window from mvPuberty GWAS. We performed fine-mapping for the 266 regions, and refined the maximum number of outputs for each region to one SNP so that we can identify variants with the highest causal likelihood. For each of the 266 regions, we found 266 potentially causal variants (Supplementary Table 10), which are correlated with the 266 original association signals from mvPuberty, while are not necessarily among the original SNPs. Because variants with posterior probability >0.95 are considered as true causal variants (3), in our analysis, we found that 19 variants passed the threshold, suggesting that these 19 loci are causal. Based on your advice, we’ve revised the description for fine-mapping in the Methods and Results.

Results section, line 85-87: We performed fine-mapping for the 266 regions of interest by including all variants up- and downstream of the 266 lead SNPs at a 250 kb window from the mvPuberty GWAS. As a result, we identified 19 causal variants (posterior probability > 0.95).

Methods section, line 542-546: We analysed 266 regions of interest by including all variants up- and downstream of the 266 lead SNPs at a 250 kb window from the mvPuberty GWAS. We performed fine-mapping for the 266 regions and refined the maximum number of outputs for each region to one SNP so that we could identify variants with the highest causal likelihood.

(3). Rosoff DB, Mavromatis LA, Bell AS, et al. Multivariate genome-wide analysis of aging-related traits identifies novel loci and new drug targets for healthy aging. *Nat Aging*. 2023;3(8):1020-1035. doi:10.1038/s43587-023-00455-5

6. Lines 92-94: Help the reader understand what it means by providing some interpretation of “We found 27 of the 266 lead SNPs surpassing the Bonferroni correction P...” – what does the Q_{SNP} P_{val} signify?

Thanks for your helpful suggestion. The null hypothesis of Q_{SNP} test is that SNP associations on the single-phenotype GWAS are statistically mediated by mvPuberty. Thus, statistically significant Q_{SNP} tests (surpassing the Bonferroni correction *P*) suggest that SNP impacts the single-phenotype

GWAS by pathways other than the shared genetics of pubertal timing modeled by mvPuberty. In our analysis, 27 of the 266 lead SNPs surpassing Bonferroni correction P in Q_{SNP} tests indicated that these SNPs influence single-phenotype GWAS (such as age of menarche) by other pathways rather than mvPuberty. Based on your advice, we've added the relevant description in the Results. Results section, line 95-102: We evaluated whether the SNP associations are appropriately modelled through a multivariate framework using Q_{SNP} heterogeneity statistics, in which significant Q_{SNP} tests in mvPuberty would suggest that SNPs impact single-phenotype GWASs via pathways other than the shared genetics of pubertal timing modelled by mvPuberty. We found that 27 of the 266 lead SNPs surpassed the Bonferroni-corrected threshold, $P < 1.88 \times 10^{-4}$ (0.05/266), whereas none of the 18 newly reported loci had a Q_{SNP} $P < 1.88 \times 10^{-4}$ (Supplementary Table 6). The results indicated that the 18 newly reported loci impact four input puberty phenotypes via mvPuberty.

7. Line 105: Provide a brief intuition of how UTMOST and FUSION work.

Thank you for your advice. We've added the relevant description in the Results section.

Results section, line 115-121: Functional summary-based imputation (FUSION) TWAS integrates gene expression measurements with summary association statistics from GWASs to identify genes whose cis-regulated expression is associated with complex traits. Unified test for molecular signatures (UTMOST) combines multiple single-tissue associations into a single powerful metric to quantify overall gene-trait associations at the organism level. We performed TWAS using FUSION and UTMOST to identify gene-level associations with the mvPuberty genetic signature.

8. Line 106: In the TWAS and PWAS, when were the expression measurements made, and in which populations? Does it matter that the gene expression was likely measured in adults (and likely quite older adults) rather than in adolescents? How might this impact the results? Also, were the protein levels measured in healthy individuals or in cases with some disease status (INTERVAL and Atherosclerosis Risk in Communities study)? This study would be improved by using general population-based protein levels/pQTLs, such as from the UK Biobank.

Thank you for pointing out this issue. For TWAS in our study, gene expression measurements are expression quantitative trait loci (eQTL) data from GTEx v8 (5,6), which includes European-American subjects aged 20-70 and died from multiple causes (such as traumatic injury, cerebrovascular, heart disease, etc.). The principle of TWAS is to build a model to predict (impute) gene expression levels from samples with matched genotypes and expression levels, and it is challenging to develop robust and accurate imputation models with limited sample sizes (6). Thus, due to age-dependent nature of transcription regulation (6), we totally agree with you that gene expression, which is measured in adults and used to build imputation models, has the potential to influence overall gene-trait association. Based on your advice, we've added relevant limitation in the Discussion section.

Expression of some genes may decrease with age, which could result in imputation models missing genes that are highly expressed during puberty. For example, the expression of *LEPR*, identified in our study, was shown to decrease from prepubertal to adult stage in mice (7). Thus, there is a chance that some other puberty-related genes were not identified through our TWAS analysis due to potential low expression in the adulthood. In our study, we used gene expression measurements from large-scale consortia GTEx by integrating multiple tissues as imputation models, which can increase the number of testable genes and identify more gene-trait associations (5,6). In addition, we used two parallel TWAS approaches (including two gene expression weights

calculated by different statistical approaches) to explore the genes related to mvPuberty, which can improve the power of TWAS (8). Although we tried various solutions to reduce the bias due to inaccurate imputation models, our analyses are still not ideal due to the current difficulty in obtaining gene expression data in adolescents. Based on your advice, we've added relevant limitation in the Discussion section.

For PWAS in our study, we used protein quantitative trait loci (pQTL) from INTERVAL and Atherosclerosis Risk in Communities study as reference weights to explore the proteins related to mvPuberty. Participants in INTERVAL study are generally in good health because blood donation criteria exclude people with a history of major diseases (9). Atherosclerosis Risk in Communities (ARIC) is a recent prospective study to investigate the etiology of atherosclerosis and its clinical sequelae and variation in cardiovascular risk factors, medical care, and disease by race, sex, place, and time (10). Although the pQTL study for ARIC did not directly mention whether it included only healthy individuals or cases with disease status, it is likely that it included participants with atherosclerosis, which may affect the accuracy of our PWAS results and potentially prioritize atherosclerosis-related proteins. We totally agree with you that general population-based protein levels/pQTLs can definitely improve our results, unfortunately, pQTL for plasma protein in large samples such as UKB are currently difficult for us to obtain, so we used two current studies with large pQTL data to improve the accuracy of our study. Based on your advice, we've added the relevant limitation in the Discussion section.

Discussion section, line 445-448: In addition, food intake data for MR and expression measurements for TWASs and PWASs, derived from adults or individuals with disease status, may not be ideal. Moreover, cell data for cell type enrichment analysis, assessed in mice, may not be comparable to those of humans.

- (5) Feng H, Mancuso N, Gusev A, et al. Leveraging expression from multiple tissues using sparse canonical correlation analysis and aggregate tests improves the power of transcriptome-wide association studies. *PLoS Genet.* 2021;17(4): e1008973. doi:10.1371/journal.pgen.1008973
- (6) Hu Y, Li M, Lu Q, et al. A statistical framework for cross-tissue transcriptome-wide association analysis. *Nat Genet.* 2019;51(3):568-576. doi:10.1038/s41588-019-0345-7
- (7) Zampieri TT, Bohlen TM, Silveira MA, et al. Postnatal Overnutrition Induces Changes in Synaptic Transmission to Leptin Receptor-Expressing Neurons in the Arcuate Nucleus of Female Mice. *Nutrients.* 2020;12(8):2425. doi: 10.3390/nu12082425
- (8) Uellendahl-Werth F, Maj C, Borisov O, et al. Cross-tissue transcriptome-wide association studies identify susceptibility genes shared between schizophrenia and inflammatory bowel disease. *Commun Biol.* 2022;5(1):80. doi: 10.1038/s42003-022-03031-6
- (9) Sun BB, Maranville JC, Peters JE, et al. Genomic atlas of the human plasma proteome. *Nature.* 2018;558(7708):73-79. doi:10.1038/s41586-018-0175-2
- (10) The Atherosclerosis Risk in Communities (ARIC) Study: design and objectives. The ARIC investigators. *Am J Epidemiol.* 1989;129(4):687-702.

9. Line 208: In what population were these dietary factors assessed? Were the participants adults or adolescents? Does it matter if these dietary factors were not measured in pre-pubertal children or pubescent adolescents? Clearly, dietary habits can change over time, so it would be hard to imagine that measuring coffee intake in a 50-year-old would be relevant for their pubertal timing 35 years earlier.

Thank you for pointing out this issue. These GWASs for dietary factors are from UKB participants, aged from 40 to 69. Mendelian randomization (MR) employs genetic variants as proxies for the exposure of interest, simulating a randomized controlled trial in which genetic alleles are randomly assigned during conception. MR analysis relies on Mendel's second law and the random distribution of genetic variants at conception, making them unlikely to be associated with potential confounders (including age, behavioural factors, etc.) (11). There is a clear, albeit modest, genetic component to diet, as demonstrated by heritability and individual genetic associations (12), which means that dietary factors meet the conditions for MR. Therefore, based on previous reports (12,13), we selected SNP instruments robustly associated with the dietary factors as proxies for the dietary factors, and used MR to explore whether SNP instruments represented dietary factors are relevant for pubertal timing. Although MR analysis seems to reduce potential confounding factor (including age, behavioural factors, etc.) by employing genetic variants as proxies for the exposure, dietary habits of different age groups indeed may have an impact on the results due to changes in dietary habits. Therefore, we totally agree with you that dietary factors, not measured in pre-pubertal children or pubescent adolescents, may not be ideal. Based on your advice, we've added relevant descriptions in the Methods section and limitation in the Discussion section.

Methods section, line 640-643: We performed MR with 25 dietary factors and micronutrients derived from a GWAS with participants of European ancestry in the UKB datasets to investigate whether mvPuberty status is causally influenced by dietary factors and micronutrients represented by SNPs (Supplementary Table 30).

Discussion section, line 445-447: In addition, food intake data for MR and expression measurements for TWASs and PWASs, derived from adults or individuals with disease status, may not be ideal.

(11) Evans DM, Davey Smith G. Mendelian Randomization: New Applications in the Coming Age of Hypothesis-Free Causality. *Annu Rev Genomics Hum Genet.* 2015;16: 327-350. doi:10.1146/annurev-genom-090314-050016

(12) Cole JB, Florez JC, Hirschhorn JN. Comprehensive genomic analysis of dietary habits in UK Biobank identifies hundreds of genetic associations. *Nat Commun.* 2020;11(1):1467. doi:10.1038/s41467-020-15193-0

(13) May-Wilson S, Matoba N, Wade KH, et al. Large-scale GWAS of food liking reveals genetic determinants and genetic correlations with distinct neurophysiological traits. *Nat Commun.* 2022;13(1):2743. doi:10.1038/s41467-022-30187-w

10. Lines 222-245: Many of these genes are well known to be involved in pubertal timing and are therefore more like positive controls than novel results. This paragraph would be more interesting and informative if you focused on 1) genes at the NOVEL loci you have identified, and 2) whether any of the prioritized genes differ from those reported previously or clear up any loci that were previously ambiguous.

Thanks for your helpful suggestion. Based on your advice, we've added discussions on new loci and genes in the Discussion section.

Discussion section, line 249-267: The gene closest to rs11556924 is *ZC3HCl*, which can regulate mitotic entry time and may promote carcinogenesis^{19,20}, was not previously associated with pubertal timing. *ZC3HCl* is closely related to the risk of developing essential hypertension through endothelial dysfunction²¹. In addition, maintaining endothelial function is essential for the nitric

oxide synthase pathway, which regulates the expression of genes associated with reproduction ^{21,22}. Therefore, *ZC3HCI* may regulate puberty-related reproductive maturation by affecting endothelial function. Another plausible causal SNP, rs5742915, located on chr15q14.1, is associated with lung function and body development ^{23,24}. The gene closest to rs5742915 is *PML*, also known as *TRIM19*. *TRIM19* is the core component of promyelocytic leukaemia nuclear bodies, which are tightly associated with the nuclear matrix. Promyelocytic leukaemia nuclear bodies, which interact with different proteins, play many instrumental roles in DNA damage responses, apoptosis, cellular senescence, and angiogenesis ²⁵. Previous studies have shown that the knockdown of *PML* inhibits the proliferation of oestrogen receptor-positive breast cancer cells and promotes the expression of oestrogen receptor target genes, thereby increasing oestrogen receptor-positive breast cancer cell stemness ²⁶. For the first time, we found that *PML* also plays an important role in the timing of puberty, which is related to the regulation of oestrogen receptor target genes.

11. Line 252: Again, the role of the HPG axis is “old news”—it would be much more interesting to focus on the novel tissue enrichment results and speculate on the role of the cerebellar hemisphere, cerebral cortex, and pancreas (as you do for pancreas, but please acknowledge what is known and what is novel in this study.)

Thanks for your helpful suggestion. Based on your advice, we’ve revised discussions in the Discussion section.

Discussion section, line 299-311: In addition, previous studies have shown that the cerebellar hemisphere and cerebral cortex, identified in our tissue enrichment analysis for mvPuberty, regulate muscle tone and coordinate movement, cognitive abilities and social behaviours through connections between neurons ³⁷. Moreover, connections between neurons in the cerebellar hemisphere and cerebral cortex are regulated by the phagocytosis of microglia during brain development ³⁸. Microglia are responsive to oestrogens and prostaglandins, which regulate each other’s synthesis in a cerebellar hemisphere- and cerebral cortex-dependent manner, thereby modulating neurogenesis and neuronal survival ^{38,39}. Therefore, adolescents with early puberty status are at significantly greater risk of poor mobility and cognitive performance than those with typical pubertal timing are ⁴⁰, which may be attributed to abnormal changes in oestrogens and prostaglandins, leading to abnormal function of microglia in brain development.

Discussion section, line 316-321: These previous findings support the links between various pancreatic cells and the timing of puberty ^{42,43}. For example, human growth hormone levels also peak during puberty and are associated with pubertal timing via positive effects on maintaining the pancreatic B-cell mass ⁴⁴. In summary, our enrichment analysis further confirmed that pancreatic cells play an important role in the onset of puberty by secreting insulin and glucagon to meet the needs of blood sugar changes.

12. Lines 265-267: Note there is a newer pubertal growth GWAS with a much larger sample size (Bradfield et al 2024: <https://pubmed.ncbi.nlm.nih.gov/38229171/>) which also examines adult health correlations in depth.

Thank you very much for pointing this out. We’ve revised the description of our pubertal growth GWAS in the manuscript. As you mentioned, the newer pubertal growth GWAS does examine adult health correlations in depth partly attributable to its larger sample size. The key feature of this newer GWAS is the trans-ethnic GWAS summary statistics, which is incompatible to Genomic SEM (<https://github.com/GenomicSEM/GenomicSEM/wiki/2.-Important-resources-and-key-information>: In order for LD-score regression to produce accurate results, it is critical that the

users include only GWAS summary statistics that were calculated within a single ethnic group and matched with LD scores for the same ethnicity). In addition, subsequent causal inference by MR analysis also requires GWAS summary statistics with same ethnicity based on MR instructions (14). Although it is not suitable for cross-trait multivariate GWAS in our study, inspired by your suggestion, trans-ethnic multivariate GWAS may be beneficial to further understand genetic characteristic for pubertal timing. Therefore, we used GWAMA (15), an approach to meta-analyze trans-ethnic GWAS summary statistics, to combine our mvPuberty GWAS and the newer pubertal growth GWAS. The results showed that only 3 of our 266 lead loci (rs1568182, rs17400325, and rs10124823) were not significant, indicating our mvPuberty GWAS are in general agreement with the latest trans-ethnic GWAS. Admittedly, our trans-ethnic meta-analysis is preliminary and we believe further trans-ancestral multivariate genome-wide association study in the future would provide more valuable information. Based on your advice, we've added the relevant descriptions in the Discussion section.

Discussion section, line 441-442: and trans-ancestral multivariate GWASs with larger sample sizes for pubertal timing need to be performed in the future.

(14) Mägi R, Morris AP. GWAMA: software for genome-wide association meta-analysis. *BMC Bioinformatics*. 2010; 11 :288. doi:10.1186/1471-2105-11-288

(15) Hemani G, Tilling K, Davey Smith G. Orienting the causal relationship between imprecisely measured traits using GWAS summary data. *PLoS Genet*. 2017;13(11): e1007081. doi:10.1371/journal.pgen.1007081

13. Did your analyses unearth any novel adult health links? Again, it is not new that early puberty is a risk factor for metabolic syndrome, so you need to highlight the novelty your study brings or whether your analysis was simply confirmatory.

Thank you for pointing out this issue. In our study, most of the adult health were chosen based on previous observational studies. Therefore, our study tends to verify and infer potential causal links at the genomic level between early puberty and adverse adult health outcomes. And what's more novel is that we included multiple biomarkers as mediating variables in our analyses, as you acknowledged in your comments. Therefore, based on your advice, we've added detailed discussion about the mediating links in the Discussion section.

Discussion section, Line 357-359: we confirmed significant genetic correlations of mvPuberty status with ageing and the risk of developing cardiovascular diseases, osteoporosis, endometrial cancer, and metabolic syndrome.

Discussion section, Line 367-369: Therefore, for the first time, we analysed the mediating role of these metabolic biomarkers in the associations between early pubertal timing and adverse health outcomes in adults.

14. The mediation analysis with biomarkers such as HDL-C, alanine aminotransferase, etc, is the most interesting novel aspect of your study and should be highlighted as such. In particular, the link between gut microbiota and osteoporosis is fascinating and the discussion of this point is excellent, drawing a line between Parabacteroides and the promotion of calcium absorption to prevent osteoporosis.

Thanks for your helpful suggestion. Based on your advice, we've revised the description for health links and mediation effects in the Discussion section.

Discussion section, Line 367-408: Therefore, for the first time, we analysed the mediating role of these metabolic biomarkers in the associations between early pubertal timing and adverse health outcomes in adults.

For cardiovascular diseases, we found that HDL-C levels mediated the causal effect of pubertal timing on the risk of developing coronary artery disease and heart failure. Observational, experimental, and genetic studies have shown that lower HDL cholesterol levels are associated with a greater risk of developing atherosclerosis and related cardiovascular complications⁶³⁻⁶⁵. HDL-C is intricately involved in cholesterol transport and inflammation modulation⁶⁶ and is associated with altered activity of the hypothalamus–pituitary–adrenal axis⁶⁷. For example, early puberty causes abnormal secretion of sex hormones, affecting the expression of sex hormone-binding receptors and proteins, including oestrogen receptor alpha and sex hormone-binding globulin, which in turn interact with lipid metabolism genes, leading to reduced expression of lipid transporters⁶⁸⁻⁷⁰. These studies suggest a closer connection between early pubertal timing and lower HDL cholesterol^{71,72}. However, the mediating role of HDL cholesterol in the relationship between pubertal timing and the risk of developing cardiovascular diseases has rarely been reported. Our results, at the genomic level, contribute to a better understanding of HDL-C as a mediator between early pubertal timing and increased risk of developing cardiovascular diseases. For the ageing process, we found that early pubertal causally elevates alanine aminotransferase, HbA1c, and HDL-C levels and is associated with greater ageing. Previous multivariate genome-wide analyses of ageing-related traits revealed that HbA1c and HDL-C are novel targets for accelerated ageing and that blood glucose and lipid levels are risk factors for ageing and cognitive impairment^{73,74}. In our study, the mvPuberty SNP-heritable cell type included pancreatic cells, suggesting that early pubertal timing may influence the level of HbA1c, which reflects the average blood glucose level over the last 90 days, by controlling pancreatic cell secretion to accelerate ageing, in line with previous findings⁷⁴. Clinical and epidemiological studies have reported an association between elevated alanine aminotransferase levels and accelerated ageing and increased cardiovascular morbidity^{75,76}. The mechanism by which puberty onset affects alanine aminotransferase is unknown, but studies of hepatocyte damage have shown that an abnormal increase in sex hormones leads to increased hepatocyte damage in mice⁷⁷. Therefore, the causal link between early pubertal timing and increased alanine aminotransferase levels, implying hepatocyte damage, may be related to a steep increase in sex hormone levels. We analysed common biomarkers as mediators of early puberty and multiple adverse health outcomes for the first time and reported that earlier puberty can cause abnormalities in blood glucose, lipids and liver function, which can lead to an increased incidence of adverse health outcomes. Our findings indicate that close monitoring of blood glucose, blood lipids and liver function in adolescents with early pubertal timing is more beneficial for their health later in life.

15 Lines 331-332: As mentioned above, there is a newer, larger pubertal height growth GWAS (the Bradfield paper), so please consider updating the results to incorporate that data rather than noting the small sample size of the previous paper as a limitation. Or, explain that future work could be done with the newer/larger data.

Thank you for pointing out this issue. As you mentioned, the newer pubertal growth GWAS with larger sample size is advantageous. The key feature of this newer GWAS is trans-ethnic GWAS summary statistics, which is incompatible to Genomic SEM (To produce accurate results, in Genomic SEM analysis, it is critical that the users include only GWAS summary statistics that

were calculated within a single ethnic group and matched with LD scores for the same ethnicity). Although it is not suitable for cross-trait multivariate GWAS in our study, inspired by your suggestion, we believe trans-ethnic multivariate GWAS could further help us understand genetic characteristic for puberty timing. Therefore, we used GWAMA (15), an approach to meta-analyze trans-ethnic GWAS summary statistics, to combine our mvPuberty GWAS and the newer pubertal growth GWAS. The results showed that only 3 of our 266 lead loci (rs1568182, rs17400325, and rs10124823) were not significant, indicating our mvPuberty GWAS are in general agreement with the latest trans-ethnic GWAS. Admittedly, our trans-ethnic meta-analysis is preliminary and we believe further trans-ancestral multivariate genome-wide association study in the future would provide more valuable information. Based on your advice, we've added the relevant descriptions in the Discussion section.

Discussion section, line 441-442: and trans-ancestral multivariate GWASs with larger sample sizes for pubertal timing need to be performed in the future.

(15) Hemani G, Tilling K, Davey Smith G. Orienting the causal relationship between imprecisely measured traits using GWAS summary data. PLoS Genet. 2017;13(11):e1007081. doi:10.1371/journal.pgen.1007081

16 Line 332: Additional limitations should be added:

-Cell data was assessed in mouse, not human

-Measurement timing of TWAS, PWAS, and food intake data may not be ideal if done in adults

Thanks for your helpful suggestion. Based on your advice, we've added these limitations in the Discussion section.

Discussion section, line 445-448: In addition, food intake data for MR and expression measurements for TWASs and PWASs, derived from adults or individuals with disease status, may not be ideal. Moreover, cell data for cell type enrichment analysis, assessed in mice, may not be comparable to those of humans.

17 The BrainXscan results are not discussed at all. Please add a paragraph on these results to the Discussion.

Thanks for your helpful suggestion. Based on your advice, we've added relevant descriptions for BrainXscan results in the Discussion section.

Discussion section, line 322-354: Notably, earlier puberty has lasting impacts on brain maturation, including strengthening of neural connections and organizing and fostering the acquisition of cognitive abilities and social behaviours, partly due to changes in pubertal hormones⁴⁵⁻⁴⁷. In our BrainXcan analysis, we found changes in the mean diffusivity and radial diffusivity of several white matter microstructures, including the superior corona radiata, superior longitudinal fasciculus, external capsule, and uncinate fasciculus, which affected emotional development⁴⁸⁻⁵⁰. These white matter microstructures were identified as possible subsequent neural signatures for advanced pubertal status due to abnormal changes in gonadal sex steroid hormones⁴⁸⁻⁵⁰. These findings indicate that adult emotional disorders may be attributed to changes in white matter microstructures caused by abnormal gonadal sex steroid hormones during puberty. In addition, we also detected changes in the mean diffusivity and radial diffusivity in two other white matter microstructures, the inferior cerebellar peduncle and cingulum cingulate gyrus, through BrainXcan analysis. The inferior cerebellar peduncle, which connects the cerebellum and other parts of the nervous system, plays an important role in locomotor and mental adaptation, age and brain lateralization and fatigue⁵¹⁻⁵³. The cingulate gyrus, which connects the cingulate cortex with other

brain regions, has also been associated with language, specifically phonological processing, including visuospatial episodic memory and spatial word memory^{54,55}. Previous studies have shown that gonadal sex steroid hormones regulate neurogenesis and neuronal survival in the inferior cerebellar peduncle and cingulum cingulate gyrus. For example, oestradiol and progesterone facilitate oligodendrocyte activity and stimulate the proliferation of Schwann cells that produce myelin proteins, thereby modulating neurogenesis and neuronal survival through regulating each other's synthesis^{56,57}. We found for the first time that the inferior cerebellar peduncle and cingulum cingulate gyrus are associated with pubertal timing, which could be attributed to the interaction between gonadal sex steroid hormones and neurogenesis and neuronal survival⁴⁸. In summary, when the body secretes pubertal hormones, specifically gonadal hormones, at an earlier age and at higher levels, the brain may experience accelerated proliferation and subsequent fine-tuning of white matter development during puberty. Future work incorporating longitudinal neuroimaging in parallel with pubertal measures may contribute to the understanding of individual variation in pubertal course and brain region development.

18 Line 348: I believe the summary stats for the GWAS of age at menarche were published WITHOUT 23andMe samples included. Please check this and amend the text as necessary.

Thank you for pointing out this issue. We apologize for this error when preparing our manuscript, which has been corrected in the manuscript.

Methods section, line 466-467: The GWAS for the age at menarche (n=252,514) included 40 studies from the ReproGen consortium (n=179,117) and UK Biobank (n=73,397)⁵.

19 Lines 399-402: As you correctly mention, the Tanner stage study sample size is quite small (I am not criticizing this, just pointing it out)—do you think this could impact the SCOUTJOY results? The SCOUTJOY results are plausible and make sense, so perhaps this is not a concern.

Thank you for pointing out this issue. As you mentioned, GWAS for Tanner stage has a small sample sizes (9,916), which may lead to insufficient statistical efficacy in heritability estimate and cross-traits analysis. However, in our study, we only tested heterogeneity of genetic effects for 266 lead loci, identified by multivariate GWAS, using value of effect sizes (beta) and standard error (se) in Tanner stage's GWAS. In SCOUTJOY analysis, the sample size was not involved in the calculation, so we believe that the sample sizes here don't have a large impact on the SCOUTJOY results. Based on your advice, we've added description for in the Methods section.

Methods section, line 519-521: We tested each of the lead SNPs from mvPuberty for differences in Tanner stage effect sizes between males and females via the values of effect sizes (beta) and standard errors (se) in the Tanner stage GWAS.

20 Line 424: I am not familiar with FUSION and UTMOST, please add a short description of these.

Thanks for your helpful suggestion. Based on your advice, we've added related description in the Methods section.

Methods section, line 550-563: The FUSION method involves three steps: (1) identify gene expression features that are cis-heritable (i.e., variants associated with gene expression within or near the genomic locus); (2) construct a linear predictor for each cis-heritable gene (i.e., a SNP-based prediction weight of the gene feature); and (3) calculate both TWAS test statistics incorporating these SNP-based prediction weights and summary-level GWAS Z scores. FUSION uses several penalized linear regression and Bayesian sparse linear mixed models (e.g., GBLUP, LASSO, elastic net, and BLSMM) and computes out-of-sample R² statistics to identify the best

model via cross-validation of each gene–GWAS model. UTMOST consists of two steps: (1) a single-tissue association test for 44 tissues is run and (2) gene–trait associations in 44 tissues are combined via the joint generalized Berk–Jones (GBJ) test. UTMOST uses the GBJ test with single-tissue association Z statistics and their covariance matrix as inputs to provide powerful inference results while explicitly taking the correlation among single-tissue test statistics into account even under a sparse alternative.

21 Line 430: How does the FUSION/UTMOST TWAS analysis determine likely causal genes? Is there any element of colocalization analysis between the eQTL and GWAS signals? Similarly, for pQTL and GWAS signals?

Thank you for pointing out this issue. As you mentioned, TWAS and PWAS using predicted expression can identify thousands of genes/proteins which are associated with complex traits and diseases, and cannot determine likely causal genes/proteins. Therefore, in our study, we further performed FOCUS analysis for results of FUSION TWAS/PWAS. FOCUS (Fine-mapping of causal gene sets) is designed to fine-map TWAS/PWAS statistics at genomic risk regions and prioritizes genes/proteins with strong evidence for causality (16). That’s why we considered our results to be causal in the manuscript. Based on your advice, we’ve added related description for FOCUS in the Methods section.

Methods section, line 567-570: We performed fine mapping for TWAS genes associated with mvPuberty, surpassing the Bonferroni corrected threshold, using FOCUS, which was designed for TWAS studies⁹⁵. FOCUS fine maps TWAS/PWAS statistics at genomic risk regions and prioritizes genes/proteins with strong evidence for causality.

(16) Mancuso N, Freund MK, Johnson R, et al. Probabilistic fine-mapping of transcriptome-wide association studies. *Nat Genet.* 2019;51(4):675-682. doi:10.1038/s41588-019-0367-1

22 The title for Table 1 should be “Novel loci associated with mvPuberty”. Add which reference genome build variant positions are reported for. “SNP function” does not add anything unless we know that the lead SNP is the most likely causal SNP. This column can be removed.

Thanks for your helpful suggestion. Based on your advice, we’ve revised the title for table 1 and removed the column of ‘SNP function’.

23 Extended data Fig. 1 and 3 are very hard to read—make the axis along the bottom larger so it is legible and the 0 line darker. Perhaps this could be split across multiple pages to make it easier to read, with larger text.

Thanks for your helpful suggestion. Based on your advice, we’ve revised the figures. If you think more revisions are needed, we would love to further revise the figures according to your guidance.

Reviewer #2 (Remarks to the Author):

The authors conduct a multivariate GWAS using a genomic structural equation modeling (SEM) approach to increase discovery power across related traits. The resulting 'Common factor' shows an appropriate model fit (Figure 2b) with strongest weighting for age at voice break (1.03) followed by age at facial hair (0.82), menarche (0.70) and then less for height spurt (0.40). However this leads to only modest progress, finding only 18 new signals out of a total of 266 independent variants.

Of more interest are some of the causal analyses that implicate mediators of the links between puberty and aging related diseases and also expression, proteins and dietary determinants of puberty timing.

Comments

1. Q_{SNP} heterogeneity - 27 of 266 SNPs show significant heterogeneity - what is the source of this, e.g. Sex differences or specific puberty features? Are these 27 SNPs still considered to be valid signals for puberty?

Thank you for pointing out this issue. The null hypothesis of Q_{SNP} test is that SNP associations on the single-phenotype GWAS are statistically mediated by mvPuberty. Thus, statistically significant Q_{SNP} tests (surpassing the Bonferroni correction P) suggest that SNP impacts the single-phenotype GWAS by pathways other than the shared genetics of pubertal timing modeled by mvPuberty. Therefore, 27 of the 266 lead SNPs surpassing the Bonferroni correction P in Q_{SNP} tests indicated that these SNPs influenced single-phenotype GWAS (such as age of menarche) by other pathways rather than mvPuberty. Based on the results of SCOUTJOY analysis that sex differences have little effect on our results, we speculate that source of heterogeneity may be specific puberty features. Although these 27 SNPs were considered to be invalid signals for mvPuberty in our cross-trait multivariate analysis, these 27 SNPs may be related to specific puberty features (detailed information was shown in Supplementary Table 6). Based on your advice, we've added related description in the Results section.

Results section, line 95-102: We evaluated whether the SNP associations are appropriately modelled through a multivariate framework using Q_{SNP} heterogeneity statistics, in which significant Q_{SNP} tests in mvPuberty would suggest that SNPs impact single-phenotype GWASs via pathways other than the shared genetics of pubertal timing modelled by mvPuberty. We found that 27 of the 266 lead SNPs surpassed the Bonferroni-corrected threshold, $P < 1.88 \times 10^{-4}$ (0.05/266), whereas none of the 18 newly reported loci had a Q_{SNP} $P < 1.88 \times 10^{-4}$ (Supplementary Table 6). The results indicated that the 18 newly reported loci impact four input puberty phenotypes via mvPuberty.

2. SCOUTJOY - describe the sample size available for Tanner stage in boys and girls and comment whether there is sufficient power in this sample as a test for sex differences.

Thank you for pointing out this issue. GWAS for Tanner stage has a small sample size (9,916), which may lead to insufficient statistical efficacy in heritability estimates and cross-traits analyses. However, in our study, we only tested heterogeneity of genetic effects for 266 lead loci, identified by multivariate GWAS meta-analysis, using value of effect sizes (beta) and standard error (se) in Tanner stage's GWAS. In SCOUTJOY analysis, the sample size was not involved in the

calculation. Therefore, we believe that the sample sizes here don't have a large impact on the SCOUTJOY results.

Methods section, line 519-521: We tested each of the lead SNPs from mvPuberty for differences in Tanner stage effect sizes between males and females via the values of effect sizes (beta) and standard errors (se) in the Tanner stage GWAS.

3. "Transcriptomic and proteome imputation" is used to implicate causal genes - clarify in the Abstract that this is a causal analysis rather than the often used simpler approach to link genes to association signals

Thanks for your helpful suggestion. Based on your advice, we've revised description for Transcriptomic and proteome imputation in the Abstract section.

Abstract section, line 12-13: Transcriptomic, proteome imputation and fine-mapping analysis revealed pubertal timing causal genes, including *KDM4C*, *LEPR*, *CCNC*, *ACPL1*, and *PCSK1L*.

4. Genetic correlations with puberty timing - is positive with lifespan (which is logical) but why negative with healthspan? Is healthspan modelled in the opposite direction? If so this seems illogical.

Thank you for pointing out this issue.

The healthspan GWAS consists of 300,477 unrelated, British-ancestry individuals from UK Biobank. The statistics were calculated by fitting Cox-Gompertz survival models with events defined as the first incidence of one of seven specific diseases (any cancer, diabetes, myocardial infarction, stroke, chronic obstructive pulmonary disease, dementia, and congestive heart failure) or death itself. As such, healthspan is highly dependent on the characteristics of the UK Biobank cohort, who were aged 40-69 years when they were recruited in 2006–2010 and of which two-thirds have yet to experience an age-related disease (17). Therefore, loci of healthspan GWAS have overrepresented effects on diseases of middle age (cancer, heart disease, etc) than age-related disease, which is a limitation acknowledged in the original study (17).

The parental lifespan GWAS consists of unrelated, European-ancestry individuals reporting a total of 512,047 mothers' and 500,193 fathers' lifespans. The statistics for each participating cohort were calculated by fitting Cox survival models to father's and mother's survival separately, adjusted for subject sex, at least 10 principal components, and study-specific covariates such as genotyping batch and array (18). Therefore, loci of parental lifespan GWAS don't have overrepresented effects on diseases of middle age.

In addition, in the original study of healthspan GWAS, mother's and father's age at death showed negative genetic correlations with healthspan (17). Inspired by your comments, we further used LDSC to detect the genetic correlations between healthspan and lifespan/longevity. The results showed that genetic correlations between healthspan and lifespan/longevity is negative (see figures below). Therefore, we tend to speculate that the negative genetic correlation between pubertal timing and healthspan in our analysis may potentially be attributed to the overrepresented effects on diseases of middle age from healthspan loci.

Due to the overrepresented effects on diseases of middle age, genetic correlation between pubertal timing and healthspan could be an indication of distinct underlying biology between pubertal timing and onset timing of specific middle age diseases, which might not necessarily be in the same context of parental lifespan and longevity, defined solely by the age. Based on your advice, we've added detailed definition for GWAS including healthspan, lifespan and longevity in the Supplementary tables 18 and 29.

Supplementary Table 18 and 29:

Aging	mvAge	1958774	Rosoff DB, Mavromatis LA, Bell AS, et al. Multivariate
	Healthspan (age the first incidence of seven specific diseases)	300477	Rosoff DB, Mavromatis LA, Bell AS, et al. Multivariate
	Parental lifespan (father's and mother's survival)	620911	Rosoff DB, Mavromatis LA, Bell AS, et al. Multivariate
	Frailty	175226	Rosoff DB, Mavromatis LA, Bell AS, et al. Multivariate
	Longevity (surviving at or beyond the age corresponding to the 90th/99th survival percentile)	7377	Rosoff DB, Mavromatis LA, Bell AS, et al. Multivariate
	PhenoAge acceleration	22209	Rosoff DB, Mavromatis LA, Bell AS, et al. Multivariate
	Hannum age acceleration	31412	Rosoff DB, Mavromatis LA, Bell AS, et al. Multivariate
	GrimAge acceleration	32418	Rosoff DB, Mavromatis LA, Bell AS, et al. Multivariate
	Estimate levels of Plasminogen Activation Inhibitor 1	18724	Rosoff DB, Mavromatis LA, Bell AS, et al. Multivariate
	Intrinsic epigenetic age acceleration	30685	Rosoff DB, Mavromatis LA, Bell AS, et al. Multivariate
	Estimated proportion of granulocytes	32423	Rosoff DB, Mavromatis LA, Bell AS, et al. Multivariate

Aging	mvAge	0.203	0.023	8.84	9.75E-19
	Healthspan (age the first incidence of seven specific diseases)	-0.202	0.032	-6.39	1.67E-10
	Parental lifespan (father's and mother's survival)	0.185	0.031	6.07	1.26E-09
	Frailty	-0.130	0.026	-5.07	3.90E-07
	Longevity (surviving at or beyond the age corresponding to the 90th/99th survival percentile)	0.141	0.047	3.00	2.70E-03
	PhenoAge acceleration	-0.102	0.048	-2.12	3.36E-02
	Hannum age acceleration	-0.104	0.051	-2.05	4.02E-02
	GrimAge acceleration	-0.078	0.050	-1.55	1.22E-01
	Estimate levels of Plasminogen Activation Inhibitor 1	0.056	0.118	0.47	6.36E-01
	Intrinsic epigenetic age acceleration	-0.017	0.036	-0.47	6.39E-01
	Estimated proportion of granulocytes	-0.014	0.056	-0.25	8.04E-01

Supplementary note, line 642-662: The healthspan GWAS consists of 300,477 unrelated, British-ancestry individuals from UK Biobank. The statistics were calculated by fitting Cox-Gompertz survival models with events defined as the first incidence of one of seven specific diseases (any cancer, diabetes, myocardial infarction, stroke, chronic obstructive pulmonary disease, dementia, and congestive heart failure) or death itself. As such, healthspan is highly dependent on the characteristics of the UK Biobank cohort, who were aged 40-69 years when they were recruited in 2006-2010 and of which two-thirds have yet to experience an age-related disease. Therefore, loci of healthspan GWAS have overrepresented effects on diseases of middle age (cancer, heart disease, etc) than age-related disease, which is a limitation acknowledged in the original study.

The parental lifespan GWAS consists of unrelated, European-ancestry individuals reporting a total of 512,047 mothers' and 500,193 fathers' lifespans. The statistics for each participating cohort were calculated by fitting Cox survival models to father's and mother's survival separately, adjusted for subject sex, at least 10 principal components, and study-specific covariates such as genotyping batch and array. Therefore, loci of parental lifespan GWAS don't have overrepresented effects on diseases of middle age.

The longevity GWAS included 11,262/3484 cases surviving at or beyond the age corresponding to the 90th/99th survival percentile, respectively, and 25,483 controls whose age at death or at last contact was at or below the age corresponding to the 60th survival percentile.

```

-----
Total Observed scale h2: 0.0229 (0.0015)
Lambda CC: 1.3
Mean Chi^2: 1.3405
Intercept: 1.0536 (0.0123)
Ratio: 0.1576 (0.0562)

Genetic Covariance
-----
Total Observed scale gencov: -0.018 (0.0012)
Mean z1+z2: -0.1989
Intercept: -0.0456 (0.006)

Genetic Correlation
-----
Genetic Correlation: -0.6959 (0.046)
Z-score: -15.1453
P: 6.1343e-52

Summary of Genetic Correlation Results
p1 p2 rg se z p h2_obs h2_obs_se h2_int h2_int_se gcov_int gcov_int_se
./Ageing_OMAS/healthspan_summary.sumstats.gz ./Ageing_OMAS/parental_lifespan.sumstats.gz -0.6959 0.046 -15.1453 0.1343e-52 0.0229 0.0015 1.0536 0.0123 -0.0456 0.006

Analysis finished at Tue Jul 30 11:31:28 2024

```

```

Lambda GC: 1.0649
Mean ChI*2: 1.0669
Intercept: 1.0076 (0.0079)
Ratio: 0.114 (0.1164)

Genetic Covariance
-----
Total Observed scale gencov: -0.0443 (0.0069)
Mean zI*2: -0.0589
Intercept: -0.008 (0.0049)

Genetic Correlation
-----
Genetic Correlation: -0.511 (0.0892)
Z-score: -5.7274
P: 1.0196e-08

Summary of Genetic Correlation Results
p1      p2      rg      se      z      p      h2_obs      h2_obs_se      h2_int      h2_int_se      gcov_int      gcov_int_se
/Ageing_OMAS/healthspan_summary.sumstats.gz  /Ageing_OMAS/longevity.sumstats.gz -0.511  0.0892 -5.7274  1.0196e-08  0.26  0.042  1.0076  0.0079 -0.008  0.0049

Analysis finished at Tue Jul 30 11:32:55 2024
Total time elapsed: 12.42s

```

```

-----
Total Observed scale h2: 0.2531 (0.0468)
Lambda GC: 1.0618
Mean ChI*2: 1.0667
Intercept: 1.0088 (0.0082)
Ratio: 0.1315 (0.1227)

Genetic Covariance
-----
Total Observed scale gencov: 0.0616 (0.0062)
Mean zI*2: 0.1191
Intercept: 0.0151 (0.0088)

Genetic Correlation
-----
Genetic Correlation: 0.8072 (0.0843)
Z-score: 9.571
P: 1.0586e-21

Summary of Genetic Correlation Results
p1      p2      rg      se      z      p      h2_obs      h2_obs_se      h2_int      h2_int_se      gcov_int      gcov_int_se
/Ageing_OMAS/parental_lifespan.sumstats.gz  /Ageing_OMAS/longevity.sumstats.gz  0.8072  0.0843  9.571  1.0586e-21  0.2531  0.0468  1.0088  0.0082  0.0151  0.0088

```

(17) Zenin A, Tsepilov Y, Sharapov S, et al. Identification of 12 genetic loci associated with human healthspan. *Commun Biol.* 2019;2:41. doi: 10.1038/s42003-019-0290-0

(18) Timmers PR, Mounier N, Lall K, et al. Genomics of 1 million parent lifespans implicates novel pathways and common diseases and distinguishes survival chances. *Elife.* 2019;8:e39856. doi:10.7554/eLife.39856

5. "Early puberty timing causally increases alanine aminotransferase and HbA1c, decreases HDL-C, and is associated with higher risk of aging, with a mediated proportion of 5.25%, 5.25%, and 9%, respectively" - they report that these biomarkers show strong genetic correlations, so it is important to clarify if these mediation estimates are independent of each other? i.e. Do they together explain 19.5% of the association between puberty and aging?

Thank you for pointing out this issue.

In our mediation analysis, firstly, we used MR-BMA to rank mediators (including HDL-C, HbA1c, alanine aminotransferase, and ApoA) for aging. MR-BMA is a two-sample multivariate MR approach to identify true causal risk factors despite high correlations of candidate factors, and can be used to determine which out of a set of related risk factors with common genetic predictors are the causal drivers of disease risk (19). The results indicated alanine aminotransferase, HbA1c, and HDL-C are the causal drivers of aging (Table 2).

And then, we used two-step MR to calculate mediating effect of three mediators. In brief, we estimated the mediation proportion by dividing the indirect effect with the total effect ($\beta_1 \times \beta_2 / \beta_3$) (20), where β_1 is the effect of mvPuberty on mediators, β_2 is the effect of mediators on aging, and β_3 is the effect of mvPuberty on aging. Finally, the calculated the mediated proportion for alanine aminotransferase, HbA1c, and HDL-C is 5.25%, 5.25%, and 9%, respectively.

Here, because we used MR-BMA to rank true causal risk factors for aging, the effect of alanine aminotransferase, HbA1c, and HDL-C on aging are not mediated by each other according to the principle of MR-BMA (for instance, if HDL-C mediated the effect of alanine aminotransferase on aging, MR-BMA would not have identified alanine aminotransferase as a causal risk factor for aging). Inspired by your comments, we used another multivariate MR in the TwoSampleMR package to detect the casual effect of alanine aminotransferase, HbA1c, and HDL-C on aging. As a result, after correcting for their mutual influence, the casual effects still present,

while the effect sizes are different (see table below). Accordingly, while the effects of alanine aminotransferase, HbA1c, and HDL-C on aging are not mediated by each other, we tend to speculate that their effect on aging are not entirely independent or additive of each other.

Meanwhile, to figure out whether pubertal timing affects alanine aminotransferase, HbA1c, and HDL-C, independently, we used multivariate MR in the TwoSampleMR package to detect the casual effect of mvPuberty on alanine aminotransferase, HbA1c, and HDL-C, respectively. After correcting for their mutual influence, the casual effect of mvPuberty on alanine aminotransferase, HbA1c, and HDL-C, changed (see table below), suggesting that the effect of puberty timing on alanine aminotransferase, HbA1c, and HDL-C is not independent of each other.

In addition, we also conducted multivariate MR to detect the casual effect of mvPuberty on aging after correcting influence of alanine aminotransferase, HbA1c, and HDL-C. As a result, causal effects are still present, but the value of the effect has become smaller (below table). These results indicated alanine aminotransferase, HbA1c, and HDL-C does have mediation effect between mvPuberty and aging.

In summary, mediation effect between mvPuberty and aging for alanine aminotransferase, HbA1c, and HDL-C, does exist, but appears to be non-independent of each other, which means that they can't altogether explain 19.5% of the association between puberty and aging. Based on your advice, we've added relevant descriptions in order to clarify the independence of the mediating effect in the Methods and Results section.

Results section, line 220-223: On the basis of the above analyses, we found that early pubertal timing causally increased alanine aminotransferase and HbA1c levels, decreases HDL-C levels, and is associated with greater ageing, with mediated proportions of 5.25%, 5.25%, and 9%, respectively.

Methods section, line 635-639: Because these mediators appear to be independent of each other, we estimated the mediation proportion separately by dividing the indirect effect by the total effect ($\beta_1 \times \beta_2 / \beta_3$)⁶⁵, where β_1 is the effect of mvPuberty on mediators, β_2 is the effect of mediators on adult phenotypes, and β_3 is the effect of mvPuberty on adult phenotypes.

	Exposure	Outcome	nsnp	b	se	pval
After correcting	HbA1c	Aging	129	-0.028	0.005	7.46E-08
	Alanine aminotransferase	Aging	51	-0.074	0.012	4.42E-10
	HDL-C	Aging	150	0.028	0.006	2.42E-06
Before correcting	HbA1c	Aging	217	-0.035	0.005	2.19E-14
	Alanine aminotransferase	Aging	138	-0.047	0.008	1.41E-09
	HDL-C	Aging	269	0.042	0.006	6.89E-14
After correcting	mvPuberty	Alanine aminotransferase	53	-0.123	0.052	1.75E-02
Before correcting	mvPuberty	Alanine aminotransferase	139	-0.093	0.025	2.09E-04
After correcting	mvPuberty	HbA1c	73	-0.098	0.043	2.19E-02
Before correcting	mvPuberty	HbA1c	139	-0.114	0.031	2.66E-04
After correcting	mvPuberty	HDL-C	60	0.030	0.068	6.60E-01
Before correcting	mvPuberty	HDL-C	139	0.104	0.024	1.79E-05
After correcting	mvPuberty	Aging	47	0.028	0.014	4.84E-02
Before correcting	mvPuberty	Aging	135	0.043	0.010	7.45E-06

- (19) Zuber V, Colijn JM, Klaver C, Burgess S. Selecting likely causal risk factors from high-throughput experiments using multivariable Mendelian randomization. *Nat Commun.* 2020;11(1):29. doi:10.1038/s41467-019-13870-3
- (20) Dai H, Hou T, Wang Q, et al. Causal relationships between the gut microbiome, blood lipids, and heart failure: a Mendelian randomization analysis. *Eur J Prev Cardiol.* 2023;30(12):1274-1282. doi:10.1093/eurjpc/zwad171

6. the MR with 25 dietary factors and micronutrients - describe the source of this data? Are these data also from UK Biobank and if so how were they derived?

Thanks for your helpful suggestion. These GWAS for dietary factors are sourced from UK Biobank. Information on the dietary factors was collected retrospectively using a dietary frequency questionnaire, which is publicly available in the UK Biobank (<https://biobank.ctsu.ox.ac.uk/crystal/label.cgi?id=100052>). Information on the micronutrients was collected by online 24-hour dietary recall questionnaire, which is publicly available in the UK Biobank (<https://biobank.ctsu.ox.ac.uk/crystal/label.cgi?id=100098>). Based on your advice, we've added relevant description in the Methods section.

Methods section, line 640-646: We performed MR with 25 dietary factors and micronutrients derived from a GWAS with participants of European ancestry in the UKB datasets to investigate whether mvPuberty status is causally influenced by dietary factors and micronutrients represented by SNPs (Supplementary Table 30). Information on the dietary factors and micronutrients that were collected and processed is publicly available on the UK Biobank website (<https://biobank.ctsu.ox.ac.uk/crystal/label.cgi?id=100052> and <https://biobank.ctsu.ox.ac.uk/crystal/label.cgi?id=100098>).

7. They report that claim that "Controlling oil fish and retinol intake is beneficial in promoting healthy pubertal development" - while they show that higher intakes is associated with earlier puberty, it would be interesting to also test if this might lead to healthier adult outcomes.

Thanks for your helpful suggestion. Based on your advice, we've conducted MR analysis for oil fish and retinol intake on adult outcomes. We found that oil fish intake is positive with aging ($P < 0.05$), indicating higher oil fish intake is associated with earlier puberty and aging.

outcome	exposure	method	nsnp	b	se	pval
Coronary_artery_disease	retinol	Inverse variance weighted	19	-0.034990106	0.062071364	0.572953078
Coronary_artery_disease	retinol	MRlap	20	-0.020676708	0.02578378	0.422595213
Coronary_artery_disease	oil fish	Inverse variance weighted	51	-0.072814971	0.127900253	0.569145367
Coronary_artery_disease	oil fish	MRlap	62	-0.016811129	0.050625141	0.739835825
ECAC	retinol	Inverse variance weighted	21	-0.212168049	0.135617123	0.117708741
ECAC	retinol	MRlap	22	-0.049070327	0.037838278	0.194685022
ECAC	oil fish	Inverse variance weighted	51	0.217989866	0.187978813	0.246190824
ECAC	oil fish	MRlap	70	0.060668449	0.055407933	0.273542241
FN_BMD	retinol	Inverse variance weighted	4	-0.233344187	0.164854616	0.156935317
FN_BMD	retinol	MRlap	4	-0.291157625	0.213133495	0.171913532
FN_BMD	oil fish	Inverse variance weighted	22	0.124747017	0.151612851	0.410621873
FN_BMD	oil fish	MRlap	33	0.069561723	0.144727427	0.630772713
Heart_failure	retinol	Inverse variance weighted	20	0.053760774	0.080532119	0.50440853
Heart_failure	retinol	MRlap	21	0.008751043	0.016310641	0.59159678

Heart_failure	oil fish	Inverse variance weighted	51	-0.113652043	0.12498117	0.363163647
Heart_failure	oil fish	MRlap	62	-0.009947155	0.025134822	0.692288017
heel_BMD	retinol	Inverse variance weighted	21	-0.03265751	0.022548043	0.147518541
heel_BMD	retinol	MRlap	22	-0.039124605	0.029201177	0.180300715
heel_BMD	oil fish	Inverse variance weighted	50	0.0644389	0.066166215	0.330109024
heel_BMD	oil fish	MRlap	61	0.163531626	0.094404381	0.083229743
LS_BMD	retinol	Inverse variance weighted	4	-0.236361232	0.213461474	0.268173675
LS_BMD	retinol	MRlap	4	-0.30633416	0.279684837	0.273392496
LS_BMD	oil fish	Inverse variance weighted	22	-0.033722795	0.165746646	0.838775589
LS_BMD	oil fish	MRlap	33	-0.40448676	0.220820945	0.06699003
mvAge.summary.EUR	retinol	Inverse variance weighted	19	0.009262177	0.008669478	0.285355391
mvAge.summary.EUR	retinol	MRlap	20	0.009900567	0.010045474	0.324341781
mvAge.summary.EUR	oil fish	Inverse variance weighted	48	0.057113472	0.02168535	0.00844513
mvAge.summary.EUR	oil fish	MRlap	58	0.057718059	0.024372777	0.017877946
parental_lifespan	retinol	Inverse variance weighted	21	0.028808755	0.033352968	0.387723119
parental_lifespan	retinol	MRlap	22	0.010135228	0.017063252	0.552525599
parental_lifespan	oil fish	Inverse variance weighted	48	0.120147609	0.066456803	0.070621234
parental_lifespan	oil fish	MRlap	60	0.043818197	0.033005673	0.184311626
Total_BMD	retinol	Inverse variance weighted	21	-0.126644831	0.062970187	0.044305693
Total_BMD	retinol	MRlap	22	-0.121765028	0.062949823	0.053073937
Total_BMD	oil fish	Inverse variance weighted	50	0.100094048	0.087404313	0.252132817
Total_BMD	oil fish	MRlap	62	0.127252183	0.088394402	0.149981873

8. Other downstream analyses seem to be conducted because they can rather than because of any well rationalised question and without meaningful interpretation - e.g. Brain imputation links 37 brain features to puberty timing. What do we learn from this?

Thank you for pointing out this issue. We've re-written related discussion to help readers understand the meaning for our downstream analyses.

Discussion section, line 322-354: Notably, earlier puberty has lasting impacts on brain maturation, including strengthening of neural connections and organizing and fostering the acquisition of cognitive abilities and social behaviours, partly due to changes in pubertal hormones⁴⁵⁻⁴⁷. In our BrainXcan analysis, we found changes in the mean diffusivity and radial diffusivity of several white matter microstructures, including the superior corona radiata, superior longitudinal fasciculus, external capsule, and uncinata fasciculus, which affected emotional development⁴⁸⁻⁵⁰. These white matter microstructures were identified as possible subsequent neural signatures for advanced pubertal status due to abnormal changes in gonadal sex steroid hormones⁴⁸⁻⁵⁰. These findings indicate that adult emotional disorders may be attributed to changes in white matter microstructures caused by abnormal gonadal sex steroid hormones during puberty. In addition, we also detected changes in the mean diffusivity and radial diffusivity in two other white matter microstructures, the inferior cerebellar peduncle and cingulum cingulate gyrus, through BrainXcan analysis. The inferior cerebellar peduncle, which connects the cerebellum and other parts of the nervous system, plays an important role in locomotor and mental adaptation, age and brain lateralization and fatigue⁵¹⁻⁵³. The cingulate gyrus, which connects the cingulate cortex with other brain regions, has also been associated with language, specifically phonological processing, including visuospatial episodic memory and spatial word memory^{54,55}. Previous studies have

shown that gonadal sex steroid hormones regulate neurogenesis and neuronal survival in the inferior cerebellar peduncle and cingulum cingulate gyrus. For example, oestradiol and progesterone facilitate oligodendrocyte activity and stimulate the proliferation of Schwann cells that produce myelin proteins, thereby modulating neurogenesis and neuronal survival through regulating each other's synthesis^{56,57}. We found for the first time that the inferior cerebellar peduncle and cingulum cingulate gyrus are associated with pubertal timing, which could be attributed to the interaction between gonadal sex steroid hormones and neurogenesis and neuronal survival⁴⁸. In summary, when the body secretes pubertal hormones, specifically gonadal hormones, at an earlier age and at higher levels, the brain may experience accelerated proliferation and subsequent fine-tuning of white matter development during puberty. Future work incorporating longitudinal neuroimaging in parallel with pubertal measures may contribute to the understanding of individual variation in pubertal course and brain region development.

References

- (1) Karlsson Linnér R, Mallard TT, Barr PB, et al. Multivariate analysis of 1.5 million people identifies genetic associations with traits related to self-regulation and addiction. *Nat Neurosci.* 2021;24(10):1367-1376. doi:10.1038/s41593-021-00908-3
- (2) Bulik-Sullivan BK, Loh PR, Finucane HK, et al. LD Score regression distinguishes confounding from polygenicity in genome-wide association studies. *Nat Genet.* 2015;47(3):291-295. doi:10.1038/ng.3211
- (3) Rosoff DB, Mavromatis LA, Bell AS, et al. Multivariate genome-wide analysis of aging-related traits identifies novel loci and new drug targets for healthy aging. *Nat Aging.* 2023;3(8):1020-1035. doi:10.1038/s43587-023-00455-5
- (4) MacArthur J, Bowler E, Cerezo M, et al. The new NHGRI-EBI Catalog of published genome-wide association studies (GWAS Catalog). *Nucleic Acids Res.* 2017;45(D1): D896-D901. doi:10.1093/nar/gkw1133
- (5) Feng H, Mancuso N, Gusev A, et al. Leveraging expression from multiple tissues using sparse canonical correlation analysis and aggregate tests improves the power of transcriptome-wide association studies. *PLoS Genet.* 2021;17(4): e1008973. doi:10.1371/journal.pgen.1008973
- (6) Hu Y, Li M, Lu Q, et al. A statistical framework for cross-tissue transcriptome-wide association analysis. *Nat Genet.* 2019;51(3):568-576. doi:10.1038/s41588-019-0345-7
- (7) Zampieri TT, Bohlen TM, Silveira MA, et al. Postnatal Overnutrition Induces Changes in Synaptic Transmission to Leptin Receptor-Expressing Neurons in the Arcuate Nucleus of Female Mice. *Nutrients.* 2020;12(8):2425. doi: 10.3390/nu12082425
- (8) Uellendahl-Werth F, Maj C, Borisov O, et al. Cross-tissue transcriptome-wide association studies identify susceptibility genes shared between schizophrenia and inflammatory bowel disease. *Commun Biol.* 2022;5(1):80. doi: 10.1038/s42003-022-03031-6
- (9) Sun BB, Maranville JC, Peters JE, et al. Genomic atlas of the human plasma proteome. *Nature.* 2018;558(7708):73-79. doi:10.1038/s41586-018-0175-2
- (10) The Atherosclerosis Risk in Communities (ARIC) Study: design and objectives. The ARIC investigators. *Am J Epidemiol.* 1989;129(4):687-702.
- (11) Evans DM, Davey Smith G. Mendelian Randomization: New Applications in the Coming Age of Hypothesis-Free Causality. *Annu Rev Genomics Hum Genet.* 2015;16: 327-350. doi:10.1146/annurev-genom-090314-050016
- (12) Cole JB, Florez JC, Hirschhorn JN. Comprehensive genomic analysis of dietary habits in UK Biobank identifies hundreds of genetic associations. *Nat Commun.* 2020;11(1):1467. doi:10.1038/s41467-020-15193-0
- (13) May-Wilson S, Matoba N, Wade KH, et al. Large-scale GWAS of food liking reveals genetic determinants and genetic correlations with distinct neurophysiological traits. *Nat Commun.* 2022;13(1):2743. doi:10.1038/s41467-022-30187-w
- (14) Mägi R, Morris AP. GWAMA: software for genome-wide association meta-analysis. *BMC Bioinformatics.* 2010;11:288. doi:10.1186/1471-2105-11-288
- (15) Hemani G, Tilling K, Davey Smith G. Orienting the causal relationship between imprecisely measured traits using GWAS summary data. *PLoS Genet.* 2017 Dec 29;13(12):e1007149. doi: 10.1371/journal.pgen.1007149.
- (16) Mancuso N, Freund MK, Johnson R, et al. Probabilistic fine-mapping of transcriptome-wide

- association studies. *Nat Genet.* 2019;51(4):675-682. doi:10.1038/s41588-019-0367-1
- (17) Zenin A, Tsepilov Y, Sharapov S, et al. Identification of 12 genetic loci associated with human healthspan. *Commun Biol.* 2019;2:41. doi: 10.1038/s42003-019-0290-0
- (18) Timmers PR, Mounier N, Lall K, et al. Genomics of 1 million parent lifespans implicates novel pathways and common diseases and distinguishes survival chances. *Elife.* 2019;8:e39856. doi:10.7554/eLife.39856
- (19) Zuber V, Colijn JM, Klaver C, Burgess S. Selecting likely causal risk factors from high-throughput experiments using multivariable Mendelian randomization. *Nat Commun.* 2020;11(1):29. doi:10.1038/s41467-019-13870-3
- (20) Dai H, Hou T, Wang Q, et al. Causal relationships between the gut microbiome, blood lipids, and heart failure: a Mendelian randomization analysis. *Eur J Prev Cardiol.* 2023;30(12):1274-1282. doi:10.1093/eurjpc/zwad171

REVIEWER COMMENTS

Reviewer #1 (Remarks to the Author):

My concerns have been addressed. Nice work!

Reviewer #2 (Remarks to the Author):

Some of my previous comments need to be addressed further:

1. QSNP heterogeneity - 27 of 266 SNPs show significant heterogeneity - what is the source of this? I understand the calculation and statistical interpretation of QSNP. I am asking for A) further understanding of what is driving this heterogeneity, e.g. Sex differences or strong association with one specific puberty feature - and which one? And B) Are these 27 SNPs still considered to be valid signals for puberty? i.e. Are they still included in downstream analyses of biological mechanisms and Mendelian randomisation?

2. SCOUTJOY - Add to the paper the sample size available for Tanner stage in boys and girls. It is a relatively small sample. While 'sample size' is not an input to SCOUTJOY, the low sample size is reflected in the standard error.

6. It would be helpful to readers to add to the Methods section the brief information given in the responses, i.e. "Information on the dietary factors was collected retrospectively using a dietary frequency questionnaire, which is publicly available in the UK Biobank (<https://biobank.ctsu.ox.ac.uk/crystal/label.cgi?id=100052>). Information on the micronutrients was collected by online 24-hour dietary recall questionnaire, which is publicly available in the UK Biobank (<https://biobank.ctsu.ox.ac.uk/crystal/label.cgi?id=100098>)."

REVIEWER COMMENTS

Reviewer #1 (Remarks to the Author):

My concerns have been addressed. Nice work!

We are grateful for the reviewer's constructive comments that truly make our work better.

Reviewer #2 (Remarks to the Author):

Some of my previous comments need to be addressed further:

1. QSNP heterogeneity - 27 of 266 SNPs show significant heterogeneity - what is the source of this? I understand the calculation and statistical interpretation of QSNP. I am asking for A) further understanding of what is driving this heterogeneity, e.g. Sex differences or strong association with one specific puberty feature - and which one? And B) Are these 27 SNPs still considered to be valid signals for puberty? i.e. Are they still included in downstream analyses of biological mechanisms and Mendelian randomisation?

Thank you for pointing out this issue. To further understand what is driving this heterogeneity, we extracted BETA, and P value of 27 SNPs from four input specific puberty features (table below).

Firstly, we evaluated the P value for 27 SNPs to determine whether their heterogeneity is driven by strong association with one specific puberty feature.

We found 13 of the 27 SNPs (rs466639, rs4672110, rs12878738, rs11867780, rs1079866, rs4754779, rs12607903, rs997295, rs2894889, rs2184968, rs2129464, rs852069, rs13196561) demonstrated genome-wide significance ($P < 5 \times 10^{-8}$) only for age of menarche, suggesting that the heterogeneity for these 13 SNPs is driven by strong association with age of menarche.

We found four of the 27 SNPs (rs6914292, rs6568401, rs11121667 and rs7781162) showed genome-wide significance ($P < 5 \times 10^{-8}$) only for age at first facial hair, suggesting the heterogeneity for these four SNPs is driven by strong association with age at first facial hair.

We found ten of the 27 SNPs showed genome-wide significance ($P < 5 \times 10^{-8}$) for multiple puberty features: eight (rs17426174, rs199528, rs16832681, rs3824915, rs10954315, rs9907841, rs12922197, rs12913832) showed genome-wide significance for age at first facial hair and age at voice break, one (rs2167733) showed genome-wide significance for age of menarche and pubertal height spurt, and one (rs1461503) showed genome-wide significance for age of menarche, age at first facial hair and age at voice break, suggesting that heterogeneity for these ten SNPs is driven by strong association with multiple but not by all four input puberty features.

Additionally, we evaluated the direction of effect (BETA) for 27 SNPs to determine whether their heterogeneity is driven by sex differences. We found three of the 27 SNPs (rs12878738, rs2184968 and rs852069) showed inconsistent direction of effect (BETA) between female (age of menarche) and male (age at first facial hair) puberty features, and demonstrated genome-wide significance ($P < 5 \times 10^{-8}$) only for female puberty feature (age of menarche). We found two of the 27 SNPs (rs12922197 and rs12913832) showed inconsistent direction of effect (BETA) between female (age of menarche) and male (age at voice break and age at first facial hair) puberty features, and demonstrated genome-wide significance ($P < 5 \times 10^{-8}$) only for male puberty features (age at first facial hair and age at voice break). These results suggested that heterogeneity for these five

SNPs is driven by sex differences or/and strong association with one specific puberty feature.

In our downstream analysis, we considered these 27 SNPs to be valid signals due to their correlation with one or more puberty features, which is in line with previous studies that use similar multivariate genomic SEM approach to analyze traits related to aging, self-regulation and addiction [1,2]. We believe downstream analysis using SNPs to reflect more comprehensive genetic architecture of puberty would be helpful to identify potential biological mechanisms. Inspired by your valuable suggestion, we re-performed downstream Mendelian randomisation analysis using instrument variants excluding these 27 SNPs. The results showed that the causal estimates are generally consistent with the causal estimates without removing the 27 SNPs (the positive and negative BETA values are consistent and the causal estimates all pass multiple tests for statistical significance; see supplementary table 1-3 at the end of this response letter).

We have added relevant description in the Results and Supplementary tables according to your advice.

Results section, line 102-103: The sources of the 27 lead SNPs with heterogeneity are shown in Supplementary Table 32.

[1] Rosoff, D. B., Mavromatis, L. A., Bell, A. S., Wagner, J., Jung, J., Marioni, R. E., Davey Smith, G., Horvath, S., & Lohoff, F. W. (2023). Multivariate genome-wide analysis of aging-related traits identifies novel loci and new drug targets for healthy aging. *Nature aging*, 3(8), 1020–1035. <https://doi.org/10.1038/s43587-023-00455-5>

[2] Karlsson Linnér, R., Mallard, T. T., Barr, P. B., Sanchez-Roige, S., Madole, J. W., Driver, M. N., Poore, H. E., de Vlaming, R., Grotzinger, A. D., Tielbeek, J. J., Johnson, E. C., Liu, M., Rosenthal, S. B., Ideker, T., Zhou, H., Kember, R. L., Pasman, J. A., Verweij, K. J. H., Liu, D. J., Vrieze, S., ... Dick, D. M. (2021). Multivariate analysis of 1.5 million people identifies genetic associations with traits related to self-regulation and addiction. *Nature neuroscience*, 24(10), 1367–1376. <https://doi.org/10.1038/s41593-021-00908-3>

SNP	BETA.menarche	P.menarche	BETA.hair	P.hair	BETA.voice	P.voice	BETA.PHS	P.PHS
rs1461503 ^d	0.060	5.48E-45	0.011	2.72E-12	0.008	2.26E-12	0.026	4.03E-02
rs466639 [#]	0.081	1.05E-36	0.006	1.41E-02	0.001	5.06E-01	-0.018	3.28E-01
rs4672110 [#]	0.070	5.34E-35	0.004	3.04E-02	0.003	5.19E-02	-0.011	4.91E-01
rs12878738 ^{**}	-0.053	4.40E-31	0.001	5.43E-01	-0.001	2.47E-01	0.036	1.06E-02
rs11867780 [#]	0.055	4.98E-31	0.000	9.86E-01	0.000	7.01E-01	0.003	8.14E-01
rs1079866 [#]	0.072	7.02E-31	-0.001	5.66E-01	0.002	1.78E-01	0.035	4.19E-02
rs4754779 [#]	0.047	7.79E-26	0.003	3.57E-02	0.001	3.22E-01	-0.001	9.37E-01
rs12607903 [#]	-0.049	2.52E-25	-0.006	7.72E-04	-0.005	4.08E-05	0.029	3.22E-02
rs997295 [#]	-0.042	8.16E-23	-0.002	2.93E-01	-0.002	3.43E-02	-0.007	5.61E-01
rs2894889 [#]	0.042	6.12E-21	0.005	2.82E-03	0.003	3.07E-02	0.013	3.06E-01
rs2167733 ^c	-0.043	7.36E-20	-0.007	4.08E-05	-0.005	9.91E-05	-0.086	8.95E-10
rs2184968 ^{**}	-0.038	8.40E-19	0.002	2.12E-01	-0.001	3.31E-01	0.062	4.25E-07
rs2129464 [#]	-0.039	2.56E-16	-0.001	5.68E-01	-0.001	3.52E-01	0.036	8.89E-03
rs852069 ^{**}	0.035	8.58E-16	-0.001	6.36E-01	0.001	4.49E-01	-0.045	4.74E-04
rs13196561 [#]	-0.042	1.04E-15	-0.004	5.98E-02	-0.001	7.14E-01	-0.033	3.45E-02
rs17426174 ^b	0.020	4.19E-04	0.021	2.76E-29	0.010	3.36E-13	-0.009	5.72E-01
rs199528 ^b	0.016	1.57E-03	0.019	1.07E-24	0.010	7.57E-12	0.004	8.26E-01
rs6914292 ^a	0.013	1.67E-03	0.013	5.10E-18	0.003	7.55E-03	-0.013	3.01E-01

rs16832681 ^b	-0.012	3.67E-02	-0.026	3.38E-46	-0.007	3.95E-08	-0.037	2.13E-02
rs6568401 ^a	-0.007	1.43E-01	-0.014	6.68E-16	-0.004	1.04E-03	0.023	9.81E-02
rs11121667 ^a	0.008	1.61E-01	0.025	2.91E-40	0.005	6.41E-04	-0.009	5.52E-01
rs3824915 ^b	0.006	1.73E-01	0.017	2.54E-28	0.007	3.20E-10	0.007	5.79E-01
rs7781162 ^a	0.003	5.87E-01	0.018	1.56E-15	0.006	1.69E-04	-0.032	8.26E-02
rs10954315 ^b	0.002	5.92E-01	0.023	3.54E-48	0.006	2.47E-08	0.012	3.42E-01
rs9907841 ^b	-0.002	7.31E-01	-0.017	8.70E-30	-0.007	9.68E-09	0.002	8.92E-01
rs12922197 ^{*b}	-0.002	7.59E-01	0.018	8.17E-14	0.010	1.67E-07	-0.042	2.66E-02
rs12913832 ^{*b}	-0.002	7.87E-01	0.034	1.62E-73	0.011	1.06E-14	-0.016	3.59E-01

Note: * SNPs with inconsistent effect (BETA) between female puberty feature and male puberty features; # SNPs with strong association for age of menarche; ^a SNPs with strong association for age at first facial hair; ^b SNPs with strong association for age at first facial hair and age at voice break; ^c SNPs with strong association for age of menarche and pubertal height spurt; ^d SNPs with strong association for three input univariate GWASs except for pubertal height spurt.

2. SCOUTJOY - Add to the paper the sample size available for Tanner stage in boys and girls. It is a relatively small sample. While 'sample size' is not an input to SCOUTJOY, the low sample size is reflected in the standard error.

Thank you for your helpful suggestion. We have added the sample size for Tanner stage in boys and girls in the Methods, and the limitation of the Discussion according to your advice.

Methods section, line 456-459: We tested each of the lead SNPs from mvPuberty for differences in Tanner stage effect sizes between males and females via the values of effect sizes (beta) and standard errors (se) in the Tanner stage GWAS, including data for 3,769 boys and 6,147 girls.

Discussion section, line 380-382: While “sample size” is not an input to SCOUTJOY analysis, the low sample size of Tanner stage GWAS is reflected in the standard error, therefore, our results of SCOUTJOY analysis should be interpreted with caution.

6. It would be helpful to readers to add to the Methods section the brief information given in the responses, i.e. "Information on the dietary factors was collected retrospectively using a dietary frequency questionnaire, which is publicly available in the UK Biobank (<https://biobank.ctsu.ox.ac.uk/crystal/label.cgi?id=100052>). Information on the micronutrients was collected by online 24-hour dietary recall questionnaire, which is publicly available in the UK Biobank (<https://biobank.ctsu.ox.ac.uk/crystal/label.cgi?id=100098>)."

Thank you for your helpful recommendation. We have added the brief information in the Methods section according to your advice.

Methods section, line 581-587: Information on the dietary factors was collected retrospectively using a dietary frequency questionnaire, which is publicly available in the UK Biobank (<https://biobank.ctsu.ox.ac.uk/crystal/label.cgi?id=100052>). Information on the micronutrients was collected by online 24-hour dietary recall questionnaire, which is publicly available in the UK Biobank (<https://biobank.ctsu.ox.ac.uk/crystal/label.cgi?id=100098>).

Supplementary Table 1. Mendelian randomization analysis investigating the causal role of mvPuberty on multi-traits using IVW after excluding 27 SNPs.

outcome	exposure	method	nsnp	b	se	pval
heel_BMD	mvPuberty	Inverse variance weighted	129	-0.175607245	0.030318491	6.95E-09
ECAC	mvPuberty	Inverse variance weighted	129	-0.54461096	0.096269204	1.54E-08
Heart_failure	mvPuberty	Inverse variance weighted	129	-0.321786908	0.061543783	1.71E-07
mvAge.summary.EUR	mvPuberty	Inverse variance weighted	126	0.050226764	0.010253676	9.66E-07
MetS	mvPuberty	Inverse variance weighted	127	-0.223279703	0.045700276	1.03E-06
LS_BMD	mvPuberty	Inverse variance weighted	129	-0.270507054	0.058751552	4.14E-06
parental_lifespan	mvPuberty	Inverse variance weighted	126	0.15033514	0.034938552	1.69E-05
Coronary_artery_disease	mvPuberty	Inverse variance weighted	129	-0.234331448	0.060555732	0.000108977
Total_BMD	mvPuberty	Inverse variance weighted	129	-0.175539334	0.04921424	0.000361312
ml_volume.abdominal_subcutaneous_fat	mvPuberty	Inverse variance weighted	129	-0.22099831	0.062426153	0.000399891
FN_BMD	mvPuberty	Inverse variance weighted	129	-0.195121351	0.056309008	0.000529859
Ovarian cancer	mvPuberty	Inverse variance weighted	129	-0.315463538	0.091792254	0.000588821
3_PGM	mvPuberty	Inverse variance weighted	129	-0.138547911	0.042298114	0.001054712
Frailty	mvPuberty	Inverse variance weighted	129	-0.084805671	0.027508156	0.002049746
Diverticulosis	mvPuberty	Inverse variance weighted	129	0.220552647	0.073947213	0.002858404
longevity	mvPuberty	Inverse variance weighted	129	0.379519178	0.127618534	0.002940828
2_GORD	mvPuberty	Inverse variance weighted	129	-0.130403415	0.044962458	0.003728375
GrimAge_EUR_summary_statistics	mvPuberty	Inverse variance weighted	129	-0.491804934	0.180775325	0.00651771
healthspan_summary	mvPuberty	Inverse variance weighted	129	-0.09978576	0.037148261	0.007228087
Pancreatic cancer	mvPuberty	Inverse variance weighted	129	-0.528285577	0.21087845	0.012239384
Attention deficit hyperactivity disorder	mvPuberty	Inverse variance weighted	128	-0.211967698	0.089333671	0.017655552
Esophagus_cancer	mvPuberty	Inverse variance weighted	129	-0.686594276	0.299785089	0.022004611
ml_volume.kidney_average	mvPuberty	Inverse variance weighted	129	-0.124668457	0.055796951	0.025461775
Gastritis_duodenitis	mvPuberty	Inverse variance weighted	129	0.117927054	0.053423408	0.027285834

Obsessive-compulsive disorder	mvPuberty	Inverse variance weighted	129	0.522212871	0.238318017	0.028434157
cervical cancer	mvPuberty	Inverse variance weighted	99	0.001953842	0.000897093	0.029408162
Autism spectrum disorder	mvPuberty	Inverse variance weighted	129	-0.202437404	0.095552214	0.034123859
5_IBD	mvPuberty	Inverse variance weighted	129	0.23058801	0.111215086	0.038139542
ml_volume.visceral_fat	mvPuberty	Inverse variance weighted	129	-0.096854584	0.047038451	0.039489324
Atrial_fibrillation	mvPuberty	Inverse variance weighted	129	0.142305725	0.069741026	0.041301748
Ischemic_stroke	mvPuberty	Inverse variance weighted	129	-0.104389103	0.051586469	0.043013871
ml_volume.pancreas	mvPuberty	Inverse variance weighted	129	0.10207069	0.053739371	0.057516477
ml_volume.spleen	mvPuberty	Inverse variance weighted	129	-0.098064017	0.051726986	0.057986505
ml_volume.liver	mvPuberty	Inverse variance weighted	129	-0.125863818	0.06805677	0.064400866
PhenoAge_EUR_summary_statistics	mvPuberty	Inverse variance weighted	129	-0.400465114	0.220742156	0.069650639
Skull_BMD	mvPuberty	Inverse variance weighted	129	-0.093886306	0.055483407	0.090617174
1_PUD	mvPuberty	Inverse variance weighted	129	-0.133040515	0.081500092	0.102595858
Fracture	mvPuberty	Inverse variance weighted	129	0.099900802	0.062444738	0.109636882
Depression	mvPuberty	Inverse variance weighted	101	-0.10288	0.06735037	0.126628203
PD_2019	mvPuberty	Inverse variance weighted	129	-0.194562601	0.130875514	0.137114305
Anorexia nervosa	mvPuberty	Inverse variance weighted	128	0.145304743	0.099735199	0.145142514
Insomnia	mvPuberty	Inverse variance weighted	129	-0.059298149	0.043013771	0.168022671
Lung cancer	mvPuberty	Inverse variance weighted	127	-0.137417965	0.110012564	0.211624274
pdff.liver	mvPuberty	Inverse variance weighted	129	-0.066104746	0.053979464	0.220715533
ALS_2021	mvPuberty	Inverse variance weighted	127	0.104677726	0.08689944	0.228363711
Acute_tubulo_interstitial_nephritis	mvPuberty	Inverse variance weighted	129	0.069212699	0.05886533	0.239682651
oropharynx cancer	mvPuberty	Inverse variance weighted	110	0.426444561	0.364808044	0.242421349
Non-Hodgkin_lymphoma	mvPuberty	Inverse variance weighted	129	-0.849480317	0.730377067	0.244800735
Hannum_EUR_summary_statistics	mvPuberty	Inverse variance weighted	129	-0.220693992	0.194047949	0.255405874
Bipolar disorder	mvPuberty	Inverse variance weighted	129	0.117430273	0.111542234	0.292438384

ml_volume.lungs	mvPuberty	Inverse variance weighted	129	0.051995594	0.04967789	0.295258894
iron.pancreas	mvPuberty	Inverse variance weighted	129	0.056730621	0.055814483	0.309432285
AD_Kunkle	mvPuberty	Inverse variance weighted	129	-0.090525226	0.092683133	0.328709091
Prostate cancer	mvPuberty	Inverse variance weighted	92	0.07371899	0.08387449	0.379444504
4_IBS	mvPuberty	Inverse variance weighted	129	-0.050216602	0.057180049	0.379824922
malignant melanoma	mvPuberty	Inverse variance weighted	121	0.001045918	0.001200297	0.383545122
Gastric cancer	mvPuberty	Inverse variance weighted	129	0.087341629	0.100759826	0.386035219
Anxiety	mvPuberty	Inverse variance weighted	129	-0.122324227	0.160413477	0.445728354
Colorectal cancer	mvPuberty	Inverse variance weighted	129	0.061970679	0.084539902	0.463537289
Laskar_31231134_GCST008225	mvPuberty	Inverse variance weighted	129	-0.193089166	0.26542474	0.466936633
IEAA_EUR_summary_statistics	mvPuberty	Inverse variance weighted	129	-0.124100691	0.185698498	0.503947665
Schizophrenia	mvPuberty	Inverse variance weighted	129	0.069195641	0.105008166	0.509924757
head and neck malignant neoplasia	mvPuberty	Inverse variance weighted	110	0.171220951	0.297347618	0.564731748
PAI1_EUR_summary_statistics	mvPuberty	Inverse variance weighted	129	-0.256343099	0.500222298	0.608330255
Gran_EUR_summary_statistics	mvPuberty	Inverse variance weighted	129	-0.001922283	0.004531196	0.67139591
iron.liver	mvPuberty	Inverse variance weighted	129	0.021764827	0.051784002	0.674266152
Thyroid cancer	mvPuberty	Inverse variance weighted	129	-0.079210135	0.224084365	0.723725967
LBD_2021_hg19	mvPuberty	Inverse variance weighted	121	0.06797603	0.231849312	0.769376498
Post-traumatic stress disorder	mvPuberty	Inverse variance weighted	129	-0.027553179	0.094761123	0.771231374
Laskar_prePMID_Males	mvPuberty	Inverse variance weighted	66	-0.082361416	0.287956388	0.774862457
Tourette syndrome	mvPuberty	Inverse variance weighted	129	-0.051290248	0.182666536	0.778874367
breast_cancer	mvPuberty	Inverse variance weighted	110	-0.015980992	0.063312045	0.800719305
oral cavity cancer	mvPuberty	Inverse variance weighted	110	-0.075151058	0.350591985	0.830270425
Bladder_cancer	mvPuberty	Inverse variance weighted	129	-0.061011117	0.314550627	0.846205019
Corpus_uteri_cancer	mvPuberty	Inverse variance weighted	129	-0.033644806	0.22081996	0.878900587
pdff.pancreas	mvPuberty	Inverse variance weighted	129	-0.004941618	0.060538829	0.934943137

Chronic_kidney_disease	mvPuberty	Inverse variance weighted	129	-0.006019613	0.092053003	0.947861172
Venous_thromboembolism	mvPuberty	Inverse variance weighted	129	-8.91E-05	0.002161515	0.967104135

Supplementary Table 2. Mendelian randomization analysis investigating the causal role of mvPuberty on serum biomarkers using IVW after excluding 27 SNPs.

outcome	exposure	method	nsnp	b	se	pval
HDL cholesterol	mvPuberty	Inverse variance weighted	129	0.116884053	0.026474132	1.01E-05
Apolipoprotein A	mvPuberty	Inverse variance weighted	129	0.11631745	0.028187026	3.68E-05
Glycated haemoglobin HbA1c	mvPuberty	Inverse variance weighted	129	-0.132075714	0.034262877	0.000115836
Alanine aminotransferase	mvPuberty	Inverse variance weighted	129	-0.09824714	0.027586703	0.00036889
Urinary sodium-potassium ratio	mvPuberty	Inverse variance weighted	129	-0.150477919	0.042372847	0.00038336
ukb-d-30720_irmt	mvPuberty	Inverse variance weighted	129	-0.12058754	0.036895088	0.001081612
ukb-d-30880_irmt	mvPuberty	Inverse variance weighted	129	-0.112775514	0.036982854	0.00229299
UKB-b-20175	mvPuberty	Inverse variance weighted	129	-0.065493574	0.023086324	0.004555399
ukb-d-30740_irmt	mvPuberty	Inverse variance weighted	129	-0.085905176	0.031561287	0.006491706
ieu-b-110	mvPuberty	Inverse variance weighted	129	0.072384567	0.030114631	0.016232994
ukb-d-30650_irmt	mvPuberty	Inverse variance weighted	129	-0.067069621	0.02828263	0.017720514
ieu-a-301	mvPuberty	Inverse variance weighted	125	0.112849193	0.051557162	0.028610143
ukb-d-30840_irmt	mvPuberty	Inverse variance weighted	129	0.056405834	0.027092475	0.037344826
ieu-b-34	mvPuberty	Inverse variance weighted	129	-0.0590461	0.031247858	0.058810841
ukb-d-30830_irmt	mvPuberty	Inverse variance weighted	129	0.08216387	0.044591423	0.065388372
ieu-b-30	mvPuberty	Inverse variance weighted	129	0.058960888	0.032111802	0.066340984
ukb-d-30890_irmt	mvPuberty	Inverse variance weighted	129	0.041008933	0.022889573	0.073197247
ieu-b-32	mvPuberty	Inverse variance weighted	129	0.048742949	0.027234918	0.073498509
ieu-b-108	mvPuberty	Inverse variance weighted	129	0.057456119	0.035262021	0.10322728

ukb-d-30640_irnt	mvPuberty	Inverse variance weighted	129	0.054770999	0.03624228	0.130725794
ukb-d-30660_irnt	mvPuberty	Inverse variance weighted	129	0.035407295	0.02502632	0.157126476
ukb-d-30680_irnt	mvPuberty	Inverse variance weighted	129	0.038635934	0.033594125	0.250110898
ukb-d-30810_irnt	mvPuberty	Inverse variance weighted	129	0.032881223	0.028688294	0.251731142
ukb-d-30610_irnt	mvPuberty	Inverse variance weighted	129	-0.077488597	0.069565951	0.265327738
ukb-d-30670_irnt	mvPuberty	Inverse variance weighted	129	-0.038490971	0.03459638	0.265892205
ukb-d-30770_irnt	mvPuberty	Inverse variance weighted	129	-0.066947726	0.068292466	0.326933565
ieu-b-31	mvPuberty	Inverse variance weighted	129	0.024628405	0.028068488	0.380247871
ieu-b-33	mvPuberty	Inverse variance weighted	129	-0.025778313	0.030125108	0.39215899
ieu-b-111	mvPuberty	Inverse variance weighted	129	-0.049975245	0.058634529	0.394037809
ukb-d-30860_irnt	mvPuberty	Inverse variance weighted	129	0.030903601	0.040247099	0.44257828
ukb-d-30790_irnt	mvPuberty	Inverse variance weighted	129	-0.01371551	0.017994521	0.445937788
ukb-d-30080_irnt	mvPuberty	Inverse variance weighted	129	0.022755408	0.034942294	0.514899172
ieu-a-1050	mvPuberty	Inverse variance weighted	122	-0.035995005	0.055810265	0.518957554
ukb-d-30700_irnt	mvPuberty	Inverse variance weighted	129	0.013032014	0.032125803	0.684995617
ieu-a-1052	mvPuberty	Inverse variance weighted	126	0.016458125	0.067057285	0.806120548
ukb-d-30730_irnt	mvPuberty	Inverse variance weighted	129	-0.006703557	0.048040508	0.889023704
ieu-b-29	mvPuberty	Inverse variance weighted	129	0.002578208	0.018517776	0.889269357
ieu-b-72	mvPuberty	Inverse variance weighted	129	-0.003035844	0.026697717	0.909466302
ukb-d-30710_irnt	mvPuberty	Inverse variance weighted	129	0.000377822	0.074991319	0.995980106

Supplementary Table 3. Mendelian randomization analysis investigating the causal role of mvPuberty on gut microbiota using IVW after excluding 27 SNPs.

outcome	exposure	n SNP	b	se	pval
k_Bacteria.p_Bacteroidetes.c_Bacteroidia.o_Bacteroidales.f_Porphyrimonadaceae.g_Parabacteroides.s_Parabacteroides_unclassified	mvPuberty	127	-0.47	0.13	4.17E-04
k_Bacteria.p_Bacteroidetes.c_Bacteroidia.o_Bacteroidales.f_Porphyrimonadaceae.g_Parabacteroides	mvPuberty	127	-0.27	0.10	8.55E-03
k_Bacteria.p_Bacteroidetes.c_Bacteroidia.o_Bacteroidales.f_Porphyrimonadaceae	mvPuberty	127	-0.24	0.09	1.06E-02
k_Bacteria.p_Firmicutes.c_Bacilli.o_Lactobacillales.f_Streptococcaceae.g_Streptococcus.s_Streptococcus_parasanguinis	mvPuberty	127	-0.51	0.21	1.39E-02
k_Bacteria.p_Firmicutes.c_Clostridia.o_Clostridiales.f_Oscillospiraceae.g_Oscillibacter.s_Oscillibacter_unclassified	mvPuberty	127	-0.26	0.11	1.65E-02
k_Bacteria.p_Firmicutes.c_Clostridia.o_Clostridiales.f_Oscillospiraceae	mvPuberty	127	-0.26	0.11	1.77E-02
k_Bacteria.p_Firmicutes.c_Clostridia.o_Clostridiales.f_Oscillospiraceae.g_Oscillibacter	mvPuberty	127	-0.26	0.11	1.79E-02
k_Bacteria.p_Proteobacteria.c_Deltaproteobacteria.o_Desulfovibrionales.f_Desulfovibrionaceae.g_Desulfovibrio.s_Desulfovibrio_piger	mvPuberty	109	-0.37	0.18	3.32E-02
k_Bacteria.p_Bacteroidetes.c_Bacteroidia.o_Bacteroidales.f_Bacteroidaceae.g_Bacteroides.s_Bacteroides_faecis	mvPuberty	117	-0.48	0.23	4.11E-02
k_Bacteria.p_Actinobacteria.c_Actinobacteria.o_Coriobacteriales.f_Coriobacteriaceae.g_Adlercreutzia	mvPuberty	126	-0.25	0.14	6.90E-02
k_Bacteria.p_Firmicutes.c_Clostridia.o_Clostridiales.f_Lachnospiraceae.g_Roseburia.s_Roseburia_inulinivorans	mvPuberty	127	-0.18	0.10	8.56E-02
k_Bacteria.p_Actinobacteria.c_Actinobacteria.o_Coriobacteriales.f_Coriobacteriaceae.g_Adlercreutzia.s_Adlercreutzia_equolifaciens	mvPuberty	127	-0.24	0.14	8.63E-02
k_Bacteria.p_Actinobacteria.c_Actinobacteria.o_Coriobacteriales.f_Coriobacteriaceae.g_Eggerthella.s_Eggerthella_unclassified	mvPuberty	118	0.34	0.20	8.63E-02
k_Bacteria.p_Proteobacteria.c_Deltaproteobacteria.o_Desulfovibrionales.f_Desulfovibrionaceae	mvPuberty	127	-0.17	0.10	8.67E-02
k_Bacteria.p_Proteobacteria.c_Deltaproteobacteria.o_Desulfovibrionales	mvPuberty	127	-0.17	0.10	8.69E-02
k_Bacteria.p_Proteobacteria.c_Deltaproteobacteria	mvPuberty	127	-0.17	0.10	8.69E-02
k_Bacteria.p_Proteobacteria.c_Gammaproteobacteria.o_Enterobacteriales	mvPuberty	127	0.18	0.11	9.72E-02
k_Bacteria.p_Proteobacteria.c_Gammaproteobacteria.o_Enterobacteriales.f_Enterobacteriaceae	mvPuberty	127	0.18	0.11	9.74E-02
k_Bacteria.p_Bacteroidetes.c_Bacteroidia.o_Bacteroidales.f_Bacteroidaceae.g_Bacteroides.s_Bacteroides_dorei	mvPuberty	127	-0.17	0.11	1.05E-01
k_Bacteria.p_Firmicutes.c_Clostridia.o_Clostridiales.f_Clostridiaceae.g_Clostridium	mvPuberty	127	0.21	0.13	1.25E-01
k_Bacteria.p_Bacteroidetes.c_Bacteroidia.o_Bacteroidales.f_Rikenellaceae.g_Alistipes.s_Alistipes_senegalensis	mvPuberty	123	0.17	0.11	1.32E-01
k_Bacteria.p_Proteobacteria.c_Gammaproteobacteria.o_Enterobacteriales.f_Enterobacteriaceae.g_Escherichia	mvPuberty	127	0.16	0.11	1.35E-01
k_Bacteria.p_Bacteroidetes.c_Bacteroidia.o_Bacteroidales.f_Bacteroidaceae.g_Bacteroides.s_Bacteroides_caccae	mvPuberty	127	-0.17	0.12	1.41E-01
k_Bacteria.p_Firmicutes.c_Erysipelotrichia.o_Erysipelotrichales.f_Erysipelotrichaceae.g_Holdemania.s_Holdemania_unclassified	mvPuberty	121	-0.27	0.18	1.41E-01

k__Bacteria.p__Bacteroidetes.c__Bacteroidia.o__Bacteroidales.f__Bacteroidaceae.g__Bacteroides.s__Bacteroides_coprocola	mvPuberty	127	0.31	0.21	1.42E-01
k__Bacteria.p__Firmicutes.c__Clostridia.o__Clostridiales.f__Lachnospiraceae.g__Lachnospiraceae_noname	mvPuberty	127	-0.14	0.09	1.43E-01
k__Bacteria.p__Firmicutes.c__Clostridia.o__Clostridiales.f__Eubacteriaceae.g__Eubacterium.s__Eubacterium_siraeum	mvPuberty	123	0.16	0.11	1.49E-01
k__Bacteria.p__Proteobacteria.c__Gammaproteobacteria	mvPuberty	118	0.15	0.10	1.51E-01
k__Bacteria.p__Actinobacteria.c__Actinobacteria.o__Coriobacteriales.f__Coriobacteriaceae.g__Eggerthella	mvPuberty	127	0.26	0.18	1.53E-01
k__Bacteria.p__Firmicutes.c__Clostridia.o__Clostridiales.f__Ruminococcaceae.g__Subdoligranulum	mvPuberty	127	0.13	0.10	1.57E-01
k__Bacteria.p__Firmicutes.c__Clostridia.o__Clostridiales.f__Lachnospiraceae.g__Dorea.s__Dorea_unclassified	mvPuberty	127	0.27	0.19	1.64E-01
k__Bacteria.p__Bacteroidetes.c__Bacteroidia.o__Bacteroidales.f__Porphyromonadaceae.g__Parabacteroides.s__Parabacteroides_distasonis	mvPuberty	127	-0.15	0.11	1.66E-01
k__Bacteria.p__Bacteroidetes.c__Bacteroidia.o__Bacteroidales.f__Bacteroidaceae.g__Bacteroides.s__Bacteroides_stercoris	mvPuberty	127	-0.18	0.13	1.68E-01
k__Bacteria.p__Proteobacteria.c__Deltaproteobacteria.o__Desulfovibrionales.f__Desulfovibrionaceae.g__Desulfovibrio	mvPuberty	120	-0.21	0.16	1.71E-01
k__Bacteria.p__Actinobacteria.c__Actinobacteria.o__Bifidobacteriales.f__Bifidobacteriaceae.g__Bifidobacterium.s__Bifidobacterium_catenuatum	mvPuberty	127	-0.33	0.25	1.78E-01
k__Bacteria.p__Proteobacteria.c__Gammaproteobacteria.o__Pasteurellales.f__Pasteurellaceae.g__Haemophilus	mvPuberty	127	0.22	0.17	1.83E-01
k__Bacteria.p__Bacteroidetes.c__Bacteroidia.o__Bacteroidales.f__Porphyromonadaceae.g__Parabacteroides.s__Parabacteroides_merdae	mvPuberty	127	-0.14	0.11	1.83E-01
k__Bacteria.p__Proteobacteria.c__Gammaproteobacteria.o__Pasteurellales.f__Pasteurellaceae	mvPuberty	127	0.22	0.16	1.84E-01
k__Bacteria.p__Proteobacteria.c__Gammaproteobacteria.o__Pasteurellales	mvPuberty	127	0.22	0.16	1.84E-01
k__Bacteria.p__Firmicutes.c__Clostridia.o__Clostridiales	mvPuberty	127	0.12	0.09	1.91E-01
k__Bacteria.p__Firmicutes.c__Clostridia.o__Clostridiales.f__Lachnospiraceae.g__Roseburia.s__Roseburia_unclassified	mvPuberty	127	-0.20	0.15	1.91E-01
k__Bacteria.p__Firmicutes	mvPuberty	110	0.13	0.10	2.12E-01
k__Bacteria.p__Firmicutes.c__Clostridia.o__Clostridiales.f__Lachnospiraceae.g__Blautia	mvPuberty	127	-0.12	0.10	2.14E-01
k__Bacteria.p__Proteobacteria.c__Gammaproteobacteria.o__Pasteurellales.f__Pasteurellaceae.g__Haemophilus.s__Haemophilus_parainfluenzae	mvPuberty	127	0.21	0.17	2.17E-01
k__Bacteria.p__Actinobacteria.c__Actinobacteria.o__Bifidobacteriales.f__Bifidobacteriaceae.g__Bifidobacterium.s__Bifidobacterium_longum	mvPuberty	127	-0.14	0.11	2.21E-01
k__Bacteria.p__Firmicutes.c__Erysipelotrichia.o__Erysipelotrichales.f__Erysipelotrichaceae.g__Erysipelotrichaceae_noname.s__Eubacterium_biforme	mvPuberty	127	-0.22	0.18	2.23E-01
k__Bacteria.p__Firmicutes.c__Clostridia.o__Clostridiales.f__Clostridiaceae	mvPuberty	127	0.16	0.13	2.31E-01
k__Bacteria.p__Firmicutes.c__Clostridia	mvPuberty	118	0.12	0.10	2.34E-01
k__Bacteria.p__Firmicutes.c__Clostridia.o__Clostridiales.f__Ruminococcaceae	mvPuberty	127	0.11	0.09	2.35E-01
k__Bacteria.p__Proteobacteria.c__Betaproteobacteria.o__Burkholderiales.f__Sutterellaceae	mvPuberty	127	-0.11	0.10	2.47E-01

k__Bacteria.p__Bacteroidetes.c__Bacteroidia.o__Bacteroidales.f__Prevotellaceae.g__Paraprevotella.s__Paraprevotella_clara	mvPuberty	119	-0.16	0.14	2.55E-01
k__Bacteria.p__Proteobacteria.c__Deltaproteobacteria.o__Desulfovibrionales.f__Desulfovibrionaceae.g__Bilophila.s__Bilophila_unclassified	mvPuberty	127	-0.11	0.10	2.55E-01
k__Bacteria.p__Firmicutes.c__Clostridia.o__Clostridiales.f__Lachnospiraceae.g__Roseburia	mvPuberty	127	-0.11	0.10	2.58E-01
k__Bacteria.p__Firmicutes.c__Erysipelotrichia.o__Erysipelotrichales.f__Erysipelotrichaceae.g__Holdemania	mvPuberty	127	-0.16	0.15	2.58E-01
k__Bacteria.p__Firmicutes.c__Negativicutes.o__Selenomonadales.f__Veillonellaceae.g__Dialister.s__Dialister_invisus	mvPuberty	127	0.17	0.15	2.63E-01
k__Bacteria.p__Firmicutes.c__Bacilli.o__Lactobacillales.f__Streptococcaceae.g__Streptococcus	mvPuberty	127	-0.16	0.14	2.64E-01
k__Bacteria.p__Proteobacteria.c__Gammaproteobacteria.o__Enterobacteriales.f__Enterobacteriaceae.g__Escherichia.s__Escherichia_unclassified	mvPuberty	127	0.13	0.12	2.66E-01
k__Bacteria.p__Proteobacteria.c__Deltaproteobacteria.o__Desulfovibrionales.f__Desulfovibrionaceae.g__Bilophila	mvPuberty	127	-0.11	0.10	2.66E-01
k__Bacteria.p__Bacteroidetes.c__Bacteroidia.o__Bacteroidales.f__Rikenellaceae.g__Alistipes.s__Alistipes_finegoldii	mvPuberty	127	-0.11	0.10	2.67E-01
k__Bacteria.p__Bacteroidetes	mvPuberty	127	-0.10	0.09	2.72E-01
k__Bacteria.p__Bacteroidetes.c__Bacteroidia.o__Bacteroidales	mvPuberty	127	-0.10	0.09	2.72E-01
k__Bacteria.p__Bacteroidetes.c__Bacteroidia	mvPuberty	127	-0.10	0.09	2.72E-01
k__Bacteria.p__Bacteroidetes.c__Bacteroidia.o__Bacteroidales.f__Porphyromonadaceae.g__Coprobacter.s__Coprobacter_fastidiosus	mvPuberty	127	-0.15	0.14	2.80E-01
k__Bacteria.p__Bacteroidetes.c__Bacteroidia.o__Bacteroidales.f__Bacteroidales_noname.g__Bacteroidales_noname.s__Bacteroidales_bacterium_ph8	mvPuberty	127	-0.10	0.10	2.82E-01
k__Bacteria.p__Firmicutes.c__Clostridia.o__Clostridiales.f__Lachnospiraceae.g__Lachnospiraceae_noname.s__Lachnospiraceae_bacterium_7_1_58FAA	mvPuberty	127	-0.10	0.10	2.82E-01
k__Bacteria.p__Bacteroidetes.c__Bacteroidia.o__Bacteroidales.f__Bacteroidaceae.g__Bacteroides.s__Bacteroides_plebeius	mvPuberty	127	0.22	0.21	2.86E-01
k__Bacteria.p__Bacteroidetes.c__Bacteroidia.o__Bacteroidales.f__Porphyromonadaceae.g__Coprobacter	mvPuberty	127	-0.15	0.14	2.89E-01
k__Bacteria.p__Bacteroidetes.c__Bacteroidia.o__Bacteroidales.f__Bacteroidales_noname	mvPuberty	127	-0.10	0.10	2.94E-01
k__Bacteria.p__Bacteroidetes.c__Bacteroidia.o__Bacteroidales.f__Bacteroidales_noname.g__Bacteroidales_noname	mvPuberty	127	-0.10	0.10	2.95E-01
k__Bacteria.p__Bacteroidetes.c__Bacteroidia.o__Bacteroidales.f__Bacteroidaceae.g__Bacteroides.s__Bacteroides_massiliensis	mvPuberty	122	0.16	0.16	2.97E-01
k__Bacteria.p__Proteobacteria.c__Betaproteobacteria.o__Burkholderiales.f__Sutterellaceae.g__Sutterella	mvPuberty	127	-0.11	0.11	3.04E-01
k__Bacteria.p__Bacteroidetes.c__Bacteroidia.o__Bacteroidales.f__Bacteroidaceae.g__Bacteroides.s__Bacteroides_vulgatus	mvPuberty	127	0.10	0.10	3.09E-01
k__Bacteria.p__Firmicutes.c__Clostridia.o__Clostridiales.f__Ruminococcaceae.g__Faecalibacterium	mvPuberty	127	0.09	0.09	3.26E-01
k__Bacteria.p__Bacteroidetes.c__Bacteroidia.o__Bacteroidales.f__Bacteroidaceae.g__Bacteroides.s__Bacteroides_thetaiotaomicron	mvPuberty	127	-0.11	0.11	3.27E-01
k__Bacteria.p__Bacteroidetes.c__Bacteroidia.o__Bacteroidales.f__Bacteroidaceae	mvPuberty	127	-0.09	0.09	3.27E-01
k__Bacteria.p__Bacteroidetes.c__Bacteroidia.o__Bacteroidales.f__Porphyromonadaceae.g__Parabacteroides.s__Parabacteroides_johnsonii	mvPuberty	127	-0.21	0.21	3.28E-01

k_Bacteria.p_Firmicutes.c_Bacilli.o_Lactobacillales.f_Lactobacillaceae.g_Lactobacillus.s_Lactobacillus_delbrueckii	mvPuberty	127	0.25	0.25	3.29E-01
k_Bacteria.p_Bacteroidetes.c_Bacteroidia.o_Bacteroidales.f_Bacteroidaceae.g_Bacteroides	mvPuberty	127	-0.09	0.09	3.29E-01
k_Bacteria.p_Proteobacteria.c_Betaproteobacteria.o_Burkholderiales.f_Sutterellaceae.g_Sutterella.s_Sutterella_wadsworthensis	mvPuberty	127	-0.11	0.11	3.31E-01
k_Bacteria.p_Firmicutes.c_Clostridia.o_Clostridiales.f_Ruminococcaceae.g_Faecalibacterium.s_Faecalibacterium_prausnitzii	mvPuberty	127	0.09	0.09	3.43E-01
k_Bacteria.p_Firmicutes.c_Bacilli.o_Lactobacillales.f_Streptococcaceae	mvPuberty	127	-0.13	0.14	3.47E-01
k_Bacteria.p_Bacteroidetes.c_Bacteroidia.o_Bacteroidales.f_Bacteroidaceae.g_Bacteroides.s_Bacteroides_nordii	mvPuberty	127	-0.26	0.28	3.63E-01
k_Bacteria.p_Verrucomicrobia.c_Verrucomicrobiae.o_Verrucomicrobiales.f_Verrucomicrobiaceae	mvPuberty	127	-0.11	0.12	3.65E-01
k_Bacteria.p_Verrucomicrobia.c_Verrucomicrobiae.o_Verrucomicrobiales	mvPuberty	127	-0.11	0.12	3.65E-01
k_Bacteria.p_Verrucomicrobia.c_Verrucomicrobiae	mvPuberty	127	-0.11	0.12	3.65E-01
k_Bacteria.p_Verrucomicrobia.c_Verrucomicrobiae.o_Verrucomicrobiales.f_Verrucomicrobiaceae.g_Akkermansia	mvPuberty	127	-0.11	0.12	3.67E-01
k_Bacteria.p_Firmicutes.c_Clostridia.o_Clostridiales.f_Lachnospiraceae.g_Lachnospiraceae_noname.s_Lachnospiraceae_bacterium_3_1_46FAA	mvPuberty	127	-0.14	0.15	3.68E-01
k_Bacteria.p_Firmicutes.c_Clostridia.o_Clostridiales.f_Lachnospiraceae.g_Dorea	mvPuberty	127	0.09	0.10	3.69E-01
k_Bacteria.p_Firmicutes.c_Clostridia.o_Clostridiales.f_Ruminococcaceae.g_Subdoligranulum.s_Subdoligranulum_unclassified	mvPuberty	113	0.09	0.10	3.73E-01
k_Bacteria.p_Proteobacteria.c_Betaproteobacteria.o_Burkholderiales.f_Burkholderiales_noname.g_Burkholderiales_noname	mvPuberty	127	-0.14	0.16	3.79E-01
k_Bacteria.p_Proteobacteria.c_Betaproteobacteria.o_Burkholderiales.f_Burkholderiales_noname	mvPuberty	127	-0.14	0.15	3.80E-01
k_Bacteria.p_Verrucomicrobia.c_Verrucomicrobiae.o_Verrucomicrobiales.f_Verrucomicrobiaceae.g_Akkermansia.s_Akkermansia_muciniphila	mvPuberty	126	-0.10	0.12	3.84E-01
k_Bacteria.p_Bacteroidetes.c_Bacteroidia.o_Bacteroidales.f_Prevotellaceae.g_Paraprevotella.s_Paraprevotella_xylaniphila	mvPuberty	127	-0.16	0.18	3.84E-01
k_Bacteria.p_Firmicutes.c_Bacilli	mvPuberty	127	-0.11	0.13	3.91E-01
k_Bacteria.p_Bacteroidetes.c_Bacteroidia.o_Bacteroidales.f_Rikenellaceae.g_Alistipes.s_Alistipes_putredinis	mvPuberty	127	-0.08	0.09	3.94E-01
k_Bacteria.p_Proteobacteria.c_Betaproteobacteria.o_Burkholderiales.f_Sutterellaceae.g_Sutterellaceae_unclassified	mvPuberty	127	-0.14	0.17	3.94E-01
k_Bacteria.p_Bacteroidetes.c_Bacteroidia.o_Bacteroidales.f_Rikenellaceae.g_Alistipes.s_Alistipes_onderdonkii	mvPuberty	127	-0.08	0.10	3.99E-01
k_Bacteria.p_Firmicutes.c_Clostridia.o_Clostridiales.f_Ruminococcaceae.g_Ruminococcaceae_noname.s_Ruminococcaceae_bacterium_D16	mvPuberty	120	-0.18	0.21	4.01E-01
k_Bacteria.p_Firmicutes.c_Clostridia.o_Clostridiales.f_Clostridiaceae.g_Clostridium.s_Clostridium_asparagiforme	mvPuberty	127	0.19	0.22	4.03E-01
k_Bacteria.p_Verrucomicrobia	mvPuberty	125	-0.10	0.12	4.04E-01
k_Bacteria.p_Bacteroidetes.c_Bacteroidia.o_Bacteroidales.f_Bacteroidaceae.g_Bacteroides.s_Bacteroides_fragilis	mvPuberty	126	-0.12	0.15	4.04E-01
k_Bacteria.p_Firmicutes.c_Clostridia.o_Clostridiales.f_Eubacteriaceae.g_Eubacterium.s_Eubacterium_eligens	mvPuberty	127	-0.09	0.10	4.04E-01

k__Bacteria.p__Firmicutes.c__Clostridia.o__Clostridiales.f__Ruminococcaceae.g__Ruminococcus.s__Ruminococcus_callidus	mvPuberty	117	0.14	0.17	4.06E-01
k__Bacteria.p__Firmicutes.c__Clostridia.o__Clostridiales.f__Clostridiales_noname.g__Pseudoflavonifactor.s__Pseudoflavonifactor_capillosus	mvPuberty	127	-0.11	0.14	4.28E-01
k__Bacteria.p__Firmicutes.c__Erysipelotrichia.o__Erysipelotrichales.f__Erysipelotrichaceae	mvPuberty	127	-0.09	0.11	4.30E-01
k__Bacteria.p__Firmicutes.c__Erysipelotrichia.o__Erysipelotrichales	mvPuberty	127	-0.09	0.11	4.30E-01
k__Bacteria.p__Firmicutes.c__Erysipelotrichia	mvPuberty	127	-0.09	0.11	4.31E-01
k__Bacteria.p__Firmicutes.c__Negativicutes.o__Selenomonadales.f__Veillonellaceae.g__Veillonella.s__Veillonella_unclassified	mvPuberty	127	-0.13	0.17	4.32E-01
k__Bacteria.p__Bacteroidetes.c__Bacteroidia.o__Bacteroidales.f__Porphyromonadaceae.g__Odoribacter	mvPuberty	127	-0.08	0.11	4.44E-01
k__Bacteria.p__Firmicutes.c__Clostridia.o__Clostridiales.f__Clostridiales_noname.g__Pseudoflavonifactor	mvPuberty	117	-0.11	0.14	4.45E-01
k__Bacteria.p__Firmicutes.c__Clostridia.o__Clostridiales.f__Ruminococcaceae.g__Anaerotruncus	mvPuberty	127	-0.17	0.22	4.57E-01
k__Bacteria.p__Firmicutes.c__Clostridia.o__Clostridiales.f__Lachnospiraceae.g__Lachnospiraceae_noname.s__Lachnospiraceae_bacterium_1_1_57FAA	mvPuberty	126	-0.19	0.25	4.58E-01
k__Bacteria.p__Firmicutes.c__Clostridia.o__Clostridiales.f__Ruminococcaceae.g__Ruminococcaceae_noname	mvPuberty	127	-0.15	0.21	4.70E-01
k__Bacteria.p__Firmicutes.c__Clostridia.o__Clostridiales.f__Lachnospiraceae.g__Dorea.s__Dorea_formicigenerans	mvPuberty	124	0.09	0.13	4.89E-01
k__Bacteria.p__Bacteroidetes.c__Bacteroidia.o__Bacteroidales.f__Bacteroidaceae.g__Bacteroides.s__Bacteroides_uniformis	mvPuberty	127	-0.07	0.09	4.90E-01
k__Bacteria.p__Proteobacteria.c__Betaproteobacteria.o__Burkholderiales	mvPuberty	127	-0.07	0.10	4.93E-01
k__Bacteria.p__Bacteroidetes.c__Bacteroidia.o__Bacteroidales.f__Bacteroidaceae.g__Bacteroides.s__Bacteroides_intestinalis	mvPuberty	127	-0.15	0.21	4.95E-01
k__Bacteria.p__Firmicutes.c__Negativicutes.o__Selenomonadales.f__Veillonellaceae.g__Dialister	mvPuberty	127	0.10	0.15	4.96E-01
k__Bacteria.p__Bacteroidetes.c__Bacteroidia.o__Bacteroidales.f__Rikenellaceae	mvPuberty	127	-0.06	0.09	5.12E-01
k__Bacteria.p__Firmicutes.c__Clostridia.o__Clostridiales.f__Ruminococcaceae.g__Ruminococcus.s__Ruminococcus_bromii	mvPuberty	126	0.08	0.11	5.13E-01
k__Bacteria.p__Bacteroidetes.c__Bacteroidia.o__Bacteroidales.f__Rikenellaceae.g__Alistipes	mvPuberty	127	-0.06	0.09	5.16E-01
k__Bacteria.p__Bacteroidetes.c__Bacteroidia.o__Bacteroidales.f__Bacteroidaceae.g__Bacteroides.s__Bacteroides_cellulosilyticus	mvPuberty	127	-0.09	0.15	5.25E-01
k__Bacteria.p__Firmicutes.c__Clostridia.o__Clostridiales.f__Lachnospiraceae.g__Butyrivibrio.s__Butyrivibrio_crossotus	mvPuberty	127	-0.16	0.26	5.30E-01
k__Bacteria.p__Firmicutes.c__Negativicutes	mvPuberty	127	0.06	0.10	5.42E-01
k__Bacteria.p__Firmicutes.c__Negativicutes.o__Selenomonadales	mvPuberty	127	0.06	0.10	5.42E-01
k__Bacteria.p__Firmicutes.c__Clostridia.o__Clostridiales.f__Lachnospiraceae.g__Butyrivibrio	mvPuberty	127	-0.15	0.26	5.45E-01
k__Bacteria.p__Proteobacteria.c__Betaproteobacteria.o__Burkholderiales.f__Oxalobacteraceae.g__Oxalobacter.s__Oxalobacter_formigenes	mvPuberty	125	0.09	0.16	5.53E-01
k__Bacteria.p__Firmicutes.c__Erysipelotrichia.o__Erysipelotrichales.f__Erysipelotrichaceae.g__Erysipelotrichaceae_noname	mvPuberty	127	-0.09	0.15	5.67E-01

k_Bacteria.p__Proteobacteria.c__Betaproteobacteria.o__Burkholderiales.f__Oxalobacteraceae.g__Oxalobacter	mvPuberty	127	0.09	0.15	5.68E-01
k_Bacteria.p__Firmicutes.c__Negativicutes.o__Selenomonadales.f__Acidaminococcaceae.g__Phascolarctobacterium	mvPuberty	127	0.10	0.17	5.68E-01
k_Bacteria.p__Firmicutes.c__Clostridia.o__Clostridiales.f__Eubacteriaceae.g__Eubacterium.s__Eubacterium_ventriosum	mvPuberty	127	0.09	0.16	5.68E-01
k_Bacteria.p__Proteobacteria.c__Betaproteobacteria.o__Burkholderiales.f__Oxalobacteraceae	mvPuberty	127	0.09	0.15	5.69E-01
k_Bacteria.p__Proteobacteria.c__Betaproteobacteria	mvPuberty	122	-0.05	0.09	5.71E-01
k_Bacteria.p__Firmicutes.c__Clostridia.o__Clostridiales.f__Lachnospiraceae.g__Dorea.s__Dorea_longicatena	mvPuberty	127	-0.06	0.11	5.71E-01
k_Bacteria.p__Bacteroidetes.c__Bacteroidia.o__Bacteroidales.f__Porphyromonadaceae.g__Parabacteroides.s__Parabacteroides_goldsteinii	mvPuberty	127	0.11	0.20	5.74E-01
k_Bacteria.p__Proteobacteria.c__Betaproteobacteria.o__Burkholderiales.f__Sutterellaceae.g__Parasutterella	mvPuberty	127	0.07	0.13	5.81E-01
k_Bacteria.p__Proteobacteria.c__Betaproteobacteria.o__Burkholderiales.f__Sutterellaceae.g__Parasutterella.s__Parasutterella_excrementihominis	mvPuberty	127	0.07	0.13	5.88E-01
k_Bacteria.p__Firmicutes.c__Negativicutes.o__Selenomonadales.f__Acidaminococcaceae.g__Phascolarctobacterium.s__Phascolarctobacterium_succinatutens	mvPuberty	127	0.09	0.17	5.91E-01
k_Bacteria.p__Firmicutes.c__Erysipelotrichia.o__Erysipelotrichales.f__Erysipelotrichaceae.g__Holdemania.s__Holdemania_filiformis	mvPuberty	127	0.09	0.18	5.92E-01
k_Bacteria.p__Firmicutes.c__Clostridia.o__Clostridiales.f__Lachnospiraceae.g__Blautia.s__Ruminococcus_obeum	mvPuberty	127	-0.06	0.12	5.95E-01
k_Bacteria.p__Firmicutes.c__Clostridia.o__Clostridiales.f__Ruminococcaceae.g__Ruminococcus	mvPuberty	127	0.05	0.10	6.00E-01
k_Bacteria.p__Bacteroidetes.c__Bacteroidia.o__Bacteroidales.f__Porphyromonadaceae.g__Barnesiella.s__Barnesiella_intestinihominis	mvPuberty	127	-0.05	0.10	6.13E-01
k_Bacteria.p__Firmicutes.c__Clostridia.o__Clostridiales.f__Eubacteriaceae.g__Eubacterium.s__Eubacterium_rectale	mvPuberty	127	-0.05	0.09	6.21E-01
k_Bacteria.p__Bacteroidetes.c__Bacteroidia.o__Bacteroidales.f__Porphyromonadaceae.g__Barnesiella	mvPuberty	127	-0.05	0.10	6.23E-01
k_Bacteria.p__Bacteroidetes.c__Bacteroidia.o__Bacteroidales.f__Bacteroidaceae.g__Bacteroides.s__Bacteroides_salyersiae	mvPuberty	127	-0.10	0.22	6.33E-01
k_Bacteria.p__Firmicutes.c__Clostridia.o__Clostridiales.f__Lachnospiraceae.g__Roseburia.s__Roseburia_intestinalis	mvPuberty	126	-0.06	0.13	6.37E-01
k_Bacteria.p__Proteobacteria	mvPuberty	108	-0.05	0.10	6.40E-01
k_Bacteria.p__Firmicutes.c__Bacilli.o__Lactobacillales	mvPuberty	127	-0.06	0.13	6.40E-01
k_Bacteria.p__Firmicutes.c__Clostridia.o__Clostridiales.f__Ruminococcaceae.g__Ruminococcus.s__Ruminococcus_lactaris	mvPuberty	126	-0.06	0.13	6.45E-01
k_Bacteria.p__Bacteroidetes.c__Bacteroidia.o__Bacteroidales.f__Porphyromonadaceae.g__Odoribacter.s__Odoribacter_splanchnicus	mvPuberty	127	-0.05	0.11	6.47E-01
k_Bacteria.p__Actinobacteria.c__Actinobacteria.o__Bifidobacteriales.f__Bifidobacteriaceae.g__Bifidobacterium.s__Bifidobacterium_bifidum	mvPuberty	127	-0.08	0.18	6.47E-01
k_Bacteria.p__Firmicutes.c__Negativicutes.o__Selenomonadales.f__Veillonellaceae	mvPuberty	127	0.05	0.12	6.61E-01
k_Bacteria.p__Firmicutes.c__Clostridia.o__Clostridiales.f__Eubacteriaceae.g__Eubacterium.s__Eubacterium_ramulus	mvPuberty	127	0.06	0.14	6.81E-01
k_Bacteria.p__Firmicutes.c__Negativicutes.o__Selenomonadales.f__Veillonellaceae.g__Veillonella	mvPuberty	127	-0.07	0.16	6.84E-01

k_Bacteria.p_Firmicutes.c_Clostridia.o_Clostridiales.f_Eubacteriaceae.g_Eubacterium.s_Eubacterium_hallii	mvPuberty	127	0.05	0.14	6.88E-01
k_Bacteria.p_Bacteroidetes.c_Bacteroidia.o_Bacteroidales.f_Rikenellaceae.g_Alistipes.s_Alistipes_shahii	mvPuberty	127	0.04	0.10	7.06E-01
k_Bacteria.p_Firmicutes.c_Clostridia.o_Clostridiales.f_Lachnospiraceae.g_Coproccoccus	mvPuberty	127	0.04	0.10	7.17E-01
k_Bacteria.p_Bacteroidetes.c_Bacteroidia.o_Bacteroidales.f_Prevotellaceae.g_Prevotella.s_Prevotella_copri	mvPuberty	127	0.04	0.11	7.23E-01
k_Bacteria.p_Firmicutes.c_Clostridia.o_Clostridiales.f_Clostridiales_noname.g_Flavonifractor	mvPuberty	127	-0.06	0.18	7.28E-01
k_Bacteria.p_Bacteroidetes.c_Bacteroidia.o_Bacteroidales.f_Bacteroidaceae.g_Bacteroides.s_Bacteroides_xylanisolvens	mvPuberty	124	-0.04	0.12	7.39E-01
k_Bacteria.p_Bacteroidetes.c_Bacteroidia.o_Bacteroidales.f_Bacteroidaceae.g_Bacteroides.s_Bacteroides_ovatus	mvPuberty	127	-0.03	0.10	7.41E-01
k_Bacteria.p_Firmicutes.c_Clostridia.o_Clostridiales.f_Lachnospiraceae.g_Coproccoccus.s_Coproccoccus_comes	mvPuberty	127	0.04	0.11	7.43E-01
k_Bacteria.p_Firmicutes.c_Clostridia.o_Clostridiales.f_Lachnospiraceae.g_Lachnospiraceae_noname.s_Lachnospiraceae_bacterium_5_1_63FAA	mvPuberty	127	-0.09	0.26	7.43E-01
k_Bacteria.p_Firmicutes.c_Bacilli.o_Lactobacillales.f_Lactobacillaceae.g_Lactobacillus	mvPuberty	127	0.06	0.21	7.63E-01
k_Bacteria.p_Proteobacteria.c_Gammaproteobacteria.o_Enterobacteriales.f_Enterobacteriaceae.g_Escherichia.s_Escherichia_coli	mvPuberty	127	0.04	0.13	7.64E-01
k_Bacteria.p_Bacteroidetes.c_Bacteroidia.o_Bacteroidales.f_Prevotellaceae.g_Paraprevotella.s_Paraprevotella_unclassified	mvPuberty	127	0.04	0.13	7.74E-01
k_Bacteria.p_Firmicutes.c_Bacilli.o_Lactobacillales.f_Lactobacillaceae	mvPuberty	127	0.06	0.21	7.76E-01
k_Bacteria.p_Firmicutes.c_Bacilli.o_Lactobacillales.f_Streptococcaceae.g_Streptococcus.s_Streptococcus_salivarius	mvPuberty	127	-0.06	0.22	7.83E-01
k_Bacteria.p_Actinobacteria.c_Actinobacteria.o_Bifidobacteriales.f_Bifidobacteriaceae.g_Bifidobacterium.s_Bifidobacterium_adolescentis	mvPuberty	127	-0.03	0.11	7.84E-01
k_Bacteria.p_Actinobacteria.c_Actinobacteria.o_Actinomycetales.f_Micrococcaceae.g_Rothia.s_Rothia_mucilaginosa	mvPuberty	127	0.05	0.19	7.88E-01
k_Bacteria.p_Actinobacteria.c_Actinobacteria.o_Actinomycetales	mvPuberty	127	-0.05	0.18	7.89E-01
k_Bacteria.p_Firmicutes.c_Clostridia.o_Clostridiales.f_Lachnospiraceae.g_Lachnospiraceae_noname.s_Lachnospiraceae_bacterium_8_1_57FAA	mvPuberty	127	0.05	0.19	8.04E-01
k_Bacteria.p_Proteobacteria.c_Deltaproteobacteria.o_Desulfovibrionales.f_Desulfovibrionaceae.g_Bilophila.s_Bilophila_wadsworthia	mvPuberty	127	-0.03	0.11	8.06E-01
k_Bacteria.p_Actinobacteria.c_Actinobacteria.o_Bifidobacteriales	mvPuberty	127	-0.02	0.10	8.15E-01
k_Bacteria.p_Actinobacteria.c_Actinobacteria.o_Bifidobacteriales.f_Bifidobacteriaceae	mvPuberty	127	-0.02	0.10	8.16E-01
k_Bacteria.p_Bacteroidetes.c_Bacteroidia.o_Bacteroidales.f_Bacteroidaceae.g_Bacteroides.s_Bacteroides_clarus	mvPuberty	127	-0.04	0.20	8.39E-01
k_Bacteria.p_Bacteroidetes.c_Bacteroidia.o_Bacteroidales.f_Rikenellaceae.g_Alistipes.s_Alistipes_indistinctus	mvPuberty	127	0.02	0.11	8.40E-01
k_Bacteria.p_Bacteroidetes.c_Bacteroidia.o_Bacteroidales.f_Prevotellaceae.g_Prevotella	mvPuberty	127	0.02	0.11	8.46E-01
k_Bacteria.p_Firmicutes.c_Clostridia.o_Clostridiales.f_Eubacteriaceae	mvPuberty	127	-0.02	0.09	8.49E-01
k_Bacteria.p_Actinobacteria.c_Actinobacteria.o_Actinomycetales.f_Micrococcaceae.g_Rothia	mvPuberty	127	0.03	0.19	8.52E-01

k_Bacteria.p_Firmicutes.c_Clostridia.o_Clostridiales.f_Eubacteriaceae.g_Eubacterium	mvPuberty	127	-0.02	0.09	8.56E-01
k_Bacteria.p_Actinobacteria.c_Actinobacteria.o_Coriobacteriales.f_Coriobacteriaceae.g_Gordonibacter	mvPuberty	127	-0.04	0.24	8.59E-01
k_Bacteria.p_Actinobacteria.c_Actinobacteria.o_Coriobacteriales.f_Coriobacteriaceae.g_Collinsella.s_Collinsella_aerofaciens	mvPuberty	127	0.02	0.10	8.65E-01
k_Bacteria.p_Firmicutes.c_Negativicutes.o_Selenomonadales.f_Acidaminococcaceae	mvPuberty	127	0.03	0.15	8.70E-01
k_Bacteria.p_Firmicutes.c_Clostridia.o_Clostridiales.f_Clostridiales_noname.g_Flavonifractor.s_Flavonifractor_plautii	mvPuberty	124	-0.03	0.18	8.77E-01
k_Bacteria.p_Actinobacteria.c_Actinobacteria.o_Coriobacteriales.f_Coriobacteriaceae.g_Gordonibacter.s_Gordonibacter_pamelaeae	mvPuberty	127	-0.04	0.24	8.80E-01
k_Bacteria.p_Firmicutes.c_Clostridia.o_Clostridiales.f_Clostridiales_noname	mvPuberty	127	0.02	0.12	8.88E-01
k_Bacteria.p_Actinobacteria.c_Actinobacteria.o_Actinomycetales.f_Micrococcaceae	mvPuberty	127	0.03	0.19	8.88E-01
k_Bacteria.p_Bacteroidetes.c_Bacteroidia.o_Bacteroidales.f_Bacteroidaceae.g_Bacteroides.s_Bacteroides_eggerthii	mvPuberty	127	-0.02	0.16	8.89E-01
k_Bacteria.p_Actinobacteria.c_Actinobacteria.o_Coriobacteriales.f_Coriobacteriaceae.g_Collinsella	mvPuberty	127	-0.01	0.10	8.95E-01
k_Bacteria.p_Bacteroidetes.c_Bacteroidia.o_Bacteroidales.f_Rikenellaceae.g_Alistipes.s_Alistipes_sp_API1	mvPuberty	127	-0.02	0.15	8.96E-01
k_Bacteria.p_Firmicutes.c_Clostridia.o_Clostridiales.f_Lachnospiraceae.g_Roseburia.s_Roseburia_hominis	mvPuberty	127	0.01	0.11	9.11E-01
k_Bacteria.p_Firmicutes.c_Clostridia.o_Clostridiales.f_Lachnospiraceae	mvPuberty	127	-0.01	0.10	9.17E-01
k_Bacteria.p_Firmicutes.c_Clostridia.o_Clostridiales.f_Clostridiaceae.g_Clostridium.s_Clostridium_leptum	mvPuberty	127	0.02	0.16	9.22E-01
k_Bacteria.p_Actinobacteria.c_Actinobacteria.o_Bifidobacteriales.f_Bifidobacteriaceae.g_Bifidobacterium	mvPuberty	127	-0.01	0.10	9.23E-01
k_Bacteria.p_Bacteroidetes.c_Bacteroidia.o_Bacteroidales.f_Bacteroidaceae.g_Bacteroides.s_Bacteroides_finegoldii	mvPuberty	127	0.02	0.20	9.35E-01
k_Bacteria.p_Actinobacteria.c_Actinobacteria.o_Coriobacteriales.f_Coriobacteriaceae	mvPuberty	127	0.01	0.10	9.40E-01
k_Bacteria.p_Actinobacteria.c_Actinobacteria.o_Coriobacteriales	mvPuberty	127	0.01	0.10	9.40E-01
k_Bacteria.p_Bacteroidetes.c_Bacteroidia.o_Bacteroidales.f_Prevotellaceae.g_Paraprevotella	mvPuberty	127	0.01	0.13	9.57E-01
k_Bacteria.p_Bacteroidetes.c_Bacteroidia.o_Bacteroidales.f_Prevotellaceae	mvPuberty	127	0.00	0.10	9.66E-01
k_Bacteria.p_Firmicutes.c_Clostridia.o_Clostridiales.f_Lachnospiraceae.g_Coprococcus.s_Coprococcus_sp_ART55_1	mvPuberty	126	-0.01	0.25	9.67E-01
k_Bacteria.p_Firmicutes.c_Bacilli.o_Lactobacillales.f_Streptococcaceae.g_Streptococcus.s_Streptococcus_thermophilus	mvPuberty	122	0.01	0.19	9.70E-01
k_Bacteria.p_Firmicutes.c_Clostridia.o_Clostridiales.f_Lachnospiraceae.g_Coprococcus.s_Coprococcus_catus	mvPuberty	127	0.00	0.14	9.74E-01
k_Bacteria.p_Firmicutes.c_Clostridia.o_Clostridiales.f_Lachnospiraceae.g_Blautia.s_Ruminococcus_torques	mvPuberty	127	0.00	0.13	9.94E-01
k_Bacteria.p_Actinobacteria.c_Actinobacteria	mvPuberty	127	0.00	0.10	9.94E-01
k_Bacteria.p_Actinobacteria	mvPuberty	127	0.00	0.10	9.94E-01

REVIEWERS' COMMENTS

Reviewer #2 (Remarks to the Author):

My remaining comments have now been adequately addressed.